# A NEW CHARACTERIZATION OF THE EDGE OF STABILITY BASED ON A SHARPNESS MEASURE AWARE OF BATCH GRADIENT DISTRIBUTION

**Sungyoon Lee & Cheongjae Jang**
Hanyang University, Seoul
{sungyoonlee,cjjang}@hanyang.ac.kr

## ABSTRACT

For full-batch gradient descent (GD), it has been empirically shown that the sharpness, the top eigenvalue of the Hessian, increases and then hovers above 2/(learning rate), and this is called "the edge of stability" phenomenon. However, it is unclear why the sharpness is somewhat larger than 2/(learning rate) and how this can be extended to general mini-batch stochastic gradient descent (SGD). We propose a new sharpness measure (interaction-aware-sharpness) aware of the *interaction* between the batch gradient distribution and the loss landscape geometry. This leads to a more refined and general characterization of the edge of stability for SGD. Moreover, based on the analysis of a concentration measure of the batch gradient, we propose a more accurate scaling rule, Linear and Saturation Scaling Rule (LSSR), between batch size and learning rate.

## 1 INTRODUCTION

For full-batch GD, it has been empirically observed that the sharpness, the top eigenvalue of the Hessian, increases and then hovers above 2/(learning rate) (Cohen et al., 2021) as the training proceeds. This observation can provide a link between two empirical results regarding generalization, (i) using larger learning rates for GD can generalize better (Bjorck et al., 2018; Li et al., 2019b; Lewkowycz et al., 2020; Smith et al., 2020) and (ii) minima with low sharpness tend to generalize better (Hochreiter & Schmidhuber, 1997; Keskar et al., 2017). This observation has a significant implication in existing neural network optimization convergence analyses since it is contrary to the frequent assumption that 'the learning rate is less than $2/\beta$ (here $\beta$ is an upper bound of the Hessian top eigenvalue)', which ensures the decrease in the training loss (Nesterov, 2003; Schmidt, 2014; Martens, 2014; Bottou et al., 2018). Even though the training loss evolves non-monotonically over short timescales due to the violation of the assumption, interestingly, the loss is observed to decrease over long timescales consistently. This regime in which GD typically occurs has been referred to as 'the edge of stability (EoS)' (Cohen et al., 2021).

There remain many aspects that are not clearly explained about the EoS regime. For example, it is not clear why and to what extent the sharpness hovers above 2/(learning rate). Moreover, the inherent mechanism is not yet elucidated for the unstable optimization to occur at the EoS consistently while prevented from entirely diverging. How this phenomenon can be generalized beyond GD, especially to mini-batch SGD, is still an open question.

In this paper we provide a new characterization of the EoS for SGD, which can serve as an answer to the above questions. As a tool to analyze the optimization process of SGD, we first propose a sharpness measure of neural network loss landscape aware of SGD batch gradient distribution (hence capturing the interaction between SGD and the loss landscape), which we refer to as the interaction-aware-sharpness (IAS) (Section 2). Based on this measure, we define the stable and unstable regions in the neural network parameter space. We then scrutinize both theoretically and empirically the transition process of the iterate from the stable to the unstable region (Section 4.1) and the mechanism to escape from the unstable region, i.e., how the optimization can occur at the EoS. We interpret the latter mechanism based on the non-quadraticity of the loss and the presence of asymmetric valleys in the loss landscape (He et al., 2019) (Section 4.2).

Based on these analyses, we propose the notion of implicit interaction regularization (IIR), i.e., the IAS is implicitly bounded during SGD, as an implicit bias of SGD (Section 4.3). The value that IAS is bounded by is the ratio of a concentration measure of the batch gradient distribution of SGD to the learning rate. This is a more refined characterization of the EoS, as it shows that IAS does not hover above a certain value, but rather hovers around. More importantly, it can be naturally applied to SGD since we do not make any impractical assumptions on the batch size or learning rate.

Our new characterization of the EoS leads to a novel scaling rule between batch size and learning rate, from the idea of preserving a similar level of IIR (Section 5). This scaling rule, referred to as the Linear and Saturation Scaling Rule (LSSR), recovers the well-known linear scaling rule (LSR) (Jastrzębski et al., 2017; Masters & Luschi, 2018; Zhang et al., 2019; Shallue et al., 2018; Smith et al., 2020; 2021) for small batch sizes and reduces to no scaling (due to saturation) for large batch sizes.

## 2 GRADIENT DISTRIBUTION AND LOSS LANDSCAPE

In this section, we review some concepts required for further discussion. See Appendix A for a quick reference for the notations. To simplify the notations, we often omit the dependence on some variables and the subscript of the expectation operation when clear from the context.

For a learning task, we use a parameterized model with model parameter $\theta \in \Theta \subset \mathbb{R}^m$. Then we train the model using training data $\mathcal{D} = \{x_i\}_{i=1}^n$ and a loss function $\ell(x; \theta)$. We denote the (total) training loss by $L(\theta) \equiv \frac{1}{n} \sum_{i=1}^n \ell(x_i; \theta)$ for training data $\mathcal{D}$. At time step $t$, we update the parameter $\theta_t$ using GD: $\theta_{t+1} = \theta_t - \eta \nabla_\theta L(\theta_t)$ with a learning rate $\eta > 0$, or using SGD: $\theta_{t+1} = \theta_t - \eta g_\xi(\theta_t)$ with a mini-batch gradient $g_\xi(\theta_t) \equiv \frac{1}{b} \sum_{x \in \mathcal{B}_\xi^t} \nabla_\theta \ell(x; \theta_t) \in \mathbb{R}^m$ for a mini-batch $\mathcal{B}_\xi^t \subset \mathcal{D}$ of size $b$ ($1 \le b \le n$). Here, we use the subscript $\xi$ to denote the random batch sampling procedure.

Now, we are ready to introduce some important matrices, $C_b, S_b$, and $H$. First, we define the covariance $C_b(\theta) \equiv \mathrm{Var}_\xi[g_\xi(\theta)] = \mathbb{E}_\xi \left[ (g_\xi(\theta) - \mathbb{E}_\xi[g_\xi(\theta)]) (g_\xi(\theta) - \mathbb{E}_\xi[g_\xi(\theta)])^\top \right] \in \mathbb{R}^{m \times m}$ and the second moment $S_b(\theta) \equiv \mathbb{E}_\xi[g_\xi(\theta)g_\xi(\theta)^\top] \in \mathbb{R}^{m \times m}$ of the mini-batch gradient $g_\xi(\theta)$ over batch sampling for a batch size $1 \le b \le n$.[1] The covariance $C_b$ and the second moment $S_b$ satisfy not only $C_b = S_b - S_n$ but also the following equation (Hoffer et al., 2017; Li et al., 2017; Wu et al., 2020):

$$C_b = \frac{\gamma_{n,b}}{b}(S_1 - S_n) = \frac{\gamma_{n,b}}{b}C_1, \text{ where } \gamma_{n,b} = \begin{cases} \frac{n-b}{n-1} & \text{for sampling } \textit{without} \text{ replacement} \\ 1 & \text{for sampling } \textit{with} \text{ replacement} \end{cases}. \quad (1)$$

We provide a self-contained proof of (1) in Appendix B.1. We note that, for sampling without replacement, many previous works approximate $\gamma_{n,b} \approx 1$ assuming $b \ll n$ (Jastrzębski et al., 2017; Hoffer et al., 2017; Smith et al., 2021), but we consider the whole range of $1 \le b \le n$ ($0 \le \gamma_{n,b} \le 1$ with $\gamma_{n,1} = 1$ and $\gamma_{n,n} = 0$). Second, we define the Hessian $H(\theta) = \nabla_\theta^2 L(\theta) = \frac{1}{n} \sum_{i=1}^n \nabla_\theta^2 \ell(x_i; \theta) \in \mathbb{R}^{m \times m}$ and denote the $i$-th largest eigenvalue and its corresponding normalized eigenvector by $\lambda_i(H) \in \mathbb{R}$ and $q_i(H) \in \mathbb{R}^m$, respectively, for $i = 1, \cdots, m$. The operator norm $\|H\| \equiv \sup_{\|u\|=1} \|Hu\|$ of $H$ is equivalent to the top eigenvalue $\lambda_1$. We emphasize that $C_b$ and $S_b$ represent the stochasticity of the batch gradients, and $H$ represents the loss landscape geometry.

Therefore, we can write one of our goals as follows: *we aim to understand how the loss landscape geometry ($H$) and the gradient distribution ($S_b$) interact with each other during SGD training.* We investigate this "interaction" in terms of matrix multiplication $HS_b$. To be specific, we consider the trace $\mathrm{tr}(HS_b)$ and its normalized value $\frac{\mathrm{tr}(HS_b)}{\mathrm{tr}(S_b)}$, and we call the latter *interaction-aware sharpness*:

**Definition 1** (Interaction-Aware Sharpness (IAS))**.**

$$\|H\|_{S_b} \equiv \frac{\mathrm{tr}(HS_b)}{\mathrm{tr}(S_b)}. \quad (2)$$

Here, $\mathrm{tr}(HS_b) \le \|H\| \mathrm{tr}(S_b)$, i.e., $\|H\|_{S_b} \le \|H\|$, and the equality holds only when every $g_\xi$ is aligned in the direction of the top eigenvector $q_1$ of $H$.

---

[1] These two matrices $C_b$ and $S_b$ are often called the second *central* and *non-central* moments, respectively. But to avoid confusion, we use the term "second moment" only for the non-central $S_b$.

## 3 RELATED WORK

Some studies investigate the interaction between the gradient distribution and the loss landscape geometry represented by $\mathrm{tr}(HS_b)$ in the context of escaping efficiency (Zhu et al., 2019, Section 3.1), stationarity (Yaida, 2019, Section 2.2), and convergence (Thomas et al., 2020, Section 3.1.1). However, they require some additional assumptions like stochastic differential equation (SDE) approximation of SGD (Zhu et al., 2019), the existence of a stationary-state distribution of the model parameter (Yaida, 2019, Section 2.3.4), and strong convexity of the training loss function (Thomas et al., 2020), respectively. In this paper, we provide a new insight into the interaction $\mathrm{tr}(HS_b)$ without these assumptions.

Convergence of full-batch GD ($b = n$) has been instead analyzed with an upper bound on the interaction $\mathrm{tr}(HS_n)$ with further assumptions for the stable optimization, such as $\beta$-smoothness of the objective and $0 < \eta < \frac{2}{\beta}$ (e.g., $\eta = \frac{1}{\beta}$) (Nesterov, 2003; Schmidt, 2014; Martens, 2014; Bottou et al., 2018). [2] However, it may lose useful information of the interaction between $H$ and $S_n$. Moreover, when we train a standard neural network with GD in practice, $\|H\|(\leq \beta)$ increases in the early phase of training and the iterate enters the EoS where $\|H\| \gtrsim \frac{2}{\eta}$, i.e., $\eta \gtrsim \frac{2}{\|H\|} \geq \frac{2}{\beta}$. This contradicts with the assumption for stable optimization and the iterate exhibits unstable behavior with a non-monotonically decreasing loss (Xing et al., 2018; Wu et al., 2018; Cohen et al., 2021). We further extend this discussion of unstable dynamics for GD at the EoS to the case of SGD.

From the generalization perspective, many studies focus on the implicit bias of SGD toward a better generalization (Neyshabur, 2017; Zhang et al., 2021; Gunasekar et al., 2017; Soudry et al., 2018; Jastrzębski et al., 2020; 2021; Barrett & Dherin, 2021; Smith et al., 2021). There are mainly two factors known to correlate with the generalization performance: the batch gradient distribution during training (Hoffer et al., 2017; Jastrzębski et al., 2017; Smith & Le, 2018; Zhu et al., 2019) and the sharpness of the loss landscape at the minimum (Hochreiter & Schmidhuber, 1997; Keskar et al., 2017; Dinh et al., 2017; Jiang et al., 2020; Foret et al., 2021; Kwon et al., 2021). We provide a link between the batch gradient distribution and the sharpness that the model is implicitly regularized to have a low sharpness when the second moment of the batch gradient is large (see Section 4.3).

We provide further discussion to reconcile our arguments with some previous studies in Appendix D

## 4 A NEW CHARACTERIZATION OF THE EDGE OF STABILITY

### 4.1 UNSTABLE OPTIMIZATION

Using the second-order Taylor expansion of the total training loss $L(\theta)$ at $\theta_t$, the change in the loss $L_t = L(\theta_t)$ as the SGD iterate moves from $\theta_t$ to $\theta_{t+1}$ at time step $t$ can be expressed as follows:

$$L_{t+1} - L_t = -\eta \nabla L(\theta_t)^\top g_\xi + \frac{\eta^2}{2} g_\xi^\top H(\theta_t) g_\xi + O(\|\delta_t\|^3), \tag{3}$$

where $\delta_t = \theta_{t+1} - \theta_t = -\eta g_\xi$. Given $\theta_t$, the expected loss difference over batch sampling $\xi$ is

$$\mathbb{E}_\xi[L_{t+1}] - L_t \tag{4}$$

$$= -\eta \nabla L(\theta_t)^\top \mathbb{E}_\xi[g_\xi] + \frac{\eta^2}{2} \mathbb{E}_\xi[g_\xi^\top H(\theta_t) g_\xi] + \epsilon \quad \text{(Taking } \mathbb{E}_\xi \text{ for the both sides of (3))} \tag{5}$$

$$= -\eta \|\nabla L(\theta_t)\|^2 + \frac{\eta^2}{2} \mathrm{tr}\left(\mathbb{E}_\xi[H(\theta_t) g_\xi g_\xi^\top]\right) + \epsilon \quad (\mathbb{E}_\xi[g_\xi] = \nabla L \text{ and } u^\top v = \mathrm{tr}(vu^\top)) \tag{6}$$

$$= \frac{\eta^2}{2} \mathrm{tr}(S_n) \left[\frac{\mathrm{tr}(HS_b)}{\mathrm{tr}(S_n)} - \frac{2}{\eta}\right] + \epsilon \quad \text{(by definition of } S_b \text{ and } S_n), \tag{7}$$

where $\epsilon = O(\mathbb{E}_\xi[\|\delta_t\|^3])$ and $\mathbb{E}_\xi[X]$ is the conditional expectation of $X$ given $\theta_t$. We consider a bounded region with a finite maximum loss near and including $\theta_t$ in which the loss is approximately quadratic. Then, when the following *instability condition* is met within the region, the loss (close to the expected loss) tends to increase and the iterate tends to escape from the region:

---

[2]$L(\theta_{t+1}) - L(\theta_t) \leq \nabla L^\top (\theta_{t+1} - \theta_t) + \frac{\beta}{2}\|\theta_{t+1} - \theta_t\|^2 = -\eta\|\nabla L\|^2 + \frac{\beta\eta^2}{2}\|\nabla L\|^2 = -\eta(1 - \frac{\beta\eta}{2})\|\nabla L\|^2$ and thus the loss monotonically decreases when $0 < \eta < \frac{2}{\beta}$.

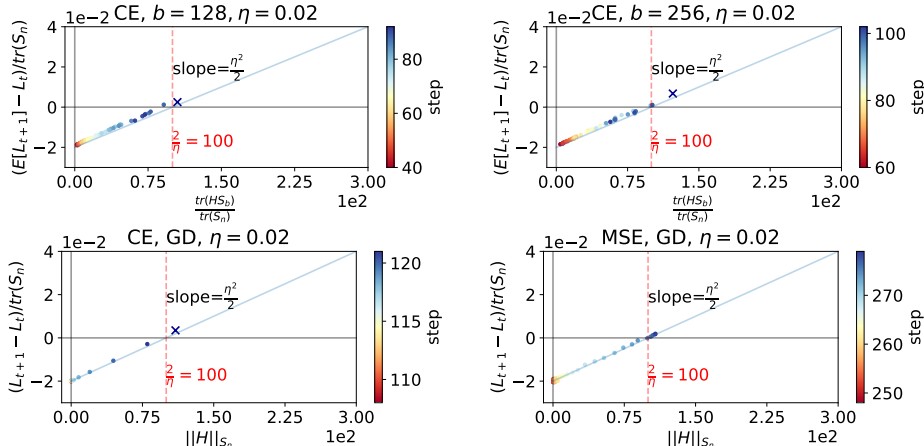

Figure 1: **[An empirical validation of (7) for SGD (top) and (12) for GD (bottom)]** In the early phase, until the iterate enters the unstable region, it validates (7) and (12) with the blue line with the slope $\frac{\eta^2}{2}$ and x-intercept $\frac{2}{\eta}$. For GD (bottom), they are plotted *after* $\|H\|$ exceeds $\frac{2}{\eta}$ after which $\|H\|_{S_n}$ starts to increase from 0 to $\frac{2}{\eta}$ in a few steps. For cross-entropy (CE) loss, we mark the end point with 'x' when the iterate enters the unstable region. For mean squared error (MSE) loss (bottom right), we plot the graph for a few more steps after the iterate enters the EoS. We train 6CNN on CIFAR-10-8k with $\eta = 0.02$ (see Remark at the end of Section 4.2).

**Theorem 1.** *For SGD on a quadratic $L$, the expected loss increases, i.e., $\mathbb{E}_\xi[L_{t+1}] - L_t > 0$, if and only if $\theta_t$ satisfies the instability condition $\frac{\text{tr}(HS_b)}{\text{tr}(S_n)} > \frac{2}{\eta}$. Furthermore, if the batch gradient $g_\xi$ is normally distributed, then the following inequalities hold for any positive $x > 0$:*

$$\mathbb{P}\left(L_{t+1} - \mathbb{E}_\xi[L_{t+1}] \geq \sqrt{2\beta x} + \frac{\eta^2 \gamma_{n,b}}{b} \|H\| \|C_1\| x \mid \theta_t\right) \leq \exp(-x), \quad (8)$$

$$\mathbb{P}\left(L_{t+1} - \mathbb{E}_\xi[L_{t+1}] \leq -\sqrt{2\beta x} \mid \theta_t\right) \leq \exp(-x), \quad (9)$$

*where $\beta = \frac{\eta^2 \gamma_{n,b}}{b}\left(v^\top C_1 v + \frac{\eta^2 \gamma_{n,b}}{2b}\text{tr}(HC_1HC_1)\right)$ and $v = (I - \eta H)\nabla L$.*

The proof is deferred to Appendix B.2. From the above theorem, we define *unstable* and *stable region*:

$$\mathbb{U} \equiv \{\theta \in \Theta : \frac{\text{tr}(HS_b)}{\text{tr}(S_n)} > \frac{2}{\eta}\} \text{ and } \mathbb{S} \equiv \mathbb{U}^c, \quad (10)$$

respectively. It has been empirically shown that, for full-batch GD, $\|H\|$ increases and then hovers above $2/\eta$ and Cohen et al. (2021) mark the EoS with $\{\theta \in \Theta : \|H(\theta)\| = \frac{2}{\eta}\}$, but we mark with

$$\partial\mathbb{S} = \{\theta \in \Theta : \frac{\text{tr}(HS_b)}{\text{tr}(S_n)} = \frac{2}{\eta}\}. \quad (11)$$

We emphasize the superiority of our $\partial\mathbb{S}$ over the previous EoS that (i) $\partial\mathbb{S}$ provides a clearer "edge" since it considers the interaction between the gradient direction and the Hessian and that (ii) it is more general since it applies to SGD with any batch size $1 \leq b \leq n$.

Figure 1 (top row) empirically validates (7), showing the normalized loss difference $\frac{\mathbb{E}_\xi[L_{t+1}] - L_t}{\text{tr}(S_n)}$ against $\frac{\text{tr}(HS_b)}{\text{tr}(S_n)}$ in the early phase of training before entering the unstable region. This result implies that the training loss $L(\theta)$ is approximately locally quadratic, i.e., $\epsilon \approx 0$, in the early phase.

Especially, for full-batch GD ($b = n$), the instability condition can be rewritten as $\|H\|_{S_n} > \frac{2}{\eta}$ and we have the following relationship between the loss difference $L_{t+1} - L_t$ and $\|H\|_{S_n}$ from (7):

$$L_{t+1} - L_t = \frac{\eta^2}{2}\text{tr}(S_n)\left(\|H\|_{S_n} - \frac{2}{\eta}\right) + \epsilon. \quad (12)$$

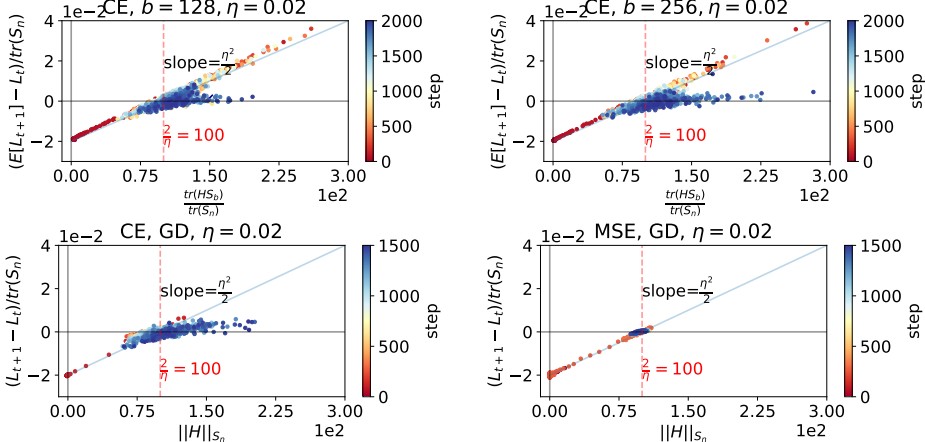

Figure 2: **[Non-quadraticity and overestimation]** The normalized loss difference $\frac{\mathbb{E}_\xi[L_{t+1}]-L_t}{\text{tr}(S_n)}$ against $\frac{\text{tr}(HS_b)}{\text{tr}(S_n)}$ during training. After the iterate enters the EoS, it often shows a more gentle slope than $\frac{\eta^2}{2}$, especially in the unstable region. See the caption of Figure 1.

Figure 1 (bottom row) shows $\|H\|_{S_n}$ soars from 0 in a few steps after $\|H\|$ exceeds $\frac{2}{\eta}$, satisfying (12) with $\epsilon \approx 0$, before the iterate enters the unstable region. This result is consistent with the following theorem for a quadratic loss $L$ and generalized momentum GD with $(\beta_1, \beta_2)$:

$$\delta_t = \beta_1 \delta_{t-1} - \eta \nabla_\theta L(\theta_t + \beta_2 \delta_{t-1}), \tag{13}$$

$$\theta_{t+1} = \theta_t + \delta_t, \tag{14}$$

where $\beta_1, \beta_2 \in [0, 1)$. Here, we have vanilla GD when $\beta_1 = \beta_2 = 0$, Polyak momentum when $\beta_1 \in (0, 1)$ and $\beta_2 = 0$ (Polyak, 1963), and Nesterov momentum when $\beta_1 = \beta_2 \in (0, 1)$ (Nesterov, 1983). We will focus on the vanilla GD as it can be easily extended to the generalized momentum variants (see Appendix C.6 for details). The proof is deferred to Appendix B.3.

**Theorem 2.** *For generalized momentum GD with $(\beta_1, \beta_2)$ on a quadratic $L$, if $0 < \lambda_i < \frac{2}{\eta}\gamma(\beta_1, \beta_2) < \lambda_1$ for all $i \neq 1$ where $\gamma(\beta_1, \beta_2) = \frac{1+\beta_1}{1+2\beta_2}$, then $q_1^\top \delta_t$ oscillates and diverges with the exponential growth of $|q_1^\top \delta_t| = \Theta(e^{ct})$ for some $c > 0$. Moreover, $|\cos(q_1, \delta_t)|$ and $\|H\|_{S_n}$ increase to 1 and $\lambda_1$, as $t \to \infty$, respectively, with $1 - |\cos(q_1, \delta_t)|, \lambda_1 - \|H\|_{S_n} = O(e^{-2ct})$.*

Note that $\gamma(\beta_1, \beta_2) = 1$ for the vanilla GD. To summarize, if $\|H\|$ exceeds $\frac{2}{\eta}$, then $\|H\|_{S_n}$ increases towards $\|H\|$ with the exponential convergence rate and also exceeds $\frac{2}{\eta}$ in a few steps, i.e., the iterate enters the unstable region $\mathbb{U}$. Together with Theorem 1, if we consider a bounded subregion $\mathbb{V} \subset \mathbb{U}$ with a finite maximum loss near and including $\theta_t$ in which the loss is approximately quadratic, then the iterate tends to escape from the region $\mathbb{V}$ (Nar & Sastry, 2018; Wu et al., 2018; Cohen et al., 2021).

## 4.2 NON-QUADRATICITY, ASYMMETRIC VALLEYS AND THE EDGE OF STABILITY

In the previous section, we have shown that the training loss is approximately locally quadratic *before* the iterate enters the unstable region. However, *after* the iterate enters the unstable region, i.e., $\frac{\text{tr}(HS_b)}{\text{tr}(S_n)}$ reaches and exceeds $\frac{2}{\eta}$, the step size is relatively large for the sharp loss landscape so that the iterate jumps across the valley (Jastrzębski et al., 2019), and the higher-order terms $\epsilon$ in (7) and (12) become non-negligible and cause a different behavior of the iterate than in the stable region.

Figure 2 shows empirical evidences for the *non-quadraticity*. After the SGD/GD iterate enters the unstable region, when the instability condition $\frac{\text{tr}(HS_b)}{\text{tr}(S_n)} > \frac{2}{\eta}$ is met, the normalized increase in the loss $\left|\frac{\mathbb{E}_\xi[L_{t+1}]-L_t}{\text{tr}(S_n)}\right|$ is often smaller than $\frac{\eta^2}{2}\left|\frac{\text{tr}(HS_b)}{\text{tr}(S_n)} - \frac{2}{\eta}\right|$ from (7) and (12) (blue line), which is the case for a locally quadratic function. This results in a gentle slope less than $\frac{\eta^2}{2}$.

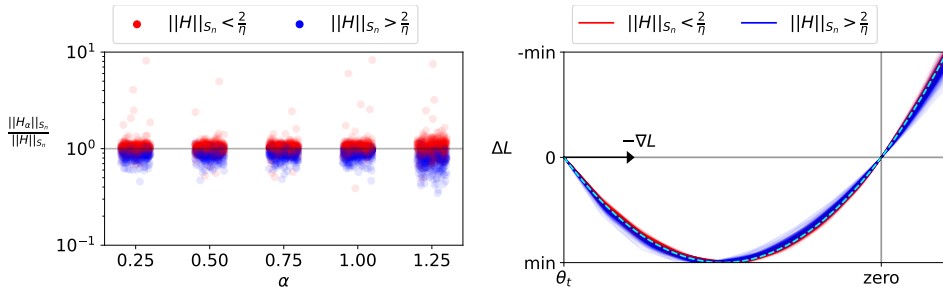

Figure 3: **[Asymmetric valleys]** Left: The ratio $\frac{\|H_\alpha\|_{S_n}}{\|H\|_{S_n}}$ where $H_\alpha = H(\theta - \alpha\eta\nabla L(\theta))$ for $\alpha = \frac{1}{4} \times [1,2,3,4,5]$ for each $t$ during training. When $\|H\|_{S_n} < \frac{2}{\eta}$ (red), $\|H_\alpha\|_{S_n}$ is usually larger than $\|H\|_{S_n}$. On the other hand, when $\|H\|_{S_n} > \frac{2}{\eta}$ (blue), $\|H_\alpha\|_{S_n}$ is usually smaller than $\|H\|_{S_n}$. Right: The training loss difference along the gradient descent direction, for each $\theta_t$. Each plot is normalized and translated to have the same minimum value and the same zero where $\Delta L(\text{zero}) = L(\text{zero}) - L(\theta_t) = 0$. We also plot the quadratic baseline (cyan dashed curve). When $\|H\|_{S_n} < \frac{2}{\eta}$ (red), it usually becomes sharper across the valley (right-shifted), while the opposite is observed (left-shifted) when $\|H\|_{S_n} > \frac{2}{\eta}$ (blue). We train 6CNN using GD with $\eta = 0.04$.

We hypothesize that due to this non-quadraticity of the training loss, the iterate is discouraged from staying within the unstable region. Note that, for a globally quadratic loss, when the iterate is in the unstable region, it diverges within the unstable region. Figure 3 demonstrates the asymmetric valley (He et al., 2019) that one side is sharp and the other is flat. In Figure 3 (left), we evaluate the directional sharpness $\|H_\alpha\|_{S_n}$ along the gradient descent direction $-\eta\nabla L(\theta)$ where $H_\alpha \equiv H(\theta - \alpha\eta\nabla L(\theta))$ for $\alpha \in \frac{1}{4} \times [1,2,3,4,5]$, and compare $\|H_\alpha\|_{S_n(\theta)}$ with $\|H\|_{S_n(\theta)}$. At the sharp side, it has a high $\|H\|_{S_n} > \frac{2}{\eta}$ (blue) with the gradient $\nabla L$ and the top eigenvector $q_1(H)$ of the Hessian being highly aligned (cf. Theorem 2). However, when the loss landscape gets far from being quadratic, the Hessian and its top eigenvector can change abruptly, $q_1(H_\alpha)$ would not always be aligned with $q_1(H)$ and $\nabla L(\theta)$, and $\|H_\alpha\|_{S_n}$ tends to decrease. This would be a possible explanation for the tendency of decreasing and then oscillating $\|H\|_{S_n}$. See Appendix C.3 for detailed empirical evidences of the above arguments. Figure 3 (right) similarly shows that when the iterate is at a sharp side of the valley, it tends to jump to the other side of a flatter area, and vice versa.

To summarize, we make the following observations for GD in order: (i) $\|H\|$ increases in the beginning (the *progressive sharpening* (Jastrzębski et al., 2019; 2020; Cohen et al., 2021)), (ii) $\|H\|$ exceeds $\frac{2}{\eta}$, (iii) the gradient $\nabla L$ becomes more aligned with the top eigenvector $q_1(H)$ in a few steps, (iv) $\|H\|_{S_n}$ reaches the threshold $\frac{2}{\eta}$ and the iterate jumps across the valley, (v) $\|H\|_{S_n}$ tends to decrease due to the non-quadraticity, and it repeats this process, while $\|H\|_{S_n}$ oscillating around $\frac{2}{\eta}$. We observe a similar behavior with oscillating $\frac{\text{tr}(HS_b)}{\text{tr}(S_n)}$ around $\frac{2}{\eta}$ for SGD. It requires further investigation into the exact underlying mechanisms (e.g., the progressive sharpening) (Arora et al., 2022; Li et al., 2022; Damian et al., 2022; Zhu et al., 2022) and we leave it as a future work.

**Remark** (Experiments in Section 4.1 and 4.2). We report the experimental results using vanilla SGD/GD without momentum and weight decay, constant learning rate, and no data augmentation. We train a simple 6-layer CNN (6CNN, $m = 0.51M$) on CIFAR-10-8k where DATASET-$n$ denotes a subset of DATASET with $|\mathcal{D}| = n$ and k $= 2^{10} = 1024$. See Appendix C.1-C.3 for the results from other datasets, learning rates and networks (ResNet-9 with $m = 2.3M$ (He et al., 2016) and WRN-28-2 with $m = 36M$ (Zagoruyko & Komodakis, 2016) where $m = \dim(\theta)$).

### 4.3 IMPLICIT INTERACTION REGULARIZATION (IIR)

In the previous sections, we have shown the SGD iterate is implicitly discouraged from staying in the unstable region. Now, we are ready to investigate this property from the regularization perspective. First, to understand the effect of batch size $b$ on the batch gradient distribution, we define the following concentration measure $\rho_b$:

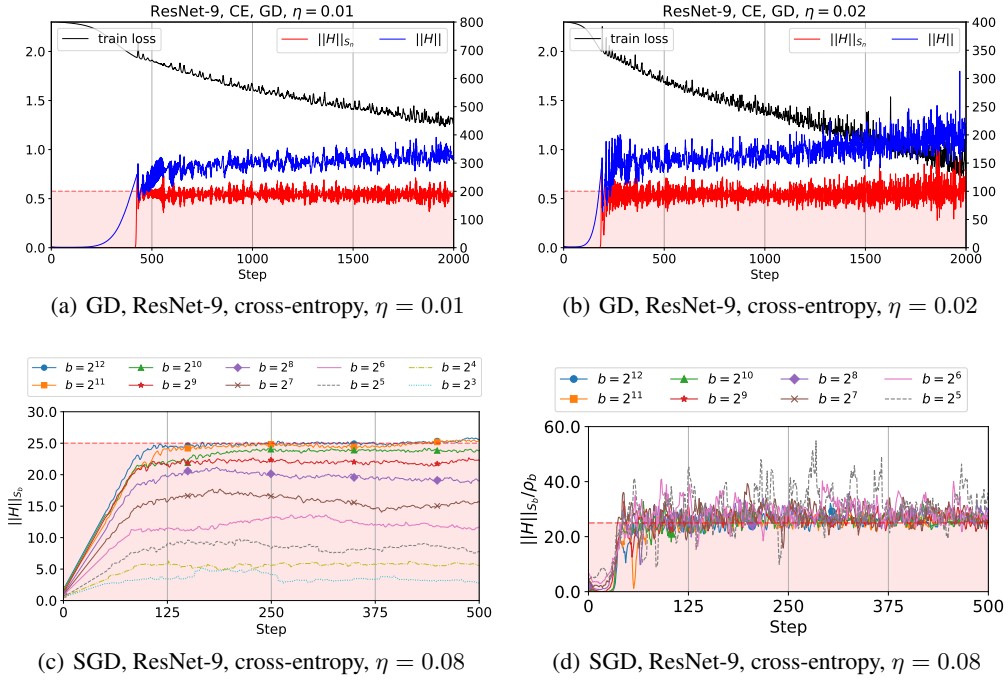

Figure 4: **[Clearer indication of the EoS]** (a)-(b): After a few steps of full-batch training, $\|H\|$ (blue) hovers **above** $\frac{2}{\eta}$ (Cohen et al., 2021), but $\|H\|_{S_n}$ (red, defined in (2)) oscillates **around** $\frac{2}{\eta}$ (red dashed horizontal line). The EoS is more evident in the latter (red). We also observe a sharp increase in $\|H\|_{S_n}$ right after $\|H\|$ exceeds $\frac{2}{\eta}$. Curves are plotted for every step. We train a model on CIFAR-10-8k ($n = 2^{13}$) using cross-entropy loss with $\eta = 0.01/0.02$, respectively. (c)-(d): We plot curves (c) $\|H\|_{S_b}$ and (d) $\frac{\|H\|_{S_b}}{\rho_b}$ when trained with various $b$. After a few steps, the curves in (c) reach the threshold $\frac{2\rho_b}{\eta}$ (see (d) together) which increases as $b$ becomes larger when $b \ll n = 2^{13}$, and saturates to $\frac{2\rho_b}{\eta} \approx \frac{2}{\eta}$ when $b$ is large. Curves are smoothed for visual clarity.

**Definition 2** (a concentration measure of the batch gradient). We define $\rho_b$ as the ratio of the squared norm of the total gradient $\|\nabla L\|^2$ to the expected squared norm of the batch gradients $\mathbb{E}_\xi[\|g_\xi\|^2]$, i.e.,

$$\rho_b \equiv \frac{\|\nabla L\|^2}{\mathbb{E}_\xi[\|g_\xi\|^2]} = \frac{\text{tr}(S_n)}{\text{tr}(S_b)}. \tag{15}$$

Here, we can write $\|\nabla L\|^2 = \|\mathbb{E}_\xi[g_\xi]\|^2$ and thus the ratio $\rho_b = \frac{\|\mathbb{E}_\xi[g_\xi]\|^2}{\mathbb{E}_\xi[\|g_\xi\|^2]} \le 1$ is similar to the square of the mean resultant length $\bar{R}_b^2 \equiv \|\mathbb{E}_\xi[\frac{g_\xi}{\|g_\xi\|}]\|^2 \le 1$ of the batch gradient $g_\xi$ (Mardia et al., 2000), especially when $\text{std}_\xi[\|g_\xi\|] \ll \mathbb{E}_\xi[\|g_\xi\|]$ (see Appendix C.5 for empirical evidences). Both $\rho_b$ and $\bar{R}_b^2$ are concentration measures and have lower values when the batch gradients $g_\xi$ are more scattered. Therefore, it is natural to expect that the ratio $\rho_b$ is small for a small batch size $b$, and we will revisit this in more detail in the following section (cf. (17)). We also note that $\rho_n = \bar{R}_n^2 = 1$. Yin et al. (2018) call $1/(n\rho_1)$ the gradient diversity.

Now, we can rewrite the instability condition $\frac{\text{tr}(HS_b)}{\text{tr}(S_n)} > \frac{2}{\eta}$ (multiplying both sides by $\rho_b$) as follows:

$$\|H\|_{S_b} > \frac{2\rho_b}{\eta}. \tag{16}$$

From Theorem 1 and 2, the instability condition (16) implies that IAS $\|H\|_{S_b}$ is implicitly regularized and bounded to be less than $\frac{2\rho_b}{\eta}$. We name this *Implicit Interaction Regularization (IIR)*. We argue that the upper constraint $\frac{2\rho_b}{\eta}$ in IIR is crucial in determining the SGD dynamics. With a low constraint,

SGD strongly bounds IAS $\|H\|_{S_b}$. We also note that IIR affects not only the magnitude $\|H\|$ but also the *directional* interaction. In other words, IIR discourages the batch gradients from aligning with the top eigensubspace of the Hessian that is spanned by a few largest eigenvectors of the Hessian (cf. Gur-Ari et al. (2018)).

Figures 4(a)-4(b) show that, for GD ($\rho_n = 1$), IAS $\|H\|_{S_n}$ (red) oscillates *around* $\frac{2}{\eta}$ and exhibits IIR. This result is consistent with Cohen et al. (2021) that $\|H\|$ hovers *above* $\frac{2}{\eta}$ for GD. This is because, as mentioned earlier, $\frac{2}{\eta} \approx \|H\|_{S_n} \leq \|H\|$ and the equality holds only when the gradient $\nabla L$ and the top eigenvector $q_1$ of $H$ are aligned, which is in general not the case. For this reason, IIR provides a tighter relation and more clearly identifies the EoS than Cohen et al. (2021). These results are also consistent with Theorem 2 that $\|H\|_{S_n}$ suddenly increases from 0 to $\frac{2}{\eta}$ in a few steps after $\|H\|$ exceeds $\frac{2}{\eta}$ (see Appendix C.3-C.4 for more). Moreover, IIR also applies to a general SGD training with $1 \leq b \leq n$. Figure 4(c)-4(d) show IIR for SGD with different batch sizes. The upper bound $\frac{2\rho_b}{\eta}$ of $\|H\|_{S_b}$ according to IIR is higher when using a larger batch size, but limited to less than $\frac{2}{\eta}$ ($\rho_b \leq 1$). We will further discuss this behavior with an investigation of $\rho_b$ in the following section.

## 5 LINEAR AND SATURATION SCALING RULE (LSSR)

In this section, we first introduce two previous scaling rules on how to tune the learning rate for varying batch sizes. Then we explain why they fail and propose a new scaling rule based on IIR.

The ratio $b/\eta$ of batch size $b$ to learning rate $\eta$ has long been believed as an important factor influencing the generalization performance, and the test accuracy has observed to be similar when trained with the same ratio $b/\eta = b'/\eta'$, i.e., $b' = kb$ and $\eta' = k\eta$ for $k > 0$. This is called the linear scaling rule (LSR) (Krizhevsky, 2014; Goyal et al., 2017; Jastrzębski et al., 2017; Smith & Le, 2018; Zhang et al., 2019). They argued that LSR holds because $\theta_{t+k} - \theta_t = -\frac{\eta}{b} \sum_{i=0}^{k-1} \sum_{x \in \mathcal{B}_\xi^{t+i}} \nabla_\theta \ell(x; \theta_{t+i}) \approx -\frac{\eta}{b} \sum_{i=0}^{k-1} \sum_{x \in \mathcal{B}_\xi^{t+i}} \nabla_\theta \ell(x; \theta_t) = -\frac{\eta'}{b'} \sum_{x \in \mathcal{B}_\xi^{t:t+k}} \nabla_\theta \ell(x; \theta_t)$ assuming $\nabla_\theta \ell(\theta_{t+i}) \approx \nabla_\theta \ell(\theta_t)$ for $0 \leq i < k$, where $\mathcal{B}_\xi^{t:t+k} \equiv \cup_{i=0}^{k-1} \mathcal{B}_\xi^{t+i}$ and $|\mathcal{B}_\xi^{t:t+k}| = kb = b'$. However, the assumption is not accurate since the gradient oscillates mostly with a negative cosine value $\cos(g_\xi(\theta_t), g_\xi(\theta_{t+1})) < 0$ between two consecutive gradients after entering the EoS (see Figure 24-25 in Appendix C.3). Moreover, LSR is known to fail when the batch size is large (Jastrzębski et al., 2017; Masters & Luschi, 2018; Zhang et al., 2019; Smith et al., 2020; 2021). On the other hand, Krizhevsky (2014); Hoffer et al. (2017) proposed the square root scaling rule (SRSR) with another ratio $\sqrt{b}/\eta$ to keep the covariance of the parameter update constant for $b \ll n$ based on $\text{Var}_\xi[\eta g_\xi] = \eta^2 C_b = \frac{\gamma_{n,b}\eta^2}{b} C_1 \approx \frac{\eta^2}{b} C_1$. However, Shallue et al. (2018) showed that both LSR and SRSR do not hold in general.

Based on the analysis of IIR with a new ratio $2\rho_b/\eta$ in the previous section, we explore why LSR fails in the large-batch regime and provide a more accurate rule to achieve similar generalization performance of the models trained with various choices of batch size and learning rate pairs $(b, \eta)$.

To this end, we investigate the concentration measure $\rho_b = \text{tr}(S_n)/\text{tr}(S_b)$. By combining two equations, $C_b = S_b - S_n$ (by definition) and $C_b = \frac{\gamma_{n,b}}{b}(S_1 - S_n)$ in (1), we can obtain $S_b = C_b + S_n = \frac{\gamma_{n,b}}{b} S_1 + (1 - \frac{\gamma_{n,b}}{b}) S_n$. Therefore, we have $\text{tr}(S_b) = \frac{\gamma_{n,b}}{b} \text{tr}(S_1) + (1 - \frac{\gamma_{n,b}}{b}) \text{tr}(S_n)$, which leads to the following equation:

$$\rho_b \equiv \frac{\text{tr}(S_n)}{\text{tr}(S_b)} = \frac{\text{tr}(S_n)}{\frac{\gamma_{n,b}}{b} \text{tr}(S_1) + (1 - \frac{\gamma_{n,b}}{b}) \text{tr}(S_n)} = \underbrace{\frac{1}{\frac{\gamma_{n,b}}{b} \frac{1}{\rho} + (1 - \frac{\gamma_{n,b}}{b})}}_{(*)} \approx \begin{cases} \frac{b}{\gamma_{n,b}} \rho \approx b\rho & \text{if } b \text{ is small} \\ 1 & \text{if } b \text{ is large} \end{cases}$$

$$(17)$$

from (15) where $\rho = \rho_1 = \text{tr}(S_n)/\text{tr}(S_1)$. Note that $\rho$ is (much) smaller than 1 because $\nabla_\theta \ell(x_i)$ has different direction for each $x_i$ and $\text{tr}(S_n) = \|\nabla L\|^2 = \|\frac{1}{n} \sum_i \nabla_\theta \ell(x_i)\|^2 \leq \frac{1}{n} \sum_i \|\nabla \ell_\theta(x_i)\|^2 = \text{tr}(S_1)$. In other words, $1/\rho$ is (much) larger than 1 (see Appendix C.5).

Figure 5 (left) demonstrates a new scaling rule with the ratio $\rho_b/\eta$, called the *Linear and Saturation Scaling Rule* (LSSR), with the two regimes that (i) $\rho_b$ is almost linear when $b \ll n$ (linear regime) and (ii) $\rho_b$ saturates when $b$ is large (saturation regime). It depends on which part of the denominator

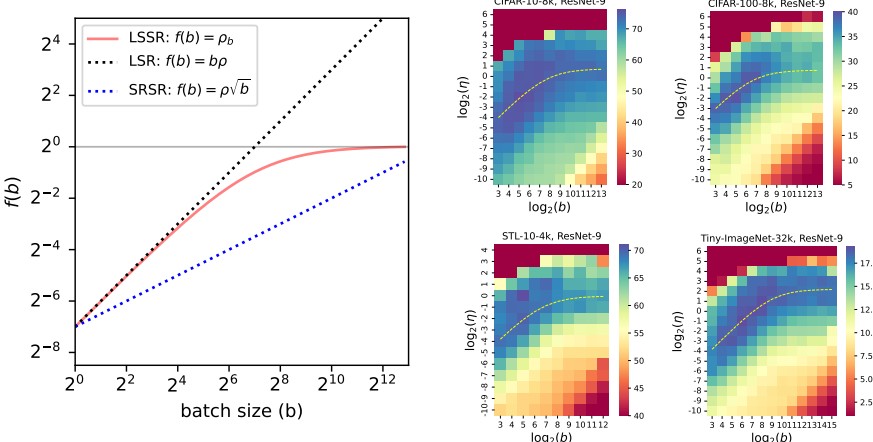

Figure 5: **[Linear and Saturation Scaling Rule (LSSR)]** Left: LSSR (red) in (17), LSR (black dotted line) (Goyal et al., 2017) and SRSR (blue dotted line) (Hoffer et al., 2017). For LSSR, we can observe both linear and saturation regions ($n = 8\text{k}, \rho = 2^{-7}$). Right: Heatmaps of test accuracy for models trained with a large number of pairs of $(b, \eta)$ on CIFAR-10-8k , CIFAR-100-8k , STL-10-4k, and Tiny-ImageNet-32k (from left to right, from top to bottom). It does not follow either LSR or SRSR, but tends to follow LSSR. We also plot $\rho_b/\eta = C$ (yellow dashed curve) for some $\rho$ and $C$ on each heatmap. Note that they are all log-log plots and thus a slope of 1 indicates linear relationship.

$(*)$ in (17) dominates the other. First, when $b \ll n$, then $\gamma_{n,b}/b$ is not very small and the first term $\frac{\gamma_{n,b}}{b}\frac{1}{\rho}$ dominates the second term $1 - \frac{\gamma_{n,b}}{b}$ since $\frac{1}{\rho} \gg 1$. Second, as $b$ becomes large, $\gamma_{n,b}/b \approx 0$ and the second term $(\approx 1)$ dominates the first term. Thus, $\rho_b$ saturates to 1 and is not linearly related to $b$, and LSR is no longer valid. The above arguments also hold for the batches sampled *with* replacement where the only modification is $\gamma_{n,b} = 1$, $\forall b$ in (17). Figure 5 (right) empirically supports LSSR with the test accuracies when trained with various combinations of pairs $(b, \eta)$. To be specific, the optimal learning rate is almost linear to $b$ when $b$ is small, but it saturates when $b$ is large. We also plot $\rho_b/\eta = C$ (the yellow dashed curve) for some $\rho$ and $C$ which shows a theoretical prediction of pairs $(\eta, b)$ that yield the optimal performance. Note that Figure 8 of Shallue et al. (2018, Section 4.7) shows similar "linear and saturation" behaviors supportive of LSSR on other datasets (see also Figure 7 of Zhang et al. (2019, Section 4.3)).

**Remark** (Experiments in Section 4.3 and 5). We train models using vanilla SGD/GD without momentum and weight decay, constant learning rate, and no data augmentation. For Figure 5, we use subsets of the datasets CIFAR-10 (Krizhevsky & Hinton, 2009), CIFAR-100 (Krizhevsky & Hinton, 2009), STL-10 (Coates et al., 2011), and Tiny-ImageNet (a subset of ImageNet (Deng et al., 2009) with $3 \times 64 \times 64$ images and 200 object classes). We use a large number of epochs (800) and batch normalization (Ioffe & Szegedy, 2015) to achieve a zero training error even with a large $b$ and a small $\eta$. In the lower right corner (red area) of each heatmap in Figure 5 (right), when $b$ is too large or $\eta$ is too small so that $\|\theta_{t+1} - \theta_t\| = \eta\|g_\xi\|$ is too small, it requires an exponentially large number of steps for the iterate to enter the EoS. Thus, in this case, the assumption in Goyal et al. (2017), $\nabla_\theta \ell(\theta_t) \approx \nabla_\theta \ell(\theta_{t+i})$ for $0 \le i < k$, approximately holds and the reasoning on LSR is valid. However, this only holds for a non-practical $(b, \eta)$ which shows a suboptimal performance. See Appendix C.4-C.5 for the results from other networks and hyperparameters.

## 6 CONCLUSION

From an analysis of unstable dynamics of SGD and the instability condition, we clearly mark the edge of stability with the interaction-aware sharpness $\|H\|_{S_b}$ and show the presence of the implicit regularization effect on the interaction between the gradient distribution and the loss landscape geometry (IIR). Moreover, introducing the concentration measure $\rho_b$ of the batch gradient, we link the second moment of the gradient distribution and the sharpness of the loss landscape, and propose a new scaling rule called Linear and Saturation Scaling Rule (LSSR). Due to the simplicity of the analysis, we hope that our insights will motivate the future work toward understanding various learning tasks.

## ACKNOWLEDGEMENTS

S. Lee is also the lead author and the corresponding author. This work was done while S. Lee was at Korea Institute for Advanced Study. S. Lee was supported by a KIAS Individual Grant (AP083601) via the Center for AI and Natural Sciences at Korea Institute for Advanced Study. C. Jang was supported in part by IITP Artificial Intelligence Graduate School Program for Hanyang University funded by MSIT (Grant No. 2020-0-01373) and NRF/MSIT (Grant No. 2021M3E5D2A01019545). This work was supported by the Center for Advanced Computation at Korea Institute for Advanced Study. This work was supported by the research fund of Hanyang University (HY-202300000000552).

## REPRODUCIBILITY STATEMENT

We refer the reader to the following pointers for reproducibility:

- Codes to reproduce some Figures: Supplementary Material.
- Proofs of the claims: Appendix B.
- Experimental settings : Remarks at the end of Sections 4.2 and 5, and Appendix C.
- Additional Figures for other hyperparameter settings: Appendix C.1-C.6.

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

# A  NOTATIONS

We summarize the notations for a quick reference.

| | |
|---|---|
| $t \in \mathbb{N}$ | time step |
| $\theta \in \Theta \subset \mathbb{R}^m$ (or indexed $\theta_t$); $\dim(\theta) = m$ | model parameter |
| $x \in \mathcal{X}$ (or indexed $x_i$) | training sample |
| $\mathcal{D} = \{x_i\}_{i=1}^n$; $|\mathcal{D}| = n$ | training data |
| $\ell(x; \theta)$ | loss function |
| $L(\theta) \equiv \dfrac{1}{n} \displaystyle\sum_{i=1}^n \ell(x_i; \theta) = \dfrac{1}{|\mathcal{D}|} \displaystyle\sum_{x \in \mathcal{D}} \ell(x; \theta)$; $L_t = L(\theta_t)$ | (total) training loss |
| $\eta > 0$ | learning rate |
| $\mathcal{B} \subset \mathcal{D}$ (or indexed $\mathcal{B}_\xi, \mathcal{B}_\xi^t$) | batch |
| $b = |\mathcal{B}|$; $1 \leq b \leq n$ | batch size |
| $g_\xi(\theta) \equiv \dfrac{1}{b} \displaystyle\sum_{x \in \mathcal{B}_\xi} \nabla \ell(x; \theta) = \dfrac{1}{|\mathcal{B}_\xi|} \displaystyle\sum_{x \in \mathcal{B}_\xi} \nabla \ell(x; \theta)$ | batch gradient |
| $\theta_{t+1} = \theta_t - \eta \nabla_\theta L(\theta_t)$ | GD |
| $\theta_{t+1} = \theta_t - \eta g_\xi(\theta_t)$ | SGD |
| $\delta_t = \theta_{t+1} - \theta_t$ | displacement/velocity vector |
| $C_b(\theta) \equiv \mathrm{Var}_\xi[g_\xi(\theta)] \in \mathbb{R}^{m \times m}$ | the covariance of the batch gradient |
| $S_b(\theta) \equiv \mathbb{E}_\xi\left[g_\xi(\theta) g_\xi(\theta)^\top\right] \in \mathbb{R}^{m \times m}$ | the second moment of the batch gradient |
| $\gamma_{n,b} = \begin{cases} \frac{n-b}{n-1} & \text{for sampling } \textit{without} \text{ replacement} \\ 1 & \text{for sampling } \textit{with} \text{ replacement} \end{cases}$ | sampling coefficient |
| $H(\theta) \equiv \nabla_\theta^2 L(\theta) = \mathbb{E}_{x \sim \mathcal{D}}[\nabla_\theta^2 \ell(x; \theta)] \in \mathbb{R}^{m \times m}$ | Hessian |
| $\|u\| = \left(\displaystyle\sum_i u_i^2\right)^{1/2}$ | the Euclidean $\ell^2$-norm of a vector $u$ |
| $\|A\| \equiv \displaystyle\sup_{u \neq 0} \dfrac{\|Au\|}{\|u\|}$ | the spectral (operator) norm of a matrix $A$ |
| $\lambda_i = \lambda_i(H) \in \mathbb{R}$ | the $i$-th largest eigenvalue of the Hessian |
| $q_i = q_i(H) \in \mathbb{R}^m$ | the corresponding $i$-th eigenvector of the Hessian |
| $\mathrm{tr}(A) = \displaystyle\sum_i A_{i,i}$ | the trace of a square matrix $A$ |
| $\|H\|_{S_b} \equiv \dfrac{\mathrm{tr}(H S_b)}{\mathrm{tr}(S_b)}$ | interaction-aware sharpness |
| $\rho_b \equiv \dfrac{\mathrm{tr}(S_n)}{\mathrm{tr}(S_b)}$ | concentration measure of the batch gradient |
| $\rho \equiv \dfrac{\mathrm{tr}(S_n)}{\mathrm{tr}(S_1)}$ | concentration measure of the per-example gradient |
| $\mathbb{U} = \{\theta \in \Theta : \|H\|_{S_b} > \dfrac{2\rho_b}{\eta}\}$ | unstable regime |
| $\mathbb{S} = \mathbb{U}^c = \{\theta \in \Theta : \|H\|_{S_b} \leq \dfrac{2\rho_b}{\eta}\}$ | stable regime |
| $\partial \mathbb{S} = \{\theta \in \Theta : \|H\|_{S_b} = \dfrac{2\rho_b}{\eta}\}$ | the edge of stability |
| $\beta_1, \beta_2 \in (0, 1]$ | hyperparameters for generalized momentum GD |

# B    PROOFS AND REMARKS

## B.1    PROOF OF (1)

We provide a proof of (1) to make the paper self-contained. Similar proofs are given in Li et al. (2017); Hoffer et al. (2017); Wu et al. (2020).

*Proof.* We start with

$$C_b = \mathbb{E}_\xi[g_\xi g_\xi^\top] - \nabla L \nabla L^\top \tag{18}$$

$$= \nabla\mathcal{L}\mathbb{E}_\xi[ww^\top]\nabla\mathcal{L}^\top - \nabla\mathcal{L}\left(\frac{1}{n}\mathbf{1}\frac{1}{n}\mathbf{1}^\top\right)\nabla\mathcal{L}^\top \tag{19}$$

$$= \nabla\mathcal{L}\left(\mathbb{E}_\xi[ww^\top] - \frac{1}{n^2}\mathbf{1}\mathbf{1}^\top\right)\nabla\mathcal{L}^\top \tag{20}$$

$$= \nabla\mathcal{L}\mathrm{Var}_\xi[w]\nabla\mathcal{L}^\top \tag{21}$$

$$= \frac{1}{b^2}\nabla\mathcal{L}\mathrm{Var}_\xi[v]\nabla\mathcal{L}^\top, \tag{22}$$

where $\nabla\mathcal{L} = [\nabla\ell_1, \cdots, \nabla\ell_n] \in \mathbb{R}^{m\times n}$, $\ell_i = \ell(x_i)$, the random vector $w = [w_1, \cdots, w_n]^\top$, each element of which represents $\frac{1}{b}\times$ "how many times the index $i$ is sampled in $\mathcal{B}_\xi$", and $v = bw$.

In case of sampling *with* replacement, we have $v = v^{(1)} + \cdots + v^{(b)}$ where $v^{(i)}$ represents sampling of a single sample. Thus, $\mathrm{Var}_\xi[v] = b\mathrm{Var}[v^{(1)}]$. We have $\mathbb{E}[v^{(1)}] = \frac{1}{n}\mathbf{1}$ and

$$\mathbb{E}[v^{(1)}v^{(1)T}]_{i,j} = \begin{cases} P[i \in \mathcal{B}^{(1)}] = \frac{1}{n} & \text{if } i = j \\ P[i \in \mathcal{B}^{(1)}, j \in \mathcal{B}^{(1)}] = 0 & \text{else} \end{cases}, \tag{23}$$

where $|\mathcal{B}^{(1)}| = 1$. Thus,

$$\mathrm{Var}_\xi[v] = b\mathrm{Var}[v^{(1)}] = b\left(\frac{1}{n}I - \frac{1}{n^2}\mathbf{1}\mathbf{1}^\top\right). \tag{24}$$

In case of sampling *without* replacement, we have $\mathbb{E}_\xi[v] = \frac{b}{n}\mathbf{1}$ and

$$\mathbb{E}_\xi[vv^\top]_{i,j} = \begin{cases} P[i \in \mathcal{B}_\xi] = \frac{C(n-1,b-1)}{C(n,b)} = \frac{b}{n} & \text{if } i = j \\ P[i \in \mathcal{B}_\xi, j \in \mathcal{B}_\xi] = \frac{C(n-2,b-2)}{C(n,b)} = \frac{b(b-1)}{n(n-1)} & \text{else} \end{cases}, \tag{25}$$

where $C(n_1, r_1)$ is the number of $r_1$-combinations from a set of $n_1$ elements. This leads to

$$\mathbb{E}_\xi[vv^\top] = \frac{b(b-1)}{n(n-1)}\mathbf{1}\mathbf{1}^\top + \left(\frac{b}{n} - \frac{b(b-1)}{n(n-1)}\right)I \tag{26}$$

and

$$\mathrm{Var}_\xi[v] = \mathbb{E}_\xi[vv^\top] - \frac{b^2}{n^2}\mathbf{1}\mathbf{1}^\top = \left(\frac{b(b-1)}{n(n-1)} - \frac{b^2}{n^2}\right)\mathbf{1}\mathbf{1}^\top + \frac{b(n-b)}{n(n-1)}I \tag{27}$$

$$= \frac{b(b-n)}{n^2(n-1)}\mathbf{1}\mathbf{1}^\top + \frac{b(n-b)}{n(n-1)}I \tag{28}$$

$$= \frac{b(n-b)}{n-1}\left(\frac{1}{n}I - \frac{1}{n^2}\mathbf{1}\mathbf{1}^\top\right) \tag{29}$$

Putting the two cases together, from (22), (24) and (29), we have

$$C_b = \frac{1}{b^2}b\gamma_{n,b}\nabla\mathcal{L}\left(\frac{1}{n}I - \frac{1}{n^2}\mathbf{1}\mathbf{1}^\top\right)\nabla\mathcal{L}^\top = \frac{\gamma_{n,b}}{b}\left(\frac{1}{n}\nabla\mathcal{L}\nabla\mathcal{L}^\top - \frac{1}{n^2}(\nabla\mathcal{L}\mathbf{1})(\nabla\mathcal{L}\mathbf{1})^\top\right) \tag{30}$$

$$= \frac{\gamma_{n,b}}{b}\left(\frac{1}{n}\sum_i \nabla\ell_i\nabla\ell_i^\top - \nabla L\nabla L^\top\right) \tag{31}$$

$$= \frac{\gamma_{n,b}}{b}(S_1 - S_n), \tag{32}$$

where

$$\gamma_{n,b} = \begin{cases} 1 & \text{for the sampling with replacement} \\ \frac{n-b}{n-1} & \text{for the sampling without replacement} \end{cases}. \tag{33}$$

$\square$

## B.2 PROOF OF THEOREM 1

First, we expect the variance of the loss at the next iteration $(t+1)$ to be small for a sufficiently large $b$.

**Lemma 3.** *If the batch gradient $g_\xi$ is normally distributed and the loss $L$ is quadratic, then*

$$\text{Var}_\xi[L_{t+1}|\theta_t] = \frac{\eta^2 \gamma_{n,b}}{b} \left( v^\top C_1 v + \frac{\eta^2 \gamma_{n,b}}{2b} \text{tr}(HC_1HC_1) \right), \tag{34}$$

*where $v = (I - \eta H)\nabla L$.*

*Proof.* Since $L$ is quadratic, we have the following:

$$L_{t+1} = L_t - \eta g^\top g_\xi + \frac{\eta^2}{2} g_\xi^\top H g_\xi \qquad\qquad (g = \nabla L) \tag{35}$$

$$= \text{constant} - \eta g^\top(g + \varepsilon) + \frac{\eta^2}{2}(g + \varepsilon)^\top H(g + \varepsilon) \qquad (\varepsilon \sim \mathcal{N}(0, \Sigma = C_b)) \tag{36}$$

$$= \text{constant} - \eta g^\top \varepsilon + \eta^2 g^\top H \varepsilon + \frac{\eta^2}{2} \varepsilon^\top H \varepsilon \tag{37}$$

$$= \text{constant} - \eta((I - \eta H)g)^\top \varepsilon + \frac{\eta^2}{2} \varepsilon^\top H \varepsilon, \tag{38}$$

Then, the variance of $L_{t+1}$ is $\text{Var}_\xi[L_{t+1}|\theta_t] = \mathbb{E}_\xi[Q_\xi^2] - \mathbb{E}_\xi[Q_\xi]^2$ where $Q_\xi = -\eta v^\top \varepsilon + \frac{\eta^2}{2} \varepsilon^\top H \varepsilon$.

First, we can obtain the square of the expected value of $Q_\xi$ as follows:

$$\mathbb{E}_\xi[Q_\xi] = -\eta v^\top \mathbb{E}_\xi[\varepsilon] + \frac{\eta^2}{2} \sum_{i,j} H_{i,j} \mathbb{E}_\xi[\varepsilon_i \varepsilon_j] = \frac{\eta^2}{2} \sum_{i,j} H_{i,j} \Sigma_{i,j} = \frac{\eta^2}{2} \text{tr}(H\Sigma), \tag{39}$$

$$\mathbb{E}_\xi[Q_\xi]^2 = \frac{\eta^4}{4} \text{tr}(H\Sigma)^2, \tag{40}$$

where the last equation holds since $\Sigma$ is symmetric and $\sum_{i,j} H_{i,j} \Sigma_{i,j} = \sum_{i,j} H_{i,j} \Sigma_{j,i} = \sum_i [H\Sigma]_{i,i} = \text{tr}(H\Sigma)$. Second, we have the expected value of the square of $Q_\xi$ as follows:

$$Q_\xi^2 = \eta^2 \sum_{i,i'} v_i v_{i'} \varepsilon_i \varepsilon_{i'} - \eta^3 \sum_{i,j,k} v_i H_{j,k} \varepsilon_i \varepsilon_j \varepsilon_k + \frac{\eta^4}{4} \sum_{j,k,j',k'} H_{j,k} H_{j',k'} \varepsilon_j \varepsilon_k \varepsilon_{j'} \varepsilon_{k'} \tag{41}$$

and, since $\mathbb{E}_\xi[\varepsilon_i \varepsilon_j \varepsilon_k] = 0$ and $\mathbb{E}[\varepsilon_j \varepsilon_k \varepsilon_{j'} \varepsilon_{k'}] = \mathbb{E}[\varepsilon_j \varepsilon_k]\mathbb{E}[\varepsilon_{j'} \varepsilon_{k'}] + \mathbb{E}[\varepsilon_j \varepsilon_{j'}]\mathbb{E}[\varepsilon_k \varepsilon_{k'}] + \mathbb{E}[\varepsilon_j \varepsilon_{k'}]\mathbb{E}[\varepsilon_{j'} \varepsilon_k]$ by Isserlis' theorem (Isserlis, 1918) for zero-mean normal random vector $\varepsilon$,

$$\mathbb{E}_\xi[Q_\xi^2] = \eta^2 \sum_{i,i'} v_i v_{i'} \Sigma_{i,i'} + \frac{\eta^4}{4} \sum_{j,k,j',k'} H_{j,k} H_{j',k'} (\Sigma_{j,k} \Sigma_{j',k'} + \Sigma_{j,j'} \Sigma_{k,k'} + \Sigma_{j,k'} \Sigma_{j',k}) \tag{42}$$

$$= \eta^2 v^\top \Sigma v + \frac{\eta^4}{4} (\text{tr}(H\Sigma)^2 + 2\,\text{tr}(H\Sigma H\Sigma)). \tag{43}$$

Finally, we have the variance

$$\text{Var}_\xi[L_{t+1}|\theta_t] = \mathbb{E}_\xi[Q_\xi^2] - \mathbb{E}_\xi[Q_\xi]^2 \tag{44}$$

$$= \eta^2 \left( v^\top \Sigma v + \frac{\eta^2}{2} \text{tr}(H\Sigma H\Sigma) \right) \tag{45}$$

$$= \frac{\eta^2 \gamma_{n,b}}{b} \left( v^\top C_1 v + \frac{\eta^2 \gamma_{n,b}}{2b} \text{tr}(HC_1HC_1) \right). \tag{46}$$

$\square$

From Lemma 3, we can easily derive the following theorem from Chebyshev's inequality:

**Theorem 4.** *For SGD on a quadratic loss $L$, the expected loss at the next step $(t+1)$ is*

$$\mathbb{E}_\xi[L_{t+1}] = L_t - \frac{\eta^2}{2}\operatorname{tr}(S_n)\left[\frac{\operatorname{tr}(HS_b)}{\operatorname{tr}(S_n)} - \frac{2}{\eta}\right]. \tag{47}$$

*Further, if the batch gradient $g_\xi$ is normally distributed, then given $\delta > 0$ and $\alpha \geq \sqrt{\beta/\delta}$, we have, with probability of at least $1 - \delta$,*

$$|L_{t+1} - \mathbb{E}_\xi[L_{t+1}]| \leq \alpha, \tag{48}$$

*where $\beta = \frac{\eta^2 \gamma_{n,b}}{b}\left(v^\top C_1 v + \frac{\eta^2 \gamma_{n,b}}{2b}\operatorname{tr}(HC_1 HC_1)\right)$ and $v = (I - \eta H)\nabla L$.*

However, we can replace this inequality with a better exponential inequality. To do so, we need a generalized Lemma 6 of the following Lemma 5 (Laurent & Massart, 2000):

**Lemma 5** (Laurent & Massart (2000)). *For i.i.d. Gaussian variables $(Y_1, \cdots, Y_D)$ with mean 0 and variance 1, the following inequality holds for any positive $x$:*

$$\mathbb{P}(Z - \mathbb{E}[Z] \geq 2\|a\|_2\sqrt{x} + 2\|a\|_\infty x) \leq \exp(-x), \tag{49}$$

$$\mathbb{P}(Z - \mathbb{E}[Z] \leq -2\|a\|_2\sqrt{x}) \leq \exp(-x), \tag{50}$$

*where $Z = \sum_i a_i Y_i^2$.*

**Lemma 6.** *For i.i.d. Gaussian variables $(Y_1, \cdots, Y_D)$ with mean 0 and variance 1, the following inequality holds for any positive $x$:*

$$\mathbb{P}\left(Z - \mathbb{E}[Z] \geq 2\sqrt{\|a\|_2^2 + \frac{\|c\|_2^2}{2}}\sqrt{x} + 2\|a\|_\infty x\right) \leq \exp(-x), \tag{51}$$

$$\mathbb{P}\left(Z - \mathbb{E}[Z] \leq -2\sqrt{\|a\|_2^2 + \frac{\|c\|_2^2}{2}}\sqrt{x}\right) \leq \exp(-x), \tag{52}$$

*where $Z = \sum_i a_i Y_i^2 + c_i Y_i$.*

*Proof.* For a standard normal random variable $Y_i \sim \mathcal{N}(0,1)$, let $\psi$ be the Cramér transform of $Y_i^2 + c_i Y_i - 1$:

$$\psi(\lambda) = \log \mathbb{E}[\exp(\lambda(Y_i^2 + c_i Y_i - 1))] \tag{53}$$

$$= \log \int_\mathbb{R} p(y; \mathcal{N}(0,1)) \exp(\lambda(y^2 + c_i y - 1))dy \tag{54}$$

$$= \log \int_\mathbb{R} \frac{1}{\sqrt{2\pi}}\exp\left(-\frac{y^2}{2}\right)\exp(\lambda(y^2 + c_i y - 1))dy \tag{55}$$

$$= \log \int_\mathbb{R} \frac{1}{\sqrt{2\pi}}\exp\left(-\frac{y^2}{2} + \lambda(y^2 + c_i y - 1)\right)dy \tag{56}$$

$$= \log \int_\mathbb{R} \frac{1}{\sqrt{2\pi}}\exp\left(-\frac{1}{2}(1-2\lambda)y^2 + c_i \lambda y - \lambda\right)dy \tag{57}$$

$$= \log \int_\mathbb{R} \frac{1}{\sqrt{2\pi}}\exp(-\lambda)\exp\left(-\frac{1}{2}(1-2\lambda)y^2\right)\exp(-c_i \lambda y)\,dy \tag{58}$$

$$= \log \int_\mathbb{R} \exp(-\lambda)\sigma\frac{1}{\sqrt{2\pi\sigma^2}}\exp\left(-\frac{1}{2\sigma^2}y^2\right)\exp(-c_i \lambda y)\,dy \qquad (\sigma^2 = (1-2\lambda)^{-1}) \tag{59}$$

$$= -\lambda + \log(\sigma) + \mathbb{E}_{y \sim \mathcal{N}(0,\sigma^2)}[\exp(-c_i \lambda y)] \tag{60}$$

$$= -\lambda - \frac{1}{2}\log(1-2\lambda) + \frac{(c_i \lambda \sigma)^2}{2} \tag{61}$$

$$\leq \frac{\lambda^2(1 + \frac{c_i^2}{2})}{1 - 2\lambda}, \tag{62}$$

where the last inequality holds for $0 < \lambda < \frac{1}{2}$ since $-\lambda - \frac{1}{2}\log(1 - 2\lambda) = -\lambda + \frac{1}{2}\sum_{k\geq 1}\frac{(2\lambda)^k}{k} = \frac{1}{2}\sum_{k\geq 2}\frac{(2\lambda)^k}{k} = 2\lambda^2\sum_{k\geq 0}\frac{(2\lambda)^k}{k+2} \leq 2\lambda^2\sum_{k\geq 0}\frac{(2\lambda)^k}{2} = \lambda^2\sum_{k\geq 0}(2\lambda)^k = \frac{\lambda^2}{1-2\lambda}$.

Therefore,

$$\log \mathbb{E}[\exp(\lambda Z)] = \sum_i \log \mathbb{E}\left[\exp\left(a_i\lambda\left(Y_i^2 + \frac{c_i}{a_i}Y_i - 1\right)\right)\right] \leq \sum_i \frac{(1 + \frac{1}{2}(\frac{c_i}{a_i})^2)a_i^2\lambda^2}{1 - 2a_i\lambda} \quad (63)$$

$$\leq \frac{(\|a\|^2 + \frac{\|c\|^2}{2})\lambda^2}{1 - 2\|a\|_\infty\lambda}. \quad (64)$$

We can obtain (69) from the following (Birgé & Massart, 1998): if $\log\mathbb{E}[\exp(\lambda Z)] \leq \frac{b_1\lambda^2}{1-b_2\lambda}$, then, for any positive $x > 0$, $\mathbb{P}(Z \geq \sqrt{b_1 x} + b_2 x) \leq \exp(-x)$. Also, for $-\frac{1}{2} < \lambda < 0$, $\psi(\lambda) \leq \lambda^2(1 + \frac{c_i^2}{2})$ and thus $\log\mathbb{E}[\exp(\lambda Z)] = (\|a\|^2 + \frac{\|c\|^2}{2})\lambda^2$ which leads to (70). $\square$

**Theorem 1.** *For SGD on a quadratic L, the expected loss increases, i.e., $\mathbb{E}_\xi[L_{t+1}] - L_t > 0$, if and only if $\theta_t$ satisfies the instability condition $\frac{\text{tr}(HS_b)}{\text{tr}(S_n)} > \frac{2}{\eta}$. Furthermore, if the batch gradient $g_\xi$ is normally distributed, then the following inequalities hold for any positive $x > 0$:*

$$\mathbb{P}\left(L_{t+1} - \mathbb{E}_\xi[L_{t+1}] \geq \sqrt{2\beta x} + \frac{\eta^2\gamma_{n,b}}{b}\|H\|\|C_1\|x \,\Big|\, \theta_t\right) \leq \exp(-x), \quad (8)$$

$$\mathbb{P}\left(L_{t+1} - \mathbb{E}_\xi[L_{t+1}] \leq -\sqrt{2\beta x} \,\Big|\, \theta_t\right) \leq \exp(-x), \quad (9)$$

*where $\beta = \frac{\eta^2\gamma_{n,b}}{b}\left(v^\top C_1 v + \frac{\eta^2\gamma_{n,b}}{2b}\text{tr}(HC_1HC_1)\right)$ and $v = (I - \eta H)\nabla L$.*

*Proof.* The first part of the statement is trivial from (7) with $\epsilon = 0$ for a quadratic L. Now, we focus on the concentration of $L_{t+1}$ from its expected value: $L_{t+1} - \mathbb{E}[L_{t+1}] = Z - \mathbb{E}[Z]$ for $Z = Q_\xi = -\eta v^\top \varepsilon + \frac{\eta^2}{2}\varepsilon^\top H\varepsilon$ where $v = (I - \eta H)\nabla L$ and $\varepsilon \sim \mathcal{N}(0, C_b)$ from (38). For $C_b = \frac{\gamma_{n,b}}{b}BB^\top$, we can represent $\varepsilon = \sqrt{\frac{\gamma_{n,b}}{b}}B\tilde{\varepsilon}$ with $\tilde{\varepsilon} \sim \mathcal{N}(0, I)$ and

$$Z = -\eta v^\top \varepsilon + \frac{\eta^2}{2}\varepsilon^\top H\varepsilon \quad (65)$$

$$= -\eta\sqrt{\frac{\gamma_{n,b}}{b}}(B^\top v)^\top \tilde{\varepsilon} + \frac{\eta^2\gamma_{n,b}}{2b}\tilde{\varepsilon}^\top B^\top HB\tilde{\varepsilon} \quad (66)$$

$$= c^\top Y + Y^\top \text{diag}(a)Y \quad (67)$$

$$= \sum_i a_i Y_i^2 + c_i Y_i, \quad (68)$$

where $A = \frac{\eta^2\gamma_{n,b}}{2b}B^\top HB = Q\,\text{diag}(a)Q^\top$, $c = -\eta\sqrt{\frac{\gamma_{n,b}}{b}}Q^\top B^\top v$ and $Y = [Y_1, \cdots, Y_m] = Q^\top\tilde{\varepsilon}$. Therefore, from the rotation invariance, $\{Y_i\}$ are i.i.d. Gaussian variables with mean 0 and variance 1 like $\tilde{\varepsilon}$. From Lemma 6, we have the following inequalities:

$$\mathbb{P}\left(L_{t+1} - \mathbb{E}[L_{t+1}] \geq 2\sqrt{\|A\|_F^2 + \frac{\|c\|_2^2}{2}}\sqrt{x} + 2\|A\|x\right) \leq \exp(-x), \quad (69)$$

$$\mathbb{P}\left(L_{t+1} - \mathbb{E}[L_{t+1}] \leq 2\sqrt{\|A\|_F^2 + \frac{\|c\|_2^2}{2}}\sqrt{x}\right) \leq \exp(-x). \quad (70)$$

Further, we have

$$\|A\|_F^2 = \text{tr}(A^\top A) = (\frac{\eta^2\gamma_{n,b}}{2b})^2\text{tr}(B^\top HBB^\top HB) = (\frac{\eta^2\gamma_{n,b}}{2b})^2\text{tr}(HBB^\top HBB^\top) \quad (71)$$

$$= (\frac{\eta^2\gamma_{n,b}}{2b})^2\text{tr}(HC_1HC_1) \quad (72)$$

and

$$\|c\|^2 = \frac{\eta^2 \gamma_{n,b}}{b} \|Q^\top B^\top v\|^2 = \frac{\eta^2 \gamma_{n,b}}{b} \|B^\top v\|^2 = \frac{\eta^2 \gamma_{n,b}}{b} v^\top C_1 v. \tag{73}$$

These lead to $\|A\|_F^2 + \frac{\|c\|_2^2}{2} = (\frac{\eta^2 \gamma_{n,b}}{2b})^2 \operatorname{tr}(HC_1HC_1) + \frac{\eta^2 \gamma_{n,b}}{2b} v^\top C_1 v = \frac{\beta}{2}$ and then (9) follows from (70). Moreover, we have $\|A\| \le \frac{\eta^2 \gamma_{n,b}}{2b} \|H\| \|C_1\|$ which leads to (8) from (69).

$\square$

### B.3 PROOF OF THEOREM 2

We consider the following linear homogeneous second-order equation for the sequence $\{y_t\}$:

$$y_{t+2} + p_1 y_{t+1} + p_2 y_t = 0 \tag{74}$$

and its the characteristic equation

$$\phi(x) = x^2 + p_1 x + p_2 = 0. \tag{75}$$

**Lemma 7** (Theorem 2.35 in Elaydi (2005)). *All solutions of (74) (i) oscillate (about zero) if and only if (75) has no positive real roots, and (ii) converges to 0 if and only if the solutions of (75) $x_1$ and $x_2$ satisfy $|x_1|, |x_2| < 1$.*

**Lemma 8** (Theorem 2.37 in Elaydi (2005)). *The conditions*

$$1 + p_1 + p_2 > 0 \tag{C1}$$
$$1 - p_1 + p_2 > 0 \tag{C1}$$
$$1 - p_2 > 0 \tag{C1}$$

*are necessary and sufficient for the equilibrium point (solution) of (74) to be asymptotically stable (i.g., all solutions converges to 0).*

**Theorem 2.** *For generalized momentum GD with $(\beta_1, \beta_2)$ on a quadratic $L$, if $0 < \lambda_i < \frac{2}{\eta} \gamma(\beta_1, \beta_2) < \lambda_1$ for all $i \ne 1$ where $\gamma(\beta_1, \beta_2) = \frac{1+\beta_1}{1+2\beta_2}$, then $q_1^\top \delta_t$ oscillates and diverges with the exponential growth of $|q_1^\top \delta_t| = \Theta(e^{ct})$ for some $c > 0$. Moreover, $|\cos(q_1, \delta_t)|$ and $\|H\|_{S_n}$ increase to 1 and $\lambda_1$, as $t \to \infty$, respectively, with $1 - |\cos(q_1, \delta_t)|, \lambda_1 - \|H\|_{S_n} = O(e^{-2ct})$.*

We provide a proof of Theorem 2 which is modified from the proofs in Appendix A in Cohen et al. (2021).

*Proof.* Put a quadratic training loss $L(\theta) = \frac{1}{2}\theta^\top H \theta + b^\top \theta + c$. The update rules for generalized momentum GD with $(\beta_1, \beta_2)$ on this quadratic function $L(\theta)$ are:

$$\delta_t = (\beta_1 I - \beta_2 \eta H)\delta_{t-1} - \eta b - \eta H \theta_t \tag{76}$$
$$\theta_{t+1} = \theta_t + \delta_t \tag{77}$$

and thus, for a pair of eigenvalue/eigenvector $(q, \lambda)$ the update rules for the quantities $\tilde{\theta}_t = q^\top \theta_t + \frac{1}{\lambda} q^\top b$ and $\tilde{\delta}_t = q^\top \delta_t$ are:

$$\tilde{\delta}_t = (\beta_1 - \beta_2 a)\tilde{\delta}_{t-1} - a\tilde{\theta}_t \tag{78}$$
$$\tilde{\theta}_{t+1} = \tilde{\theta}_t + \tilde{\delta}_t, \tag{79}$$

where $a = \eta \lambda > 0$ and this leads to:

$$\tilde{\theta}_{t+1} = \tilde{\theta}_t + \tilde{\delta}_t \tag{80}$$
$$= \tilde{\theta}_t + (\beta_1 - \beta_2 a)\tilde{\delta}_{t-1} - a\tilde{\theta}_t \tag{81}$$
$$= \tilde{\theta}_t + (\beta_1 - \beta_2 a)(\tilde{\theta}_t - \tilde{\theta}_{t-1}) - a\tilde{\theta}_t \tag{82}$$
$$= (1 + \beta_1 - (1 + \beta_2)a)\tilde{\theta}_t - (\beta_1 - \beta_2 a)\tilde{\theta}_{t-1}, \tag{83}$$
$$0 = \tilde{\theta}_{t+1} - (1 + \beta_1 - (1 + \beta_2)a)\tilde{\theta}_t + (\beta_1 - \beta_2 a)\tilde{\theta}_{t-1}, \tag{84}$$
$$0 = \tilde{\theta}_{t+1} - \tilde{\theta}_t - (1 + \beta_1 - (1 + \beta_2)a)(\tilde{\theta}_t - \tilde{\theta}_{t-1}) + (\beta_1 - \beta_2 a)(\tilde{\theta}_{t-1} - \tilde{\theta}_{t-2}), \tag{85}$$
$$0 = \tilde{\delta}_t \underbrace{- (1 + \beta_1 - (1 + \beta_2)a)}_{p_1} \tilde{\delta}_{t-1} + \underbrace{(\beta_1 - \beta_2 a)}_{p_2} \tilde{\delta}_{t-2}. \tag{86}$$

Note that $\tilde{\theta}_t$ and $\tilde{\delta}_t$ have the same characteristic equation. Thus, by Lemma 8, $\tilde{\delta}_t$ is asymptotically stable when the following three conditions (C1-3) hold:

$$1 + p_1 + p_2 = 1 - (1 + \beta_1 - (1 + \beta_2)a) + \beta_1 - \beta_2 a > 0, \qquad \text{(C1)} \qquad (87)$$
$$1 - p_1 + p_2 = 1 + (1 + \beta_1 - (1 + \beta_2)a) + \beta_1 - \beta_2 a > 0, \qquad \text{(C2)} \qquad (88)$$
$$1 - p_2 = 1 - \beta_1 + \beta_2 a > 0. \qquad \text{(C3)} \qquad (89)$$

C3 and C1 always hold because $1 - \beta_1 > 0$ and

$$1 - (1 + \beta_1 - (1 + \beta_2)a) + \beta_1 - \beta_2 a = a > 0. \qquad (90)$$

Lastly, since

$$1 + (1 + \beta_1 - (1 + \beta_2)a) + \beta_1 - \beta_2 a = 2 + 2\beta_1 - (1 + 2\beta_2)a > 0, \qquad (91)$$

The asymptotic convergence condition (C2) is equivalent to

$$\lambda < \frac{2 + 2\beta_1}{(1 + 2\beta_2)\eta} = \frac{2}{\eta}\frac{1 + \beta_1}{1 + 2\beta_2} = \frac{2}{\eta}\gamma(\beta_1, \beta_2). \qquad (92)$$

Therefore, along the direction of $q_1$, the sequence $\{q_1^\top \delta_t\}$ diverges, and along the direction of $q_i$ $(i > 1)$, the sequence $\{q_i^\top \delta_t\}$ converges to 0. The discriminant $D(a_0)$ for the characteristic equation $\phi(x; a_0)$ is positive as shown below where $a_0 = 2\gamma(\beta_1, \beta_2)$, i.e., $\phi(x; a_0)$ has two distinct real solutions.

$$D(a) = p_1^2 - 4p_2 \qquad (93)$$
$$= (1 + \beta_1 - (1 + \beta_2)a)^2 - 4(\beta_1 - \beta_2 a), \qquad (94)$$
$$D(a_0) = \left(\frac{1 + \beta_1}{1 + 2\beta_2}\right)^2 - 4\frac{\beta_1 - 2\beta_2}{1 + 2\beta_2} \qquad (a_0 = 2\gamma(\beta_1, \beta_2)) \qquad (95)$$
$$= \frac{(1 + \beta_1)^2 - 4(\beta_1 - 2\beta_2)(1 + 2\beta_2)}{(1 + 2\beta_2)^2} \qquad (96)$$
$$= \frac{(1 - \beta_1)^2 - 8((\beta_1 - 1)\beta_2 - 2\beta_2^2)}{(1 + 2\beta_2)^2} > 0. \qquad (97)$$

(94) implies that $D(a)$ is convex quadratic with respect to $a$. Thus, to show that $D(a) > 0$ ($\phi(x; a)$ has two distinct real solutions) for all $a > a_0$, it is sufficient to show that $D'(a) \geq 0$ for $a > a_0$. The following inequality holds

$$D'(a) = 2(1 + \beta_2)^2 a - 2(1 + \beta_1 - \beta_2 + \beta_1\beta_2) \geq 0 \qquad (98)$$

if $a \geq \frac{1 + \beta_1 - \beta_2 + \beta_1\beta_2}{(1 + \beta_2)^2}$. And, $D'(a) \geq 0$ for $a > a_0$ since $a > a_0 > \frac{1 + \beta_1 - \beta_2 + \beta_1\beta_2}{(1 + \beta_2)^2}$ from

$$a_0 - \frac{1 + \beta_1 - \beta_2 + \beta_1\beta_2}{(1 + \beta_2)^2} = \frac{2(1 + \beta_1)}{1 + 2\beta_2} - \frac{1 + \beta_1 - \beta_2 + \beta_1\beta_2}{(1 + \beta_2)^2} \qquad (99)$$
$$= \frac{2(1 + \beta_1)(1 + \beta_2)^2 - (1 + 2\beta_2)(1 + \beta_1 - \beta_2 + \beta_1\beta_2)}{(1 + 2\beta_2)(1 + \beta_2)^2} \qquad (100)$$
$$= \frac{1 + \beta_1 + 3\beta_2 + \beta_1\beta_2 + 4\beta_2^2}{(1 + 2\beta_2)(1 + \beta_2)^2} > 0. \qquad (101)$$

Now, for $a > a_0$, we want to show that $x_1$ for the dominant solution $(|x_1| > |x_2|)$ is negative so that the general solution $\tilde{\delta}_t = c_1 x_1^t + c_2 x_2^t = x_1^t \left[c_1 + c_2 \left(\frac{x_2}{x_1}\right)^t\right]$ oscillates where $c_i, x_i \in \mathbb{R}$ and $i \in \{1, 2\}$. The sum of the two solutions of $\phi(x; a)$ is $x_1 + x_2 = -p_1$ and this is negative for $a > a_0$

from the following:

$$p_1 = -(1+\beta_1) + (1+\beta_2)a \tag{102}$$

$$> -(1+\beta_1) + (1+\beta_2)a_0 \tag{103}$$

$$= -(1+\beta_1 - (1+\beta_2)a_0) \tag{104}$$

$$= -\left(1+\beta_1 - (1+\beta_2)\frac{2(1+\beta_1)}{1+2\beta_2}\right) \tag{105}$$

$$= -\frac{(1+\beta_1)(1+2\beta_2) - (1+\beta_2)2(1+\beta_1)}{1+2\beta_2} \tag{106}$$

$$= \frac{1+\beta_1}{1+2\beta_2} > 0. \tag{107}$$

Thus, we have that $x_1$ for the dominant solution is negative which leads to an oscillatory behavior of $q_1^\top \delta_t$. And it has the exponential growth of $|q_1^\top \delta_t| = \Theta(|x_1|^t) = \Theta(e^{ct})$ for $c = \ln|x_1| > 0$ since

$$|q_1^\top \delta_t| = |x_1|^t \left| \left[ c_1 + c_2 \left( \frac{x_2}{x_1} \right)^t \right] \right|, \tag{108}$$

$$\frac{1}{2}|c_1||x_1|^t \le |q_1^\top \delta_t| \le 2|c_1||x_1|^t \text{ for } t \ge t_0 \text{ for some } t_0 \tag{109}$$

($\lambda_1$ violates (92) with $\lambda_1 > \frac{2}{\eta}\gamma(\beta_1, \beta_2)$ which implies $|x_1| > 1$).

This makes $\delta_t$ to be asymptotically aligned with the top eigenvector $q_1$ of $H$, i.e., $\lim_{t\to\infty} |\cos(q_1, \delta_t)| = \lim_{t\to\infty} \frac{|q_1^\top \delta_t|}{\sqrt{\sum_{i=1}^m (q_i^\top \delta_t)^2}} = \lim_{t\to\infty} \frac{|\tilde{\delta}_t^{(1)}|}{\sqrt{\sum_{i=1}^m \tilde{\delta}_t^{(i)2}}} = 1$ where $\tilde{\delta}_t^{(i)} = q_i^\top \delta_t$.

Moreover, we can obtain the exponential convergence $(1 - |\cos(q_1, \delta_t)| = O(e^{-2ct}))$ as follows:

$$|\cos(q_1, \delta_t)| = \frac{|\tilde{\delta}_t^{(1)}|}{\sqrt{\tilde{\delta}_t^{(1)2} + s_t^2}} \qquad \left(s_t^2 = \sum_{i>1}^m \tilde{\delta}_t^{(i)2} \to 0\right), \tag{110}$$

$$1 - |\cos(q_1, \delta_t)| = \frac{\sqrt{\tilde{\delta}_t^{(1)2} + s_t^2} - |\tilde{\delta}_t^{(1)}|}{\sqrt{\tilde{\delta}_t^{(1)2} + s_t^2}} \tag{111}$$

$$= \frac{\tilde{\delta}_t^{(1)2} + s_t^2 - |\tilde{\delta}_t^{(1)}|\sqrt{\tilde{\delta}_t^{(1)2} + s_t^2}}{\tilde{\delta}_t^{(1)2} + s_t^2} \tag{112}$$

$$= s_t^2 \frac{\frac{\tilde{\delta}_t^{(1)2}}{s_t^2} - |\frac{\tilde{\delta}_t^{(1)}}{s_t}|\sqrt{\frac{\tilde{\delta}_t^{(1)2}}{s_t^2} + 1} + 1}{\tilde{\delta}_t^{(1)2} + s_t^2}, \tag{113}$$

$$0 \le 1 - |\cos(q_1, \delta_t)| \le \frac{1}{\tilde{\delta}_t^{(1)2}} \text{ for } t \ge t_1 \text{ for some } t_1. \tag{114}$$

The last inequality holds because $a_t = |\tilde{\delta}_t^{(1)}/s_t|$ diverges to $\infty$ and $\lim_{x\to\infty} x^2 - x\sqrt{x^2+1}+1 = \frac{1}{2}$, Moreover, we have

$$\nabla_\theta L(\theta_t) = H\theta_t + b \tag{115}$$

$$= \sum_i \lambda_i q_i q_i^\top \theta_t + \sum_i q_i q_i^\top b = \sum_i \lambda_i q_i (q_i^\top \theta_t + \frac{q_i^\top}{\lambda_i} b) = \sum_i \lambda_i \tilde{\theta}_t^{(i)} q_i, \tag{116}$$

$$\|\nabla_\theta L(\theta_t)\|^2 = \sum_i \lambda_i^2 \tilde{\theta}_t^{(i)2}, \tag{117}$$

$$H\nabla_\theta L(\theta_t) = \sum_i \lambda_i q_i q_i^\top \sum_i \lambda_i \tilde{\theta}_t^{(i)} q_i = \sum_i \lambda_i^2 \tilde{\theta}_t^{(i)} q_i, \tag{118}$$

$$\nabla_\theta L(\theta_t)^\top H \nabla_\theta L(\theta_t) = \sum_i \lambda_i \tilde{\theta}_t^{(i)} q_i^\top \sum_i \lambda_i^2 \tilde{\theta}_t^{(i)} q_i = \sum_i \lambda_i^3 \tilde{\theta}_t^{(i)2}, \tag{119}$$

where $\tilde{\theta}_t^{(i)} = q_i^\top \theta_t + \frac{q_i^\top}{\lambda_i} b$. Therefore, we have $\lambda_1 - \|H\|_{S_n} = O(e^{-2ct})$ from the following:

$$\|H\|_{S_n} = \frac{\nabla_\theta L^\top H \nabla_\theta L}{\|\nabla_\theta L\|^2} = \frac{\sum_i \lambda_i^3 \tilde{\theta}_t^{(i)2}}{\sum_i \lambda_i^2 \tilde{\theta}_t^{(i)2}} = \frac{\lambda_1 \tilde{\theta}_t^{(1)2} + \frac{\lambda_2^3}{\lambda_1^2} \tilde{\theta}_t^{(2)2} + \cdots}{\tilde{\theta}_t^{(1)2} + \frac{\lambda_2^2}{\lambda_1^2} \tilde{\theta}_t^{(2)2} + \cdots}, \tag{120}$$

$$\lambda_1 - \|H\|_{S_n} = \frac{(\frac{\lambda_2^2}{\lambda_1} - \frac{\lambda_2^3}{\lambda_1^2}) \tilde{\theta}_t^{(2)2} + \cdots}{\tilde{\theta}_t^{(1)2} + \frac{\lambda_2^2}{\lambda_1^2} \tilde{\theta}_t^{(2)2} + \cdots} = \frac{\frac{\lambda_2^2}{\lambda_1^2}(\lambda_1 - \lambda_2) \tilde{\theta}_t^{(2)2} + \cdots}{\tilde{\theta}_t^{(1)2} + \frac{\lambda_2^2}{\lambda_1^2} \tilde{\theta}_t^{(2)2} + \cdots}, \tag{121}$$

$$0 \le \lambda_1 - \|H\|_{S_n} \le \frac{1}{\tilde{\theta}_t^{(1)2}} \text{ for } t \ge t_2 \text{ for some } t_2 \tag{122}$$

since $\lim_{t \to \infty} \tilde{\theta}_t^{(i)} = 0$ for $i > 1$. $\qquad\square$

**Remark.** *Due to the exponential convergence of $\|H\|_{S_n}$ to $\lambda_1$, it only takes a few steps (5-20) for $\|H\|_{S_n}$ to exceed $\frac{2}{\eta}\gamma(\beta_1, \beta_2)$ (see Appendix C.3).*

**Remark.** *In practice, $\|H\| = \lambda_1$ keeps increasing after it exceeds $\frac{2}{\eta}\gamma(\beta_1, \beta_2)$. Therefore, we may relax the assumption as the eigenvalues $\lambda_1(\theta_t) > \frac{2}{\eta}\gamma(\beta_1, \beta_2) + \epsilon_1$ and $\epsilon_2 < \lambda_i(\theta_t) < \frac{2}{\eta}\gamma(\beta_1, \beta_2) - \epsilon_3$ for $i \ne 1$ may change within the bounds over $t$ for some $\epsilon_1, \epsilon_2, \epsilon_3 > 0$ ($q_i$'s are fixed). We can draw the same conclusion except that the limit $\|H\|_{S_n}$ may not exist because of varying $\|H\|$ according to t, but we can ensure that $\|H\|_{S_n}$ eventually exceeds $\frac{2}{\eta}\gamma(\beta_1, \beta_2)$.*

## C    EXPERIMENTAL SETTINGS AND ADDITIONAL FIGURES

We report the experimental results using vanilla SGD/GD without momentum and weight decay, constant learning rate, and no data augmentation. We use a simple 6-layer CNN (6CNN, $m = 0.51M$), ResNet-9 (He et al., 2016)[3] ($m = 2.3M$), WRN-28-2 (Zagoruyko & Komodakis, 2016) ($m = 36M$). We use subsets of the datasets CIFAR-10/100 (Krizhevsky & Hinton, 2009)[4], STL-10 (Coates et al., 2011)[5], and Tiny-ImageNet[6] (a subset of ImageNet (Deng et al., 2009) with $3 \times 64 \times 64$ images and 200 object classes) where DATASET-$n$ denotes a subset of DATASET with $|\mathcal{D}| = n$ and k=$2^{10} = 1024$.

Moreover, we use PyHessian (Yao et al., 2020)[7] to compute the Hessian-vector product (e.g., $H\nabla L$), the top eigenvalue $\lambda_1$ and its corresponding eigenvector $q_1$ of the Hessian. For these computations, we use the power iterations with a batch size of 2k, the tolerance of 0.001, and the maximum iteration of 100.

### C.1    FIGURE 1

In Figure 1 and Figure 2, we plot $\frac{\mathbb{E}_\xi[L_{t+1}] - L_t}{\text{tr}(S_n)}$ against $\frac{\text{tr}(HS_b)}{\text{tr}(S_n)}$, which is equivalent to $\frac{L_{t+1} - L_t}{\text{tr}(S_n)}$ against $\|H\|_{S_n}$ for GD. Therefore, we expect the following linear relationship with the slope $\frac{\eta^2}{2}$ and the x-intercept $\frac{2}{\eta}$ when the training loss $L$ is locally quadratic, i.e., $\epsilon = 0$:

$$\frac{\mathbb{E}_\xi[L_{t+1}] - L_t}{\text{tr}(S_n)} = \frac{\eta^2}{2}\left(\frac{\text{tr}(HS_b)}{\text{tr}(S_n)} - \frac{2}{\eta}\right), \tag{123}$$

$$\frac{L_{t+1} - L_t}{\text{tr}(S_n)} = \frac{\eta^2}{2}\left(\|H\|_{S_n} - \frac{2}{\eta}\right). \tag{124}$$

Figure 1 shows the behavior in the early phase, until the iterate enters the edge of stability. For GD, they are plotted *after* $\|H\|$ exceeds $\frac{2}{\eta}$ after which $\|H\|_{S_n}$ starts to increase from 0 to $\frac{2}{\eta}$ in a few steps. For cross-entropy loss, we mark the end point with 'x' when the iterate enters the unstable regime.

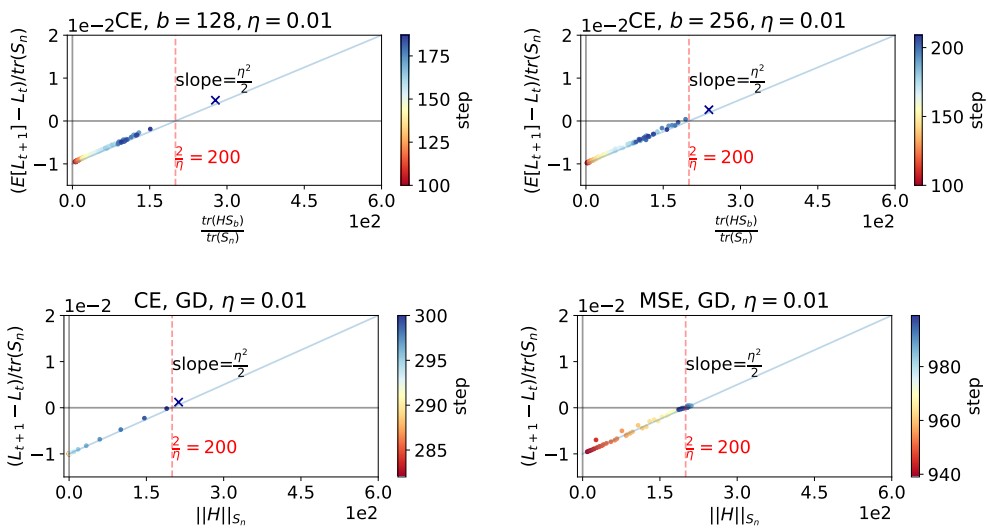

Figure 6:    6CNN with $\eta = 0.01$. See caption of Figure 1 for more details.

---

[3]from `https://github.com/wbaek/torchskeleton`, Apache-2.0 license
[4]`https://www.cs.toronto.edu/~kriz/cifar.html`
[5]`https://cs.stanford.edu/~acoates/stl10/`
[6]`https://tiny-imagenet.herokuapp.com`
[7]`https://github.com/amirgholami/PyHessian`, MIT license

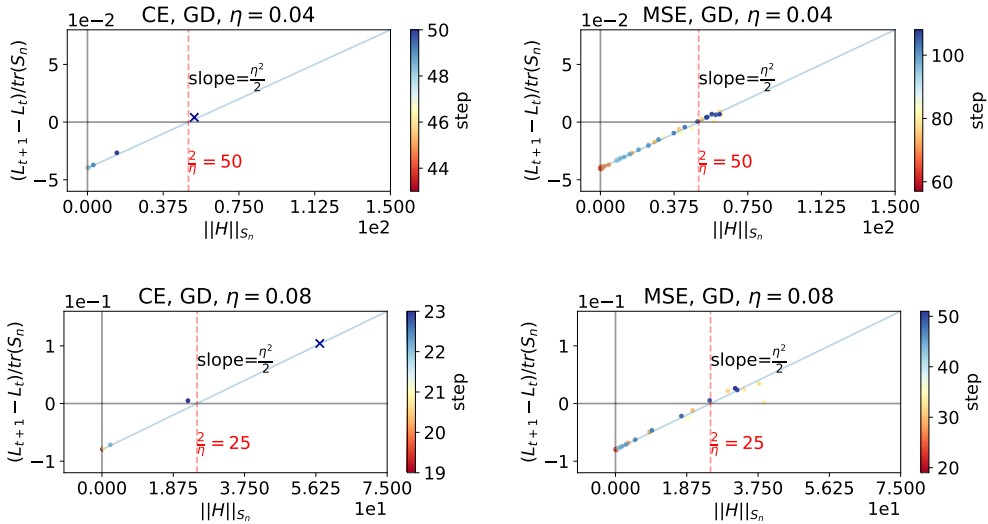

Figure 7: 6CNN with CE/MSE (left/right) and $\eta = 0.04/0.08$ (top/bottom). See caption of Figure 1 for more details.

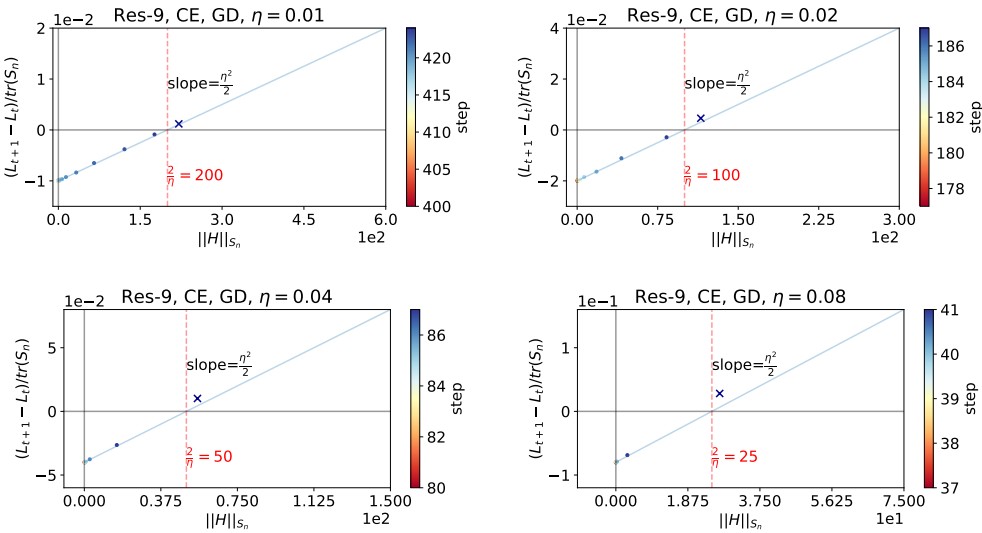

Figure 8: ResNet-9 with CE and $\eta = 0.01/0.02/0.04/0.08$. See caption of Figure 1 for more details.

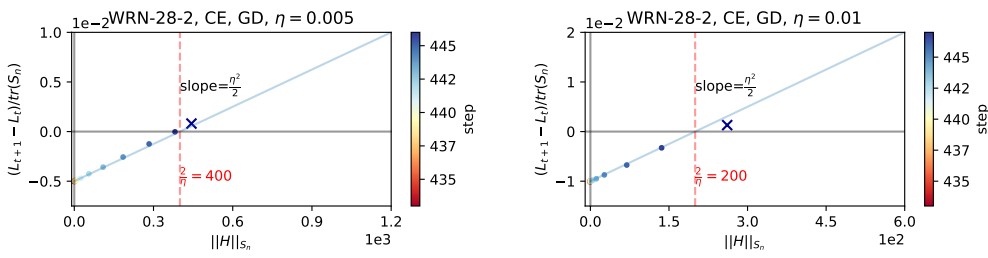

Figure 9: WRN-28-2 with $\eta = 0.005/0.01$. See caption of Figure 1 for more details.

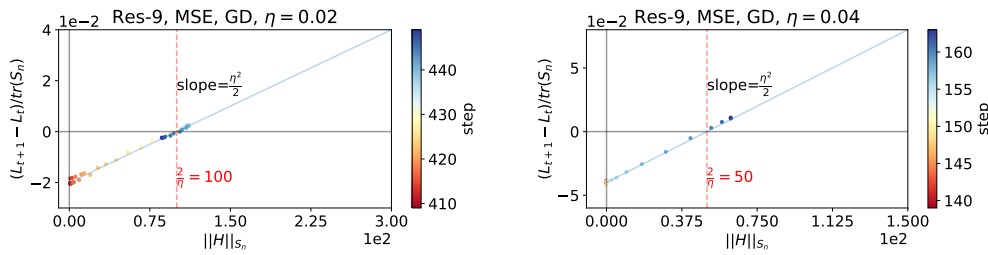

Figure 10: ResNet-9 with MSE and $\eta = 0.02/0.04$. See caption of Figure 1 for more details.

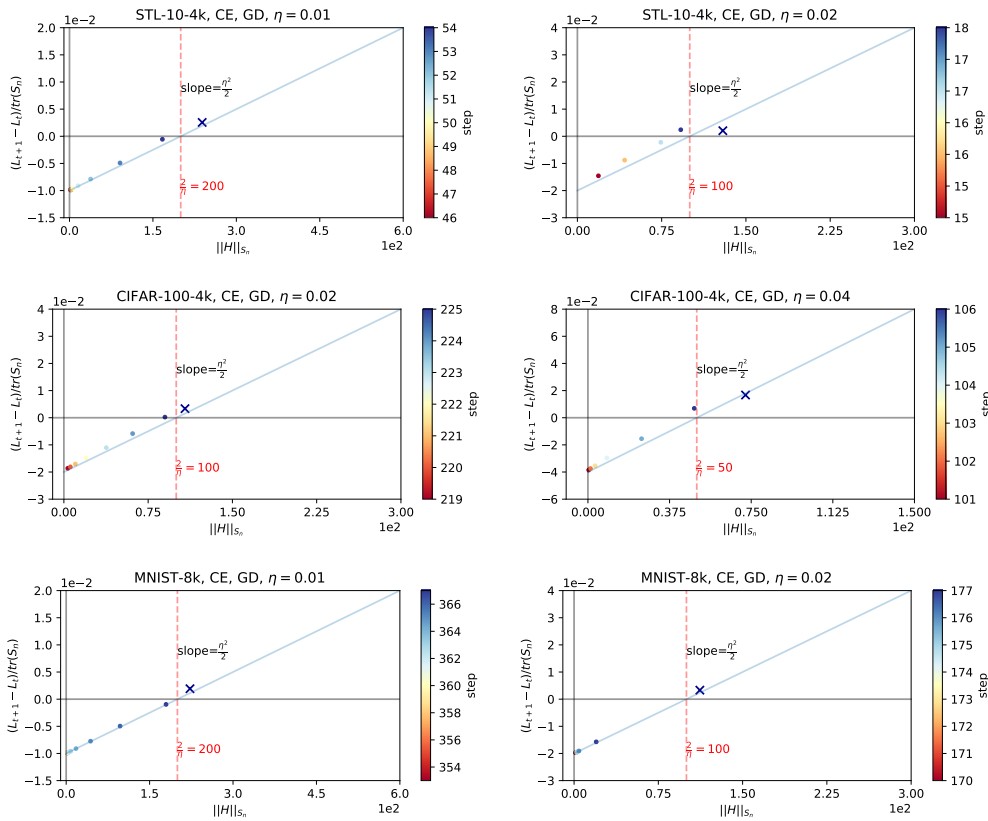

Figure 11: (DATASET, $\eta$) = (STL-10-4k, 0.01/0.02), (CIFAR-100-4k, 0.02/0.04), (MNIST-8k, 0.02/0.04). See caption of Figure 1 for more details.

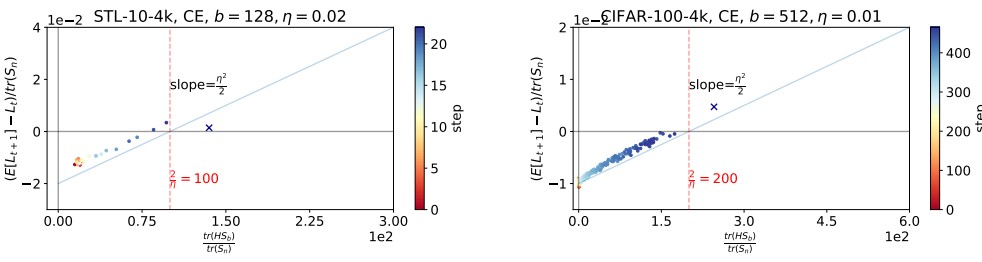

Figure 12: (DATASET, $b$, $\eta$) = (STL-10-4k, 128, 0.02), (CIFAR-100-4k, 512, 0.01), (MNIST-8k, 0.02/0.04). See caption of Figure 1 for more details.

## C.2 Figure 2

Figure 2 shows the non-quadraticity (represented by deviation from the blue line in Figure 2 and the following figures) of the training loss function $L$ *after* the iterate enters the edge of stability.

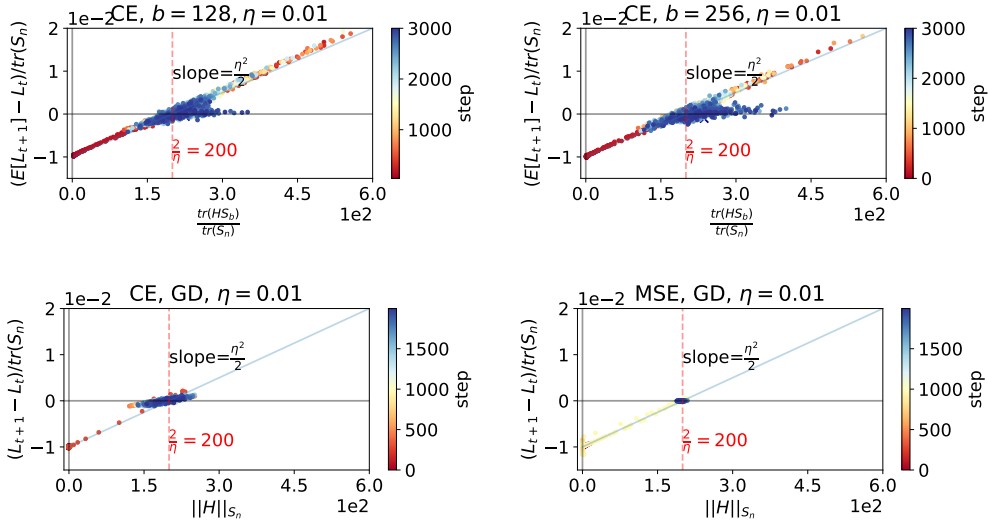

Figure 13: 6CNN with $\eta = 0.01$. See caption of Figure 2 for more details.

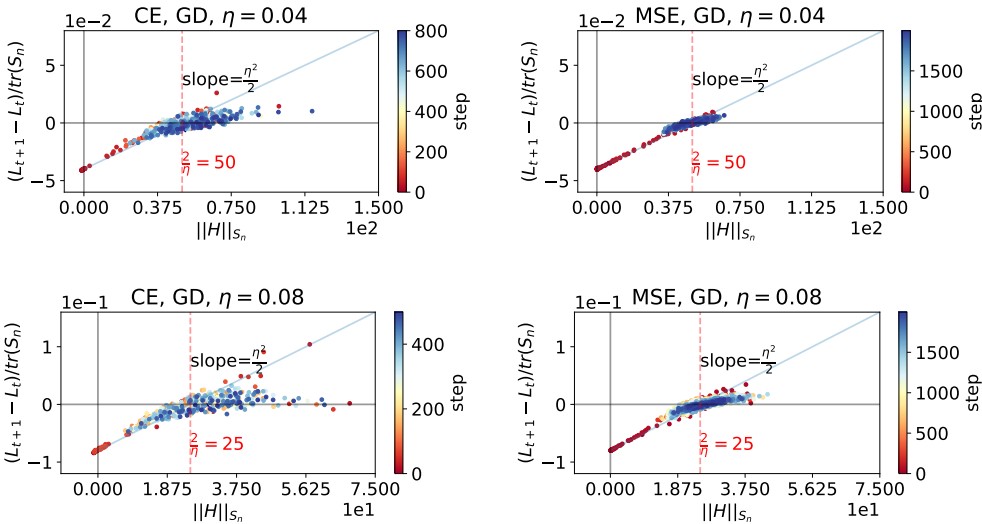

Figure 14: 6CNN with CE/MSE (left/right) and $\eta = 0.04/0.08$ (top/bottom). See caption of Figure 2 for more details.

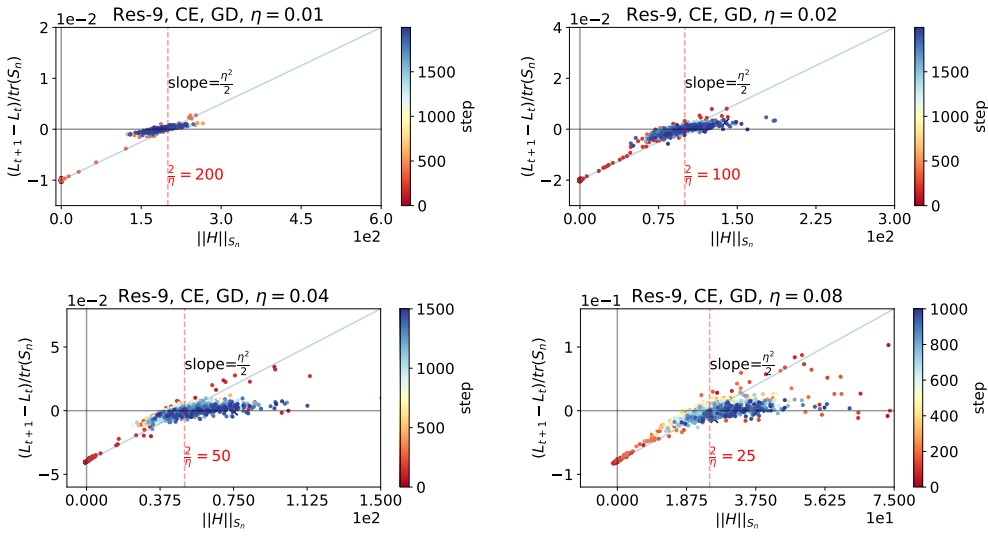

Figure 15: ResNet-9 with CE and $\eta = 0.01/0.02/0.04/0.08$. See caption of Figure 2 for more details.

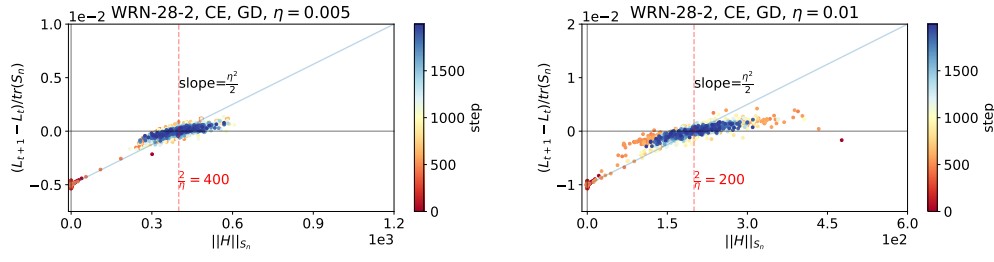

Figure 16: WRN-28-2 with $\eta = 0.005/0.01$. See caption of Figure 2 for more details.

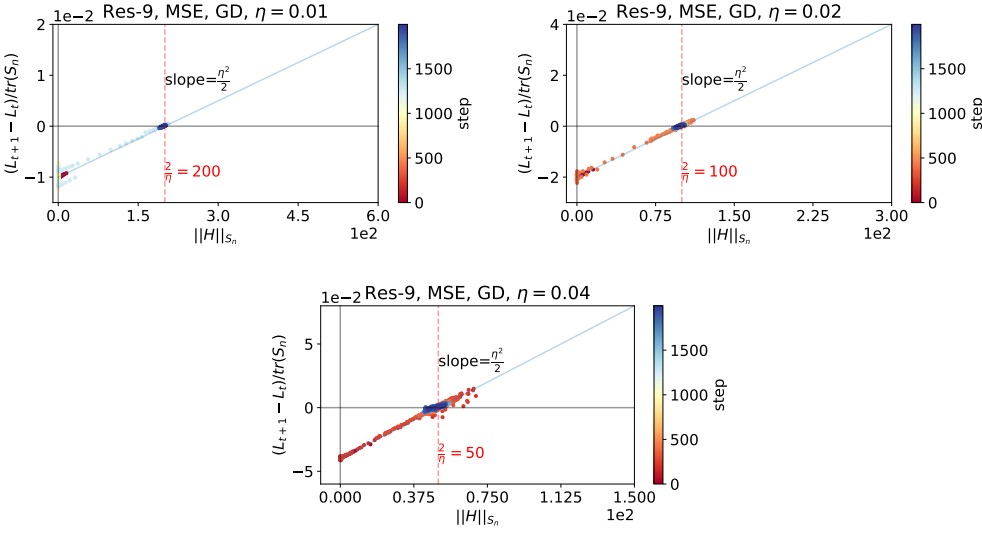

Figure 17: ResNet-9 with MSE and $\eta = 0.01/0.02/0.04$. See caption of Figure 2 for more details.

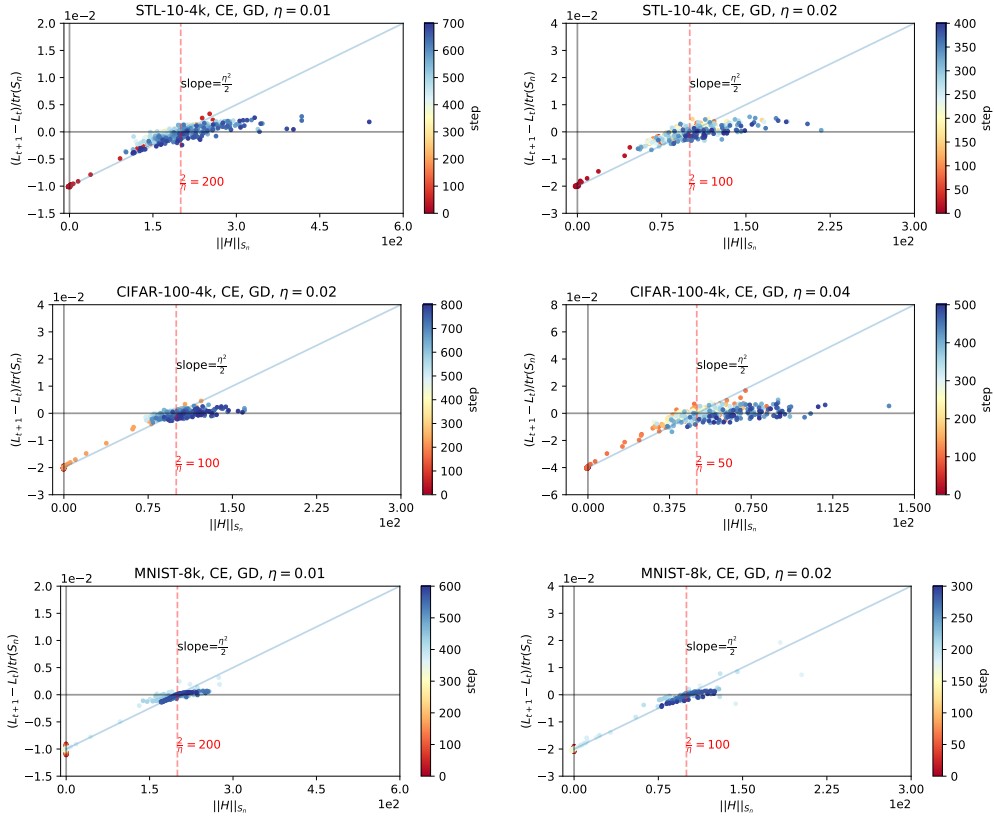

Figure 18: (DATASET, $\eta$) = (STL-10-4k, 0.01/0.02), (CIFAR-100-4k, 0.02/0.04), (MNIST-8k, 0.01/0.02). See caption of Figure 2 for more details.

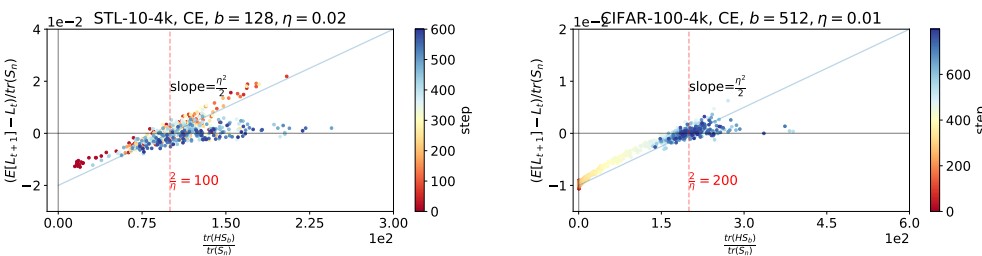

Figure 19: (DATASET, $b$, $\eta$) = (STL-10-4k, 128, 0.02), (CIFAR-100-4k, 512, 0.01). See caption of Figure 2 for more details.

### C.3 FIGURE 3 AND THEOREM 2

Figures 20-21 provide additional information of Figure 3. After these figures, we focus on providing some empirical evidences of Theorem 2 that $\|H\|_{S_n}$, $|\cos(q_1, \nabla L)|$ and $|q_1^\top \nabla L|$ increase in a few steps after $\|H\|$ exceeds $\frac{2}{\eta}$.

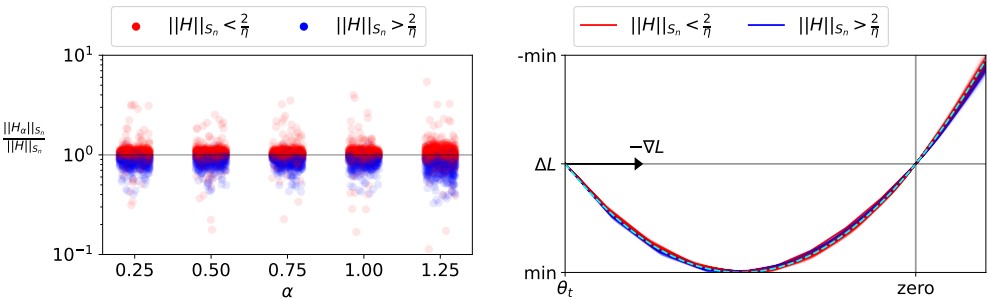

Figure 20: 6CNN with $\eta = 0.02$. See caption of Figure 3 for more details.

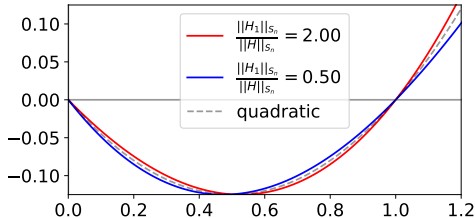

Figure 21: To gain some intuitions of Figure 3 and 20 (right), we plot three graphs —a quadratic function $f(x) = \frac{1}{2}x(x-1)$ (black dashed line), and cubic functions $f_1(x) = \alpha_1 x(x-1)(1 + \frac{1}{4}x)$ in red and $f_2(x) = \alpha_2 x(x-1)(1 - \frac{1}{5}x)$ in blue. They are chosen to satisfy $\frac{\partial^2 f_1}{\partial x^2}|_{x=1} / \frac{\partial^2 f_1}{\partial x^2}|_{x=0} = 2$ and $\frac{\partial^2 f_2}{\partial x^2}|_{x=1} / \frac{\partial^2 f_2}{\partial x^2}|_{x=0} = 0.5$. We also choose $\alpha_1$ and $\alpha_2$ for $f_1$ and $f_2$ to have the same minimum with $f$ in $x \in [0,1]$, i.e., $\min_{x \in [0,1]} f(x) = \min_{x \in [0,1]} f_i(x) = -\frac{1}{8}$ for $i = 1, 2$.

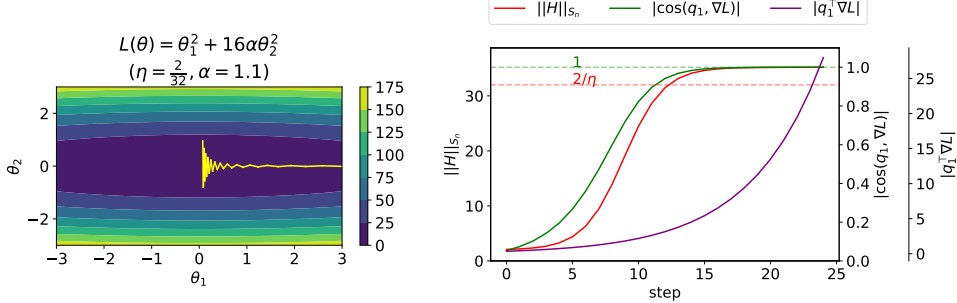

Figure 22: Left: A toy two-dimensional loss function $L(\theta) = \theta_1^2 + 16\alpha\theta_2^2$ where $\alpha = 1.1$. We optimize the loss by GD with $\eta = \frac{2}{32}$ so that $\|H\| = 32\alpha > \frac{2}{\eta} = 32$. We also plot the GD trajectory in yellow starting from $(\theta_1, \theta_2) = (3, 0.1)$. Right: We show the exponential increase in $|q_1^\top \nabla L|$ (purple) and the S-shape increase in $\|H\|_{S_n}$ (red) and $|\cos(q_1, \nabla L)|$ (green) to $\|H\|$ and 1, respectively, which empirically demonstrates Theorem 2. We also note that they start to increase in the order of $|\cos(q_1, \nabla L)|$, $\|H\|_{S_n}$ and $|q_1^\top \nabla L|$.

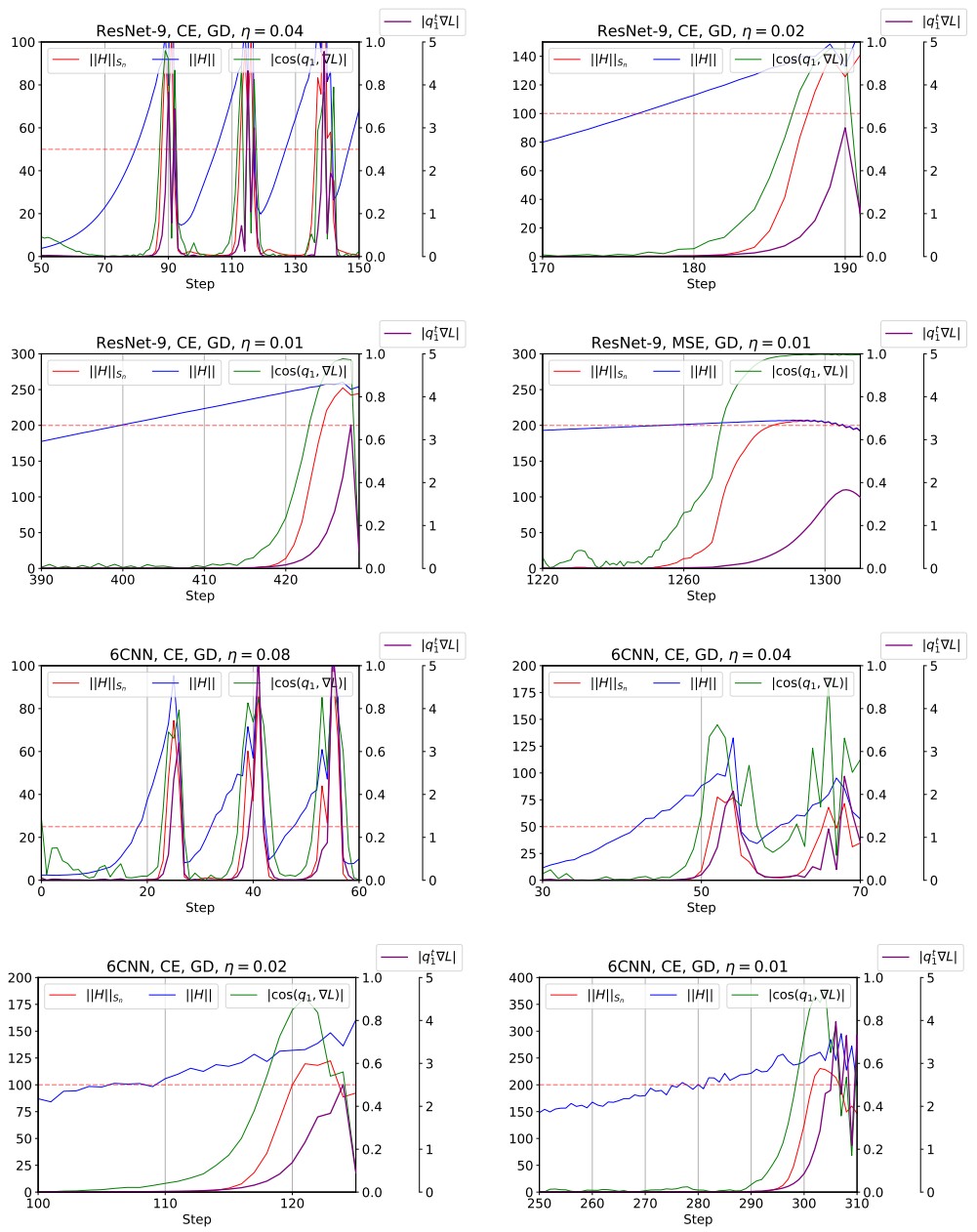

Figure 23: After $\|H\|$ (blue) exceeds $\frac{2}{\eta}$ (red dashed line) at $t \approx 80/176/400/1250/19/43/110/280$, in a few steps ($\approx 5/6/18/5/5/6/5/15$), $\|H\|_{S_n}$ (red) starts to increase. As expected in Theorem 2, $\|H\|_{S_n}$ increases together with $|\cos(q_1, \nabla L)|$ (green) and $|q_1^\top \nabla L|$ (purple). They are observed to start to increase in the order of $|\cos(q_1, \nabla L)|$, $\|H\|_{S_n}$ and $|q_1^\top \nabla L|$, as shown in Figure 22.

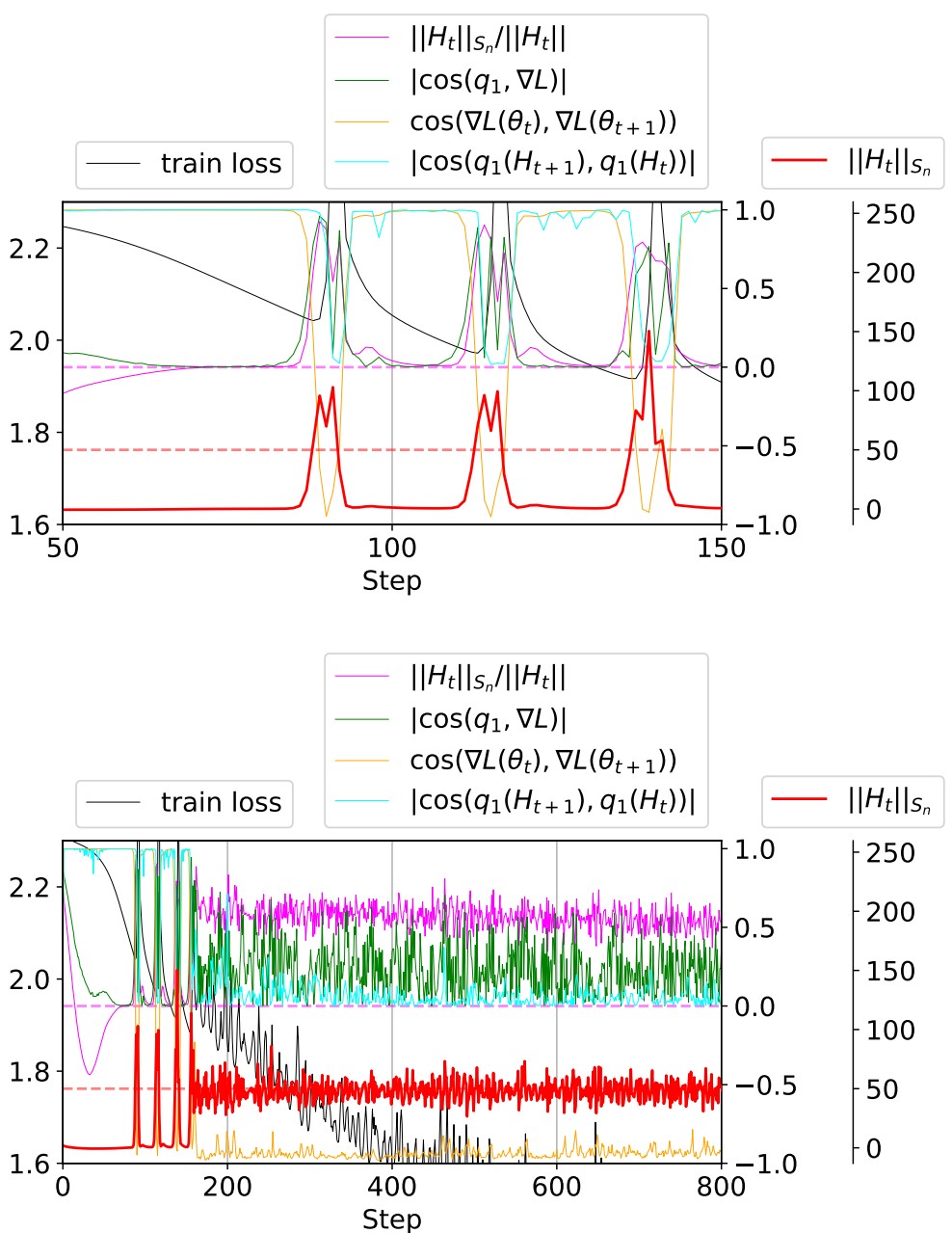

Figure 24: After $\|H\|$ exceeds $\frac{2}{\eta}$ (not shown in this Figure, see Figure 23), the cosine $|\cos(q_1(H_t), \nabla L(\theta_t))|$ (green) between the sharpest direction and the gradient gets large, where $H_t = H(\theta_t)$. Simultaneously, $\|H_t\|_{S_n}$ increases and exceeds $\frac{2}{\eta}$ (red solid > red dashed), the iterate entering the unstable regime and oscillating with $\cos(\nabla L(\theta_t), \nabla L(\theta_{t+1})) \approx -1$ (orange). However, due to the non-quadraticity, the sharpest direction changes with $|\cos(q_1(H_{t+1}), q_1(H_t))|$ (cyan) close to 0. We train ResNet-9 on CIFAR-10-8k with $\eta = 0.04$ (top: steps 50-150, bottom: steps 0-800).

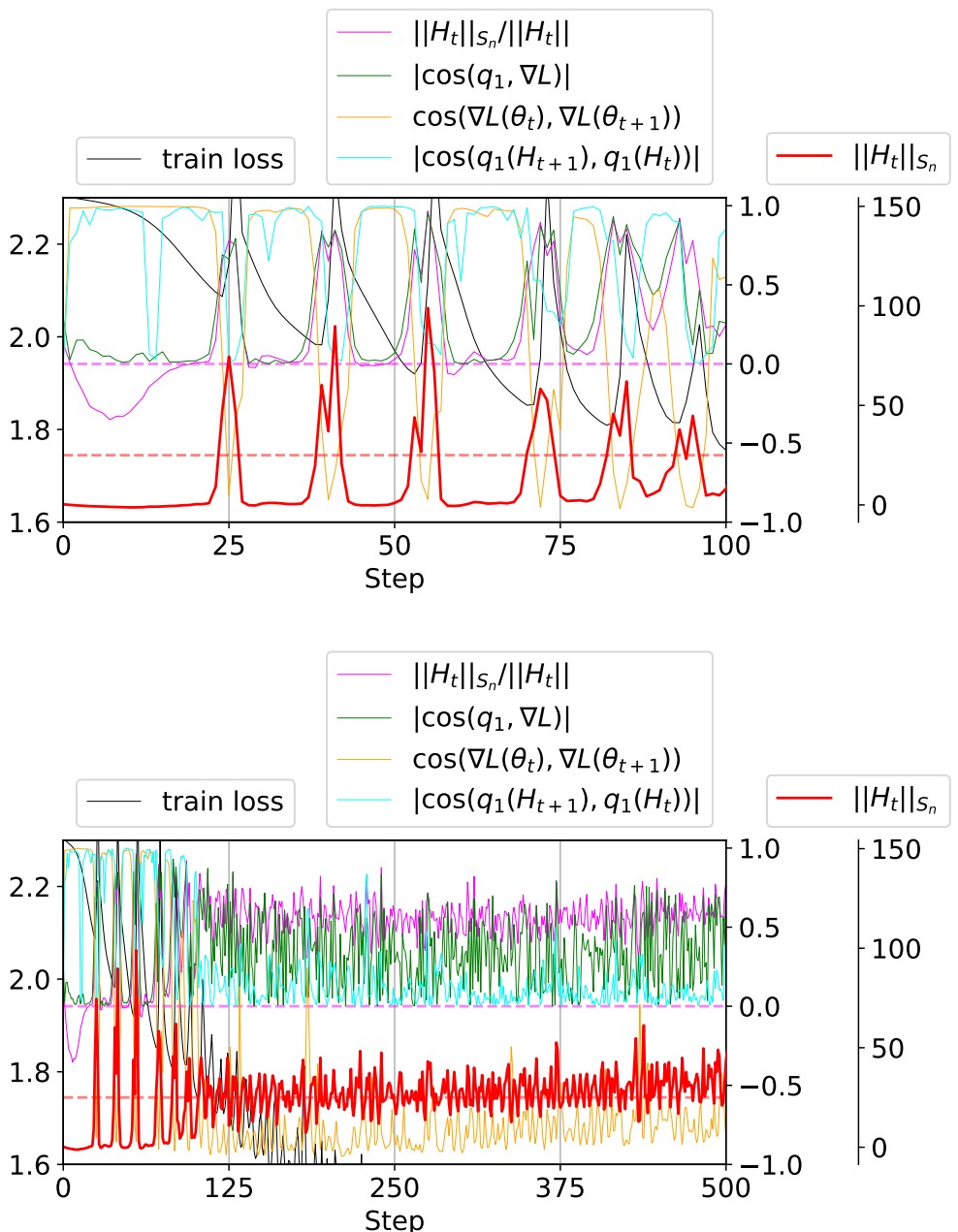

Figure 25: We train ResNet-9 on CIFAR-10-8k with $\eta = 0.08$ (top: steps 0-100, bottom: steps 0-500). See caption of Figure 24 for more details.

## C.4 FIGURE 4

### C.4.1 GD

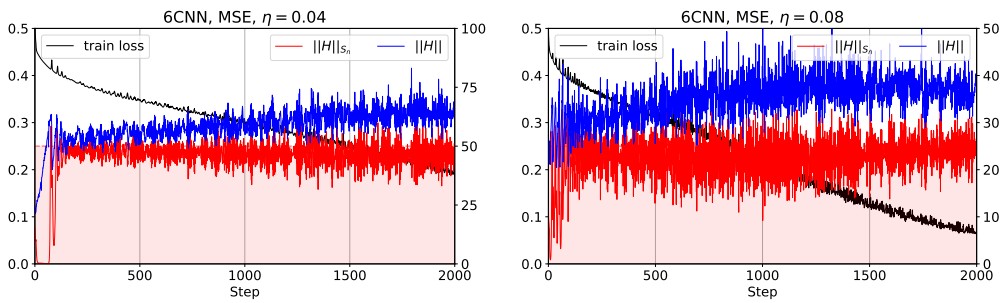

Figure 26: 6CNN, MSE, and $\eta = 0.04/0.08$. See caption of Figure 4 for more details.

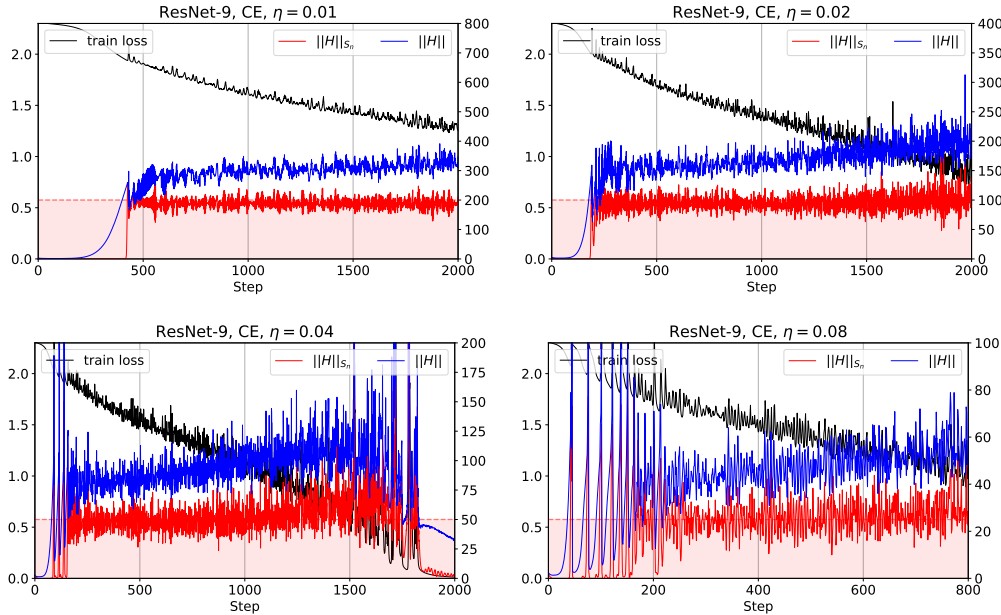

Figure 27: ResNet-9, CE, and $\eta = 0.01/0.02/0.04/0.08$. See caption of Figure 4 for more details.

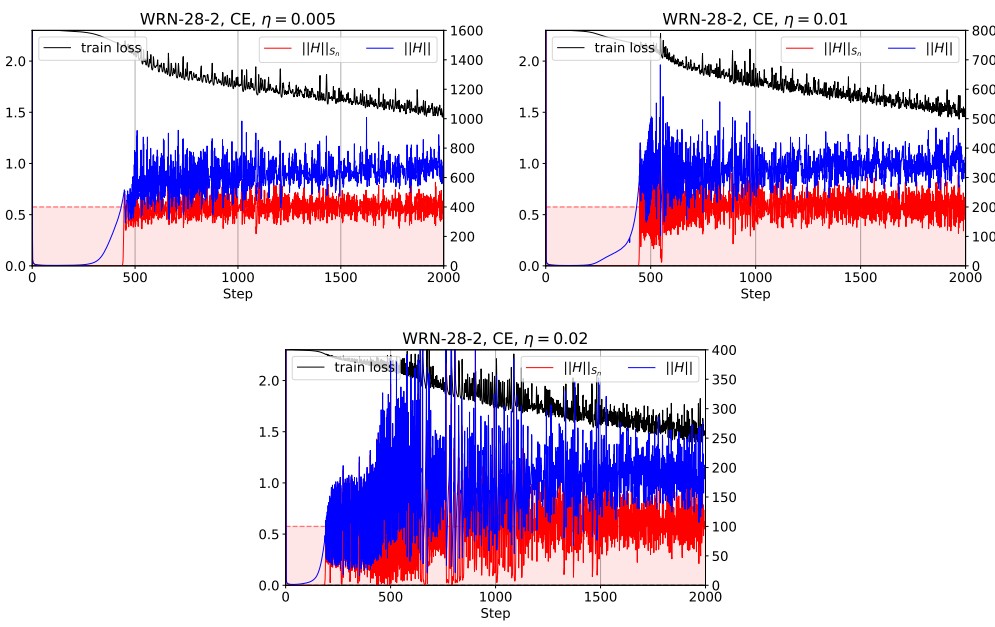

Figure 28: WRN-28-2, CE, and $\eta = 0.005/0.01/0.02$. See caption of Figure 4 for more details.

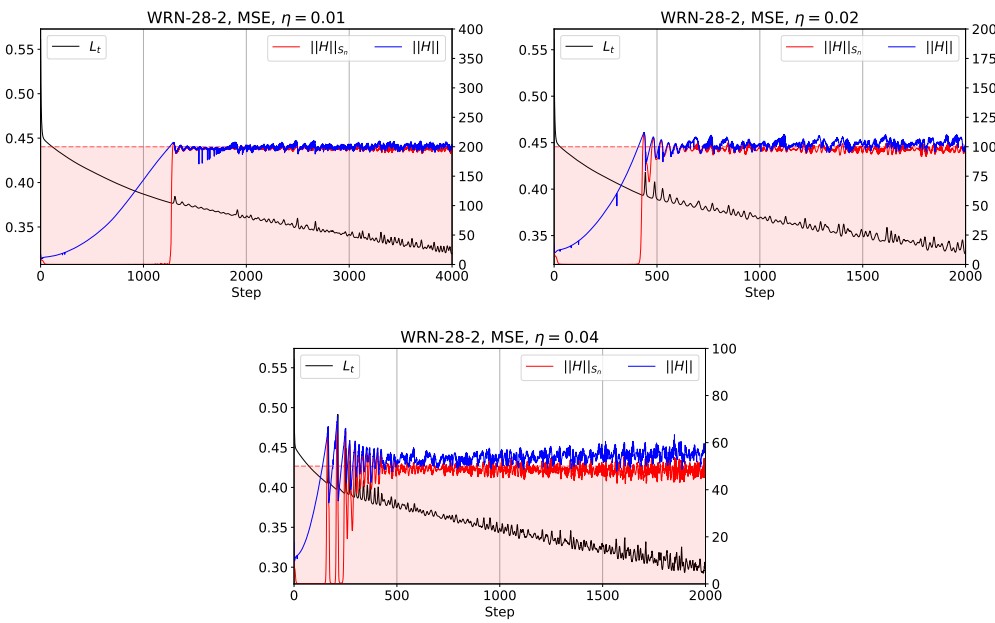

Figure 29: WRN-28-2, MSE, and $\eta = 0.01/0.02/0.04$. See caption of Figure 4 for more details.

### C.4.2 SGD

We demonstrate two types of IIR, (i) $\|H\|_{S_b} \leq \frac{2\rho_b}{\eta}$ and (ii) $\frac{\text{tr}(HS_b)}{\text{tr}(S_n)} \leq \frac{2}{\eta}$ in Figure 30 and 31, respectively. They are equivalent, but the latter shows the effects of IIR more clearly as it has the fixed threshold $\frac{2}{\eta}$ regardless of $b$.

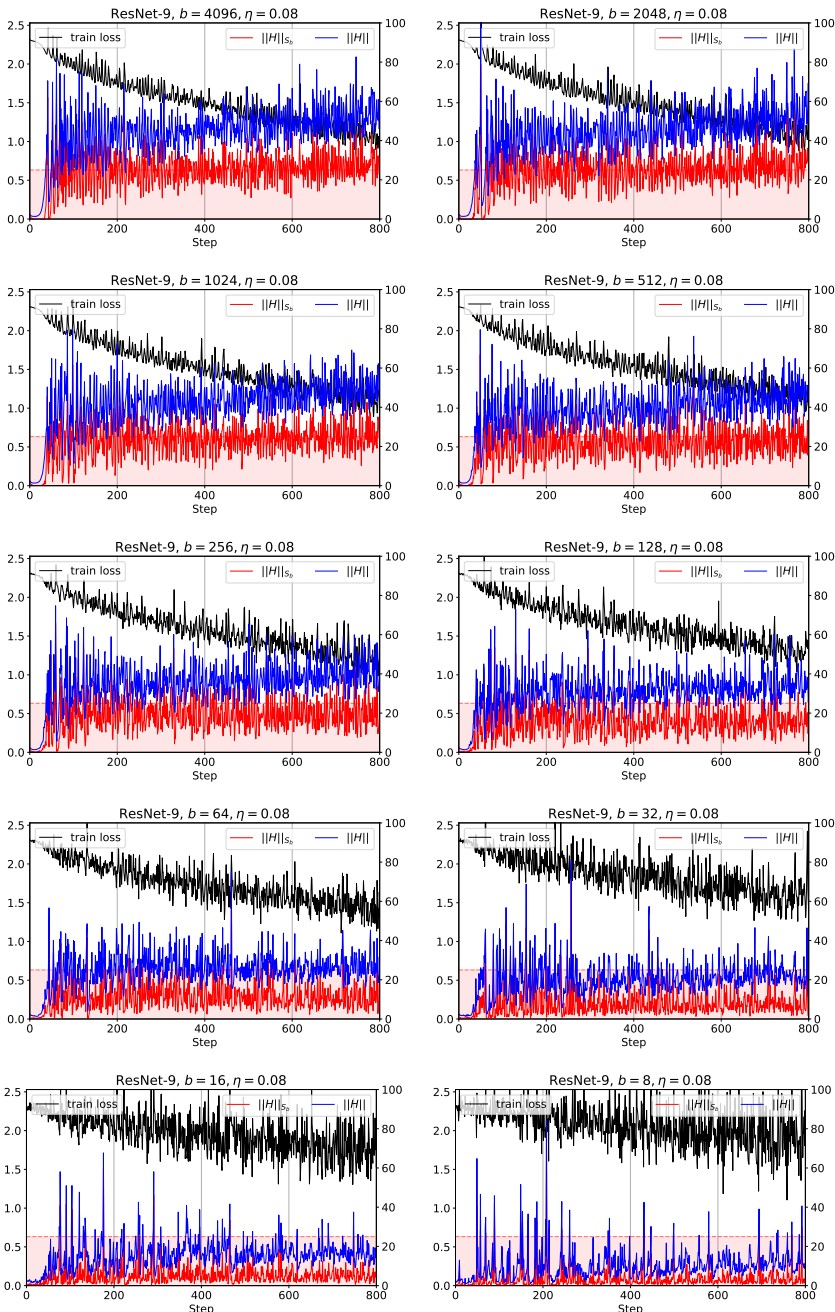

Figure 30: IIR of $\|H\|_{S_b} \leq \frac{2\rho_b}{\eta}$ for SGD. We plot $\frac{2}{\eta}$ (red dashed line) which is not the threshold for $\|H\|_{S_b}$. ResNet-9, CE, $\eta = 0.08$, and $b \in \{2^{12}, 2^{11}, \cdots, 2^3\}$. See caption of Figure 4 for more details.

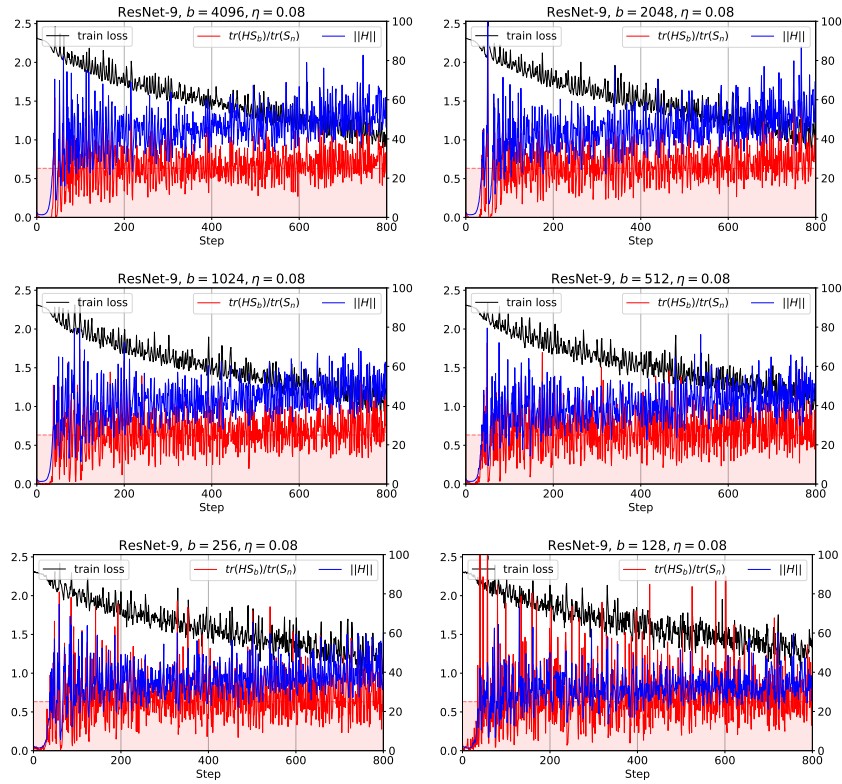

Figure 31: IIR of $\frac{\mathrm{tr}(HS_b)}{\mathrm{tr}(S_n)} \leq \frac{2}{\eta}$ for SGD. With the upper bound $\frac{2}{\eta}$ (red dashed line), this shows the effects of IIR more clearly than Figure 30. ResNet-9, CE, $\eta = 0.08$, and $b \in \{2^{12}, 2^{11}, \cdots, 2^7\}$.

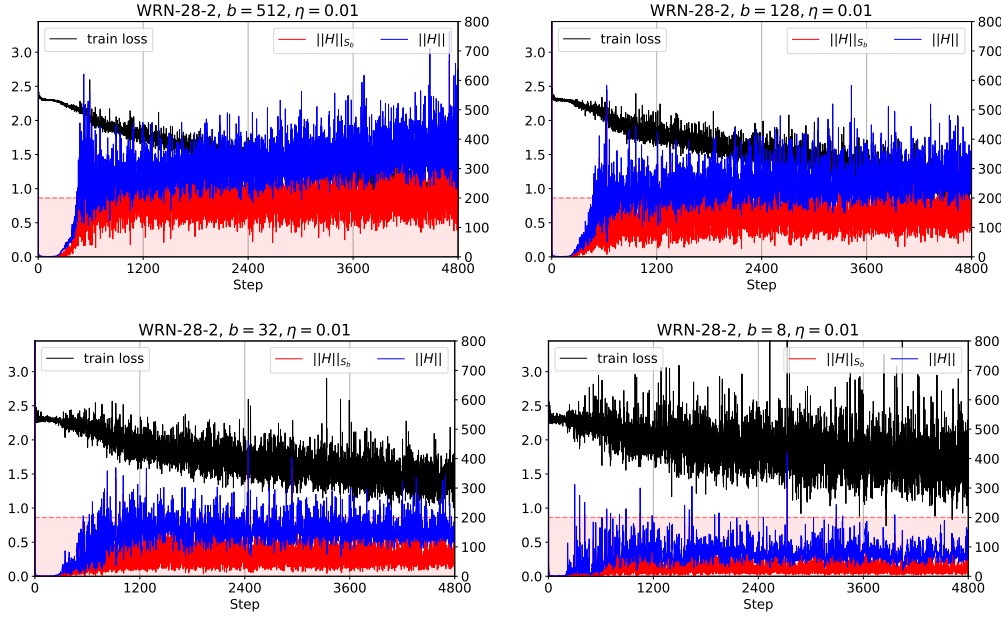

Figure 32: IIR of $\|H\|_{S_b} \leq \frac{2\rho_b}{\eta}$ for SGD. WRN-28-2, CE, $\eta = 0.01$, and $b \in \{2^9, 2^7, 2^5, 2^3\}$. See caption of Figure 4 for more details.

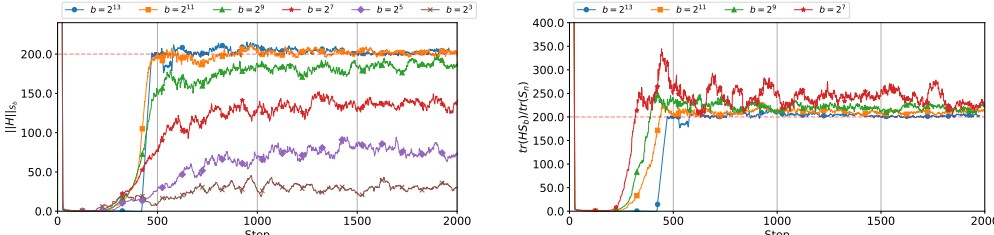

Figure 33: Left/Right: See caption of Figure 4(c)/4(d) for more details. WRN-28-2, CIFAR-10-8k, $\eta = 0.01$.

## C.5 FIGURE 5

Figure 34-35 provide some additional information of Figure 5. Figure 36 shows that $1/\rho \approx 100$ is much larger than $1$ and that $\text{std}[\|g_\xi\|]$ is 2-3$\times$ smaller than $\mathbb{E}_\xi[\|g_\xi\|]$ even in the case of $b = 1$. Therefore, we use the approximation $\|g_\xi\| \approx \mathbb{E}_\xi[\|g_\xi\|]$, and thus the square of the mean resultant length is similar to the concentration measure $\rho_b$ as shown in the following approximation:

$$\bar{R}_b^2 \equiv \left\|\mathbb{E}_\xi\left[\frac{g_\xi}{\|g_\xi\|}\right]\right\|^2 \approx \left\|\mathbb{E}_\xi\left[\frac{g_\xi}{\mathbb{E}_\xi[\|g_\xi\|]}\right]\right\|^2 = \left\|\frac{\mathbb{E}_\xi[g_\xi]}{\mathbb{E}_\xi[\|g_\xi\|]}\right\|^2 = \rho_b. \tag{125}$$

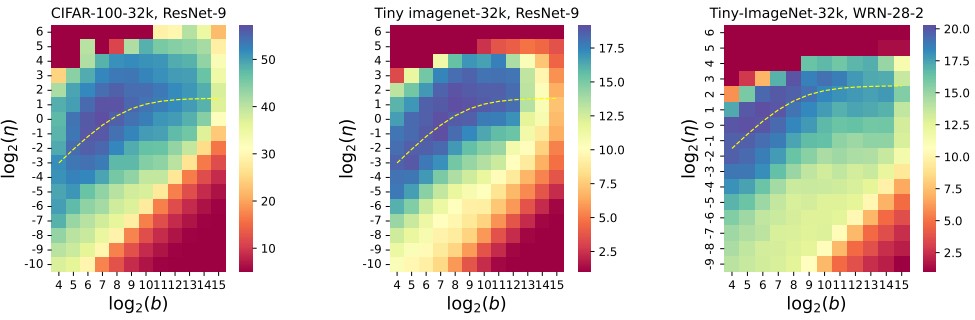

Figure 34: (DATASET, MODEL, EPOCHS) = (CIFAR-100-32k, ResNet-9, 800 epochs), (Tiny-ImageNet-32k, ResNet-9, 400 epochs), (Tiny-ImageNet-32k, WRN-28-2, 800 epochs). Middle: The model is trained for 400 epochs, which is short compared to Figure 5 (bottom right heatmap). See caption of Figure 5 (right) for more details.

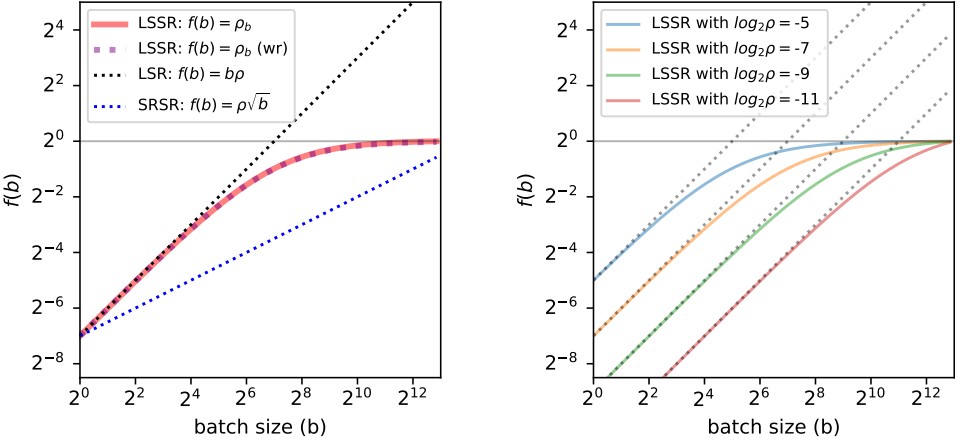

Figure 35: Left: We use $\gamma_{n,b} = 1$ for sampling with replacement ('wr', dotted purple curve), which is almost equivalent to the "without replacement" counterpart (red). Right: LSSR for different $\rho$ values with the corresponding LSR (dotted lines). See caption of Figure 5 (left) for more details.

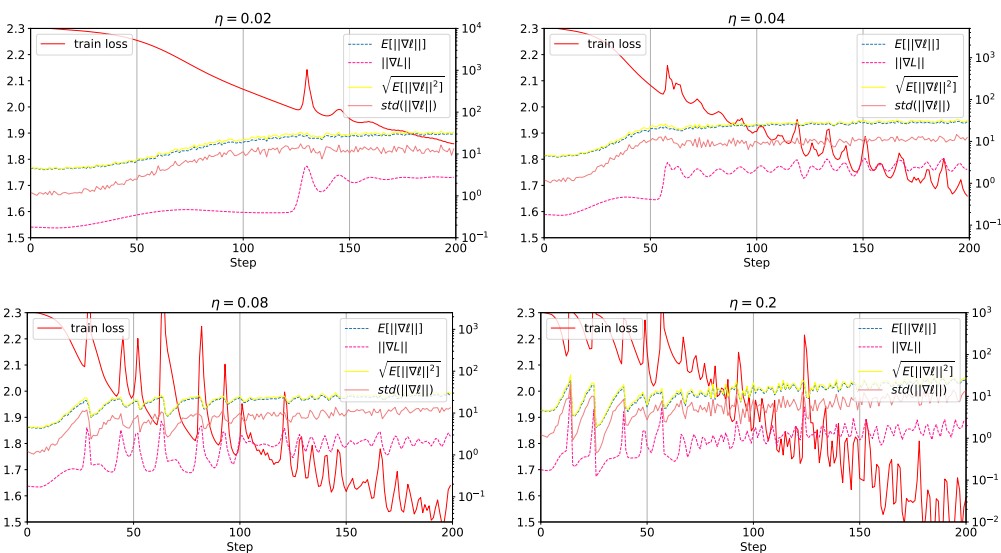

Figure 36: To further understand the concentration measure $\rho$ of the per-example gradient, we plot $\mathbb{E}[\|\nabla\ell\|], \|\nabla L\| = \|\mathbb{E}[\nabla\ell]\|, \sqrt{\operatorname{tr}(S_1)} = \mathbb{E}[\|\nabla\ell\|^2]$ and $\operatorname{std}[\|\nabla\ell\|]$. We use 100 samples to compute the expectation values and the standard deviation. Here, $\frac{1}{\rho} = \frac{\mathbb{E}[\|\nabla\ell\|^2]}{\|\nabla L\|^2}$ is about 100. $\mathbb{E}[\|\nabla\ell\|]$ is 2-3$\times$ larger than $\operatorname{std}[\|\nabla\ell\|]$. We train a 6CNN with $\eta = 0.02/0.04/0.08/0.2$ (GD).

## C.6 GENERALIZED MOMENTUM VARIANTS

For generalized momentum variants of GD with $(\beta_1, \beta_2)$, we have

$$\delta_t = \beta_1 \delta_{t-1} - \eta \nabla_\theta L(\theta_t + \beta_2 \delta_{t-1}). \tag{126}$$

When $\|H\|$ is very small in the beginning, we may approximate that $L$ is linear where $\nabla_\theta L(\theta) = g$ is constant with respect to $\theta$ in some region. Then $\delta_t$ converges to $\delta$ satisfying the following equations:

$$\delta = \beta_1 \delta - \eta g, \tag{127}$$

$$\delta = -\frac{\eta}{1 - \beta_1} g, \tag{128}$$

and thus

$$L_{t+1} - L_t \approx -\nabla_\theta L(\theta_t) \delta_t \tag{129}$$

$$\approx -\frac{\eta}{1 - \beta_1} \|\nabla_\theta L(\theta_t)\|^2. \tag{130}$$

Therefore, together with Theorem 2, we generalize (12) with the following approximation:

$$L_{t+1} - L_t \approx \frac{\eta^2}{2\gamma(\beta_1, \beta_2)(1 - \beta_1)} \operatorname{tr}(S_n) \left( \|H\|_{S_n} - \frac{2}{\eta} \gamma(\beta_1, \beta_2) \right), \tag{131}$$

where $\gamma(\beta_1, \beta_2) = \frac{1 + \beta_1}{1 + 2\beta_2}$. In Figures C.6 and C.6, we empirically validate (131) similar to Figures 1 and 4, respectively.

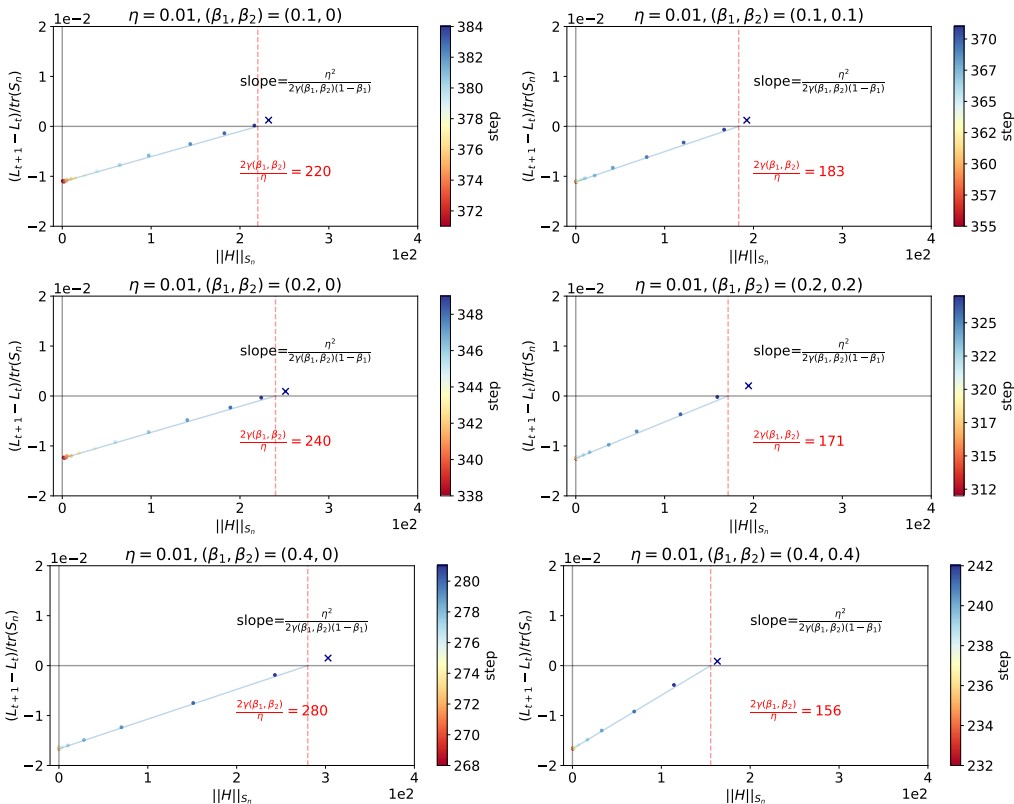

Figure 37: We use $\beta_1 = (0.1/0.2/0.4, 0)$, and $\beta_2 = 0$ (Right, Polyak momentum (Polyak, 1963)) or $\beta_2 = \beta_1$ (Left, Nesterov momentum (Nesterov, 1983)) with DATASET = CIFAR-10-8k, $\eta = 0.01$ on ResNet-9 without BN. They follows (131) before entering the edge of stability. See caption of Figure 1 for more details.

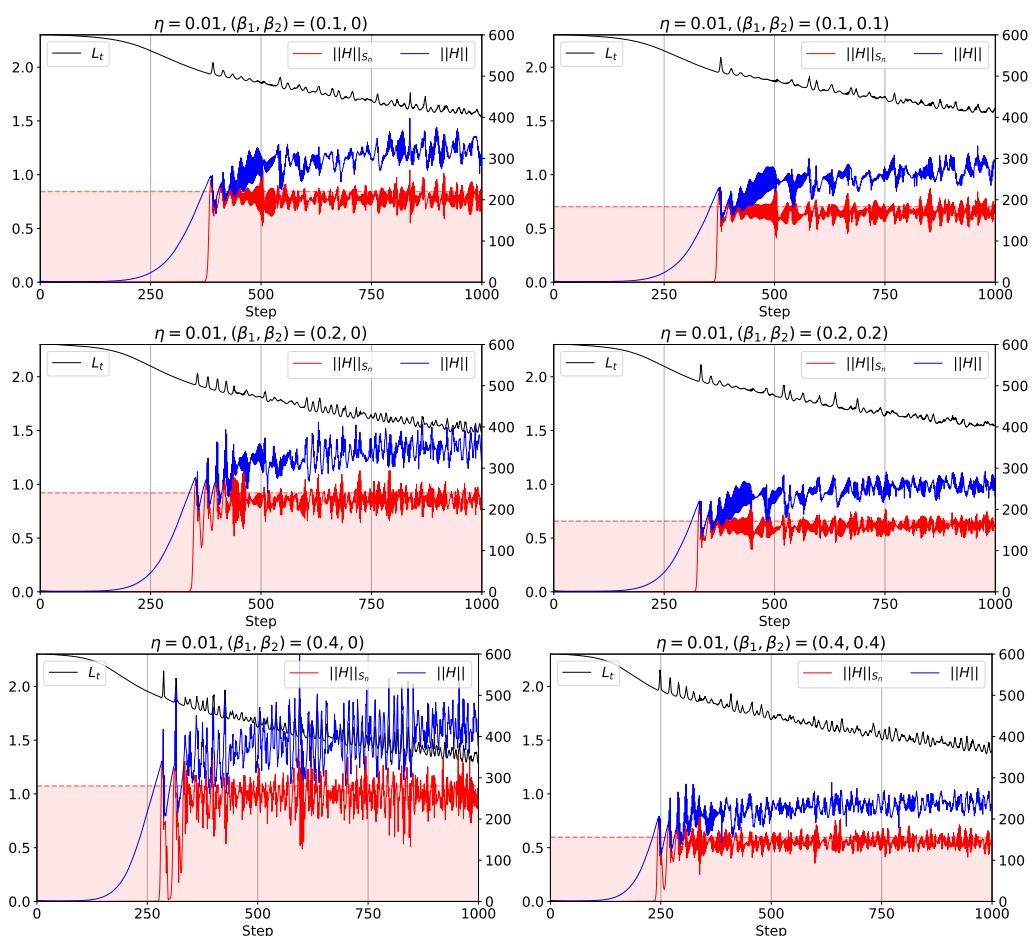

Figure 38: We use $\beta_1 = 0.1/0.2/0.4$, and $\beta_2 = 0$ (Right, Polyak momentum (Polyak, 1963)) or $\beta_2 = \beta_1$ (Left, Nesterov momentum (Nesterov, 1983)) with DATASET = CIFAR-10-8k, $\eta = 0.01$ on ResNet-9 without BN. The red horizontal lines indicate $\frac{2}{\eta}\gamma(\beta_1, \beta_2)$ where $\gamma(\beta_1, \beta_2) = \frac{1+\beta_1}{1+2\beta_2}$. See caption of Figure 4 for more details.

## D  DISCUSSION

We provide a new insight on the link between the batch gradient distribution and the sharpness of the loss landscape. In this section, we reconcile our arguments with some previous studies.

Jastrzębski et al. (2017) explain the optimization behavior of SGD with the SDE approximation $d\theta_t = -\nabla L(\theta_t)dt + \sqrt{\frac{\eta}{b}}C_1^{1/2}dW(t)$ of the SGD where $W$ is an $m$-dimensional Brownian motion. Therefore, the same ratio $\frac{\eta}{b} = \frac{\eta'}{b'}$ leads to the same SDE, which implies LSR. Moreover, a large $\frac{\eta}{b}$ implies a large diffusion in SDE, which has been linked with the escaping efficiency from a sharp local minimum in Zhu et al. (2019). Our arguments are free from any other problems raised for the SDE-based analyses of SGD which assume vanishing learning rates (Mandt et al., 2016; 2017; Hu et al., 2019; Li et al., 2017; 2019a; Jastrzębski et al., 2017; Smith & Le, 2018; Chaudhari & Soatto, 2018), e.g., the mismatch to practical finite learning rate regime or the inherent theoretical issues in the SDE approximations (Yaida, 2019; Li et al., 2021). We instead argue that a large second moment $\mathrm{tr}(S_b)$ (compared to $\mathrm{tr}(S_n)$) and a large $\eta$ lead to a low constraint $2\rho_b/\eta$ on the interaction-aware sharpness.

Wu et al. (2020) empirically show that what is important for the generalization performance of a neural network is not the class to which the gradient distribution belongs, but the second moment of the distribution. This is consistent with our arguments with the interaction $\mathrm{tr}(HS_b)$ and the

concentration measure $\rho_b = \mathrm{tr}(S_n)/\mathrm{tr}(S_b)$, because they depend on the second moment $S_b$, not on the class of the gradient distribution.

Recently, Li et al. (2021) suggest a necessary condition that the "noise-to-signal ratio" needs to be large for LSR (and the SDE assumption) to hold. This is consistent with our result on the linear regime (where $b$ and $\rho_b$ are small) because the noise-to-signal ratio is approximately the inverse of the "signal-to-noise" ratio $\rho_b = \mathrm{tr}(S_n)/\mathrm{tr}(S_b)$, but defined for an equilibrium distribution. We provide not only the necessary condition but also the sufficient condition for LSR with a novel scaling rule LSSR applicable to every batch size including where LSR fails (the saturation regime).

# E    OTHERS

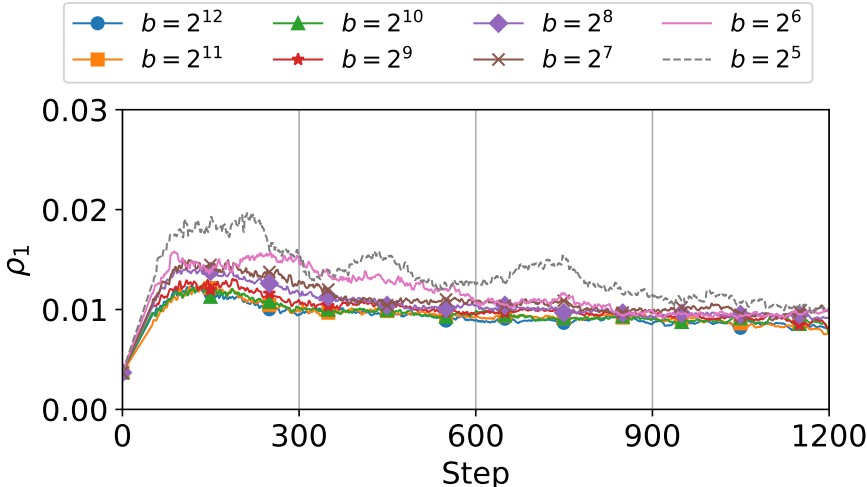

Figure 39:  $\rho_1$ is not a constant during training, but it does not change much at the EoS and shows a similar evolution for different batch sizes, especially when they are large.

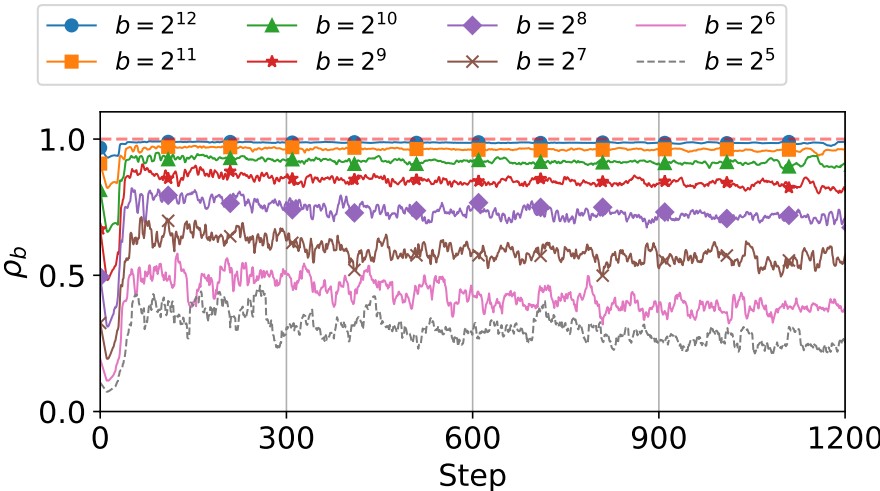

Figure 40:  $\rho_b$ is not a constant during training, but it does not change much at the EoS. $\rho_b$ is small for a small batch size.

