# OpenReview forum: "A new characterization of the edge of stability based on a sharpness measure aware of batch gradient distribution"
_ICLR.cc/2023/Conference — ICLR 2023 poster_

### Official Review · Reviewer_bH22 · 2022-10-17

**Confidence:** 3
**Correctness:** 3
**Technical Novelty And Significance:** 3
**Empirical Novelty And Significance:** 3
**Recommendation:** 6

**Clarity, Quality, Novelty And Reproducibility:**

**Clarity**: the paper is in general well written and easy to follow. (It is a bit awkward to first discuss SGD in Theorem 1 and then generalized momentum GD in Theorem 2, though)

**Quality and Novelty**: novel and significant enough to be published, from both theoretical and empirical perspectives.

**Reproducibility**: good

**Strength And Weaknesses:**

**Strength**: the paper is in general well written and the main ideas are easy to follow. The problem under study is important and the proposed theory provides novel insights and is of interest. The numerical experiments look compelling.

**Weaknesses**: the theoretical/technical depth of the paper is somewhat limited, the empirical contribution is rather significant, though. The limitations of the proposed theory/analysis are not sufficiently discussed.



**Summary Of The Paper:**

In this paper, the authors proposed a novel characterization of the "edge of stability (EoS)" phenomenon, referred to as the "interaction-aware-sharpness",  for both gradient descent (GD) and its stochastic variant SGD.
The proposed theory is of particular interest in its ability to provide novel and more precise insight into the EoS of SGD. Based on the theory, the authors further proposed a novel scaling rule (Linear and Saturation Scaling rule, LSSR) on the learning rate and SGD batch size, which extends the popular linear scaling rule (LSR).
Numerical experiments on a few (not so deep) neural networks and across a few widely used datasets were provided to empirically support the theoretical results.

**Summary Of The Review:**

The paper is in general well written and proposes an interesting and novel viewpoint of the EoS for SGD.
I do not see any major flaws in the paper.

Detailed comments:
* Definition 2: the term "concentration measure" is used without providing more context. This can be misleading to some readers with probability or statistics background who may be more familiar with the term "concentration of measure".
* It would be helpful to discuss the limitation of the proposed analysis (including the LSSR), from both theoretical and empirical perspectives possibly.
* can one go beyond SGD to more involved settings? It would be of interest to have a short discussion on this point.

---

> ### Author Response · Authors · 2022-11-13
> **Limitations and Future Work**
>
>
> > **[C1]** Definition 2: the term "concentration measure" is used without providing more context. This can be misleading to some readers with probability or statistics background who may be more familiar with the term "concentration of measure".
>
> **[A1]**
> - Thank you for the suggestion. We will change it to “a measure of concentration” of the batch gradient as it measures the concentration of the batch gradient.
>
> > **[C2]**  It would be helpful to discuss the limitation of the proposed analysis (including the LSSR), from both theoretical and empirical perspectives possibly.
>
> **[A2]**
> - We assume the progressive sharpening. It is still an open question why the progressive sharpening occurs in the neural network training.
> - We argued that the similar inductive bias leads to a similar generalization performance (from the scaling rule perspective), but we do not fully understand how the sharpness (or other measures) and the generalization performance correlate with each other to argue how to achieve a good generalization.
>
> > **[C3]** can one go beyond SGD to more involved settings? It would be of interest to have a short discussion on this point.
>
> **[A3]**
> - We’ve covered the generalized moment case (see Appendix C.6). Recently, Cohen et al. (2022) covered another adaptive gradient-based optimization algorithm such as Adam, AdamW and Rmsprop. As shown in their Figure 6, their experiments in the minibatch setting are also consistent with our theory (the smaller the batch size, the smaller the sharpness).
>
> **[References]**
> - Cohen et al. (2022), Adaptive Graident method at the Edge of Stability (arxiv:2207.14484)

---

### Official Review · Reviewer_9x87 · 2022-10-19

**Confidence:** 4
**Correctness:** 2
**Technical Novelty And Significance:** 3
**Empirical Novelty And Significance:** 3
**Recommendation:** 6

**Clarity, Quality, Novelty And Reproducibility:**

The paper is hard to read and the material is not well-organized. The analysis is interesting but not well explained

**Strength And Weaknesses:**

### Strong point

The derivation of new characterization is simple but powerful. The new sharpness also seems to capture the EoS phenomenon more precisely. The application of explaining the learning rate rescaling in large-batch training is consistent with empirical observations, which might be practically relevant. In general, I think this paper makes some interesting observations and I believe they are useful for further study on relevant topics.


----
### Weak point

However, I feel that this paper is written in a hurry and the work is still somewhat preliminary at this stage. Listed below are a few concerns in my mind.

1. **The discussions of prior works are unfair**

   The phenomenon of  EoS was first observed by (Wu et al., 2018) at the end of training; (Jastrze ̨bski et al., 2020) further observed the progressive sharpening phenomenon in training neural nets. (Cohen et al., 2021) followed these works by providing a systematical investigation of these phenomena and also additionally found the non-monotonic convergence at EoS.  All three works are crucially important in this line of research. Only attributing the discovery of EoS to (Cohen et al., 2021) is unfair (at least in my opinion).

2.  **The explanation of implicit interaction regularization is quite hand-waving and not convincing**
   - First, the definition of $\rho_b$ (a concentration measure of batch gradient)  is the same as the gradient diversity defined in (Yin et al., 2018)  and is never mentioned.
   - I find equation  (16) misleading since both sides depend on where the iterate locates. In iterations, both sides keep changing. The statement that *$\|H\|_{S_b}$ is implicitly regularized and bounded to be less than $2\rho_b/\eta$* is technically correct. But it implies nothing since both sides keep changing. For example, we cannot claim that SGD tends to find solutions with smaller $\|H\|_{S_b}$ since $\rho_b$ is not a constant.

 	3. In Section 5, I do not understand why keeping $2\rho_b/\eta$ to be constant is a good choice for generalization. This is never explained.



### Other comments

1.  In the second paragraph, the authors ask three questions. But it seems that this paper only answers the third one. Therefore, claiming "we provide a new characterization of the EoS for SGD, which can serve as an answer to the above questions" is not appropriate.

2.  In Section 2, I suggest focusing on the case of sampling with replacement for simplicity. Otherwise, the statement is much more complicated and no new insights are brought.

3.  In Section 3, the authors claim that existing works using tr(HS_b) make some additional assumptions, but they do not. In my opinion, this claim is quite misleading, since this paper also needs to make some assumptions, e.g., the validity of local quadratic approximation. In my opinion, the real difference is that this paper uses this quantity to analyze EoS, which is not carried out previously.

4.  In Figure 1, I do not understand why the authors only mark the endpoint with 'x' for cross-entropy loss. Also, it should be clarified in the caption the meaning of MSE and CE.

5. In Theorem 1, the batch gradient never follows a normal distribution, thus the Gaussian assumption is too strong. Also, I do not understand why  (8) and (9) are interesting.

6. In the definition of the new edge (11), I think the author should explain the intuitive difference between $||H||_{S_n}$ and $||H||$. In my understanding, the former considers the curvatures in typical directions but the latter only considers the curvature in the sharpness direction:

$$
\qquad ||H||_{S_n} = \frac{\theta^TH^2\theta}{\theta^T H \theta}, \qquad ||H|| = \sup \frac{\theta^TH^2\theta}{\theta^T H \theta},
$$
where the supremum is taken wrt the direction theta.

8. In Figure 2, what do you mean by **overestimation**? In addition, are the "iteratre enter the EoS" and "in the unstable region" the same thing?

9. In Equations (13) and (14) and Theorem 2, suddenly momentum GD is studied. I do not understand why? Do the authors want to say the EoS is the same as vanilla (S)GD? Theorem 2 is also repeatedly referred to argue the convergence of directions, but it is stated for momentum GD.  In addition, in Theorem 2, what is $\lambda_i$? What do you mean by quadratic $L$? Is each sample loss also quadratic?




---
### Reference

[1] Yin et al., Gradient diversity: a key ingredient for scalable distributed learning

[2] Wu et al., How SGD selects the global minima in over-parameterized learning: A dynamical stability perspective

[3] Jastrzebski et al.,The break-even point on optimization trajectories of deep neural networks

**Summary Of The Paper:**

This paper provides a fine-grained analysis of the edge of stability (EoS) phenomenon and in particular, their characterization also applies to SGD. In my understanding, the new characterization not only uses the largest eigenvalue of Hessian, which corresponds to the sharpest direction but also considers the curvature along other directions. To this end, an interaction-aware sharpness is defined. The numerical experiments show that the characterization is quite "accurate" and the authors provide an interesting application in explaining the linear and saturating scaling rule in large-batch training.

**Summary Of The Review:**

The more precise characterization of EoS and the empirical findings are interesting, but this work is still somewhat preliminary and more work is needed.


=====
The author's response partially addressed my concern. I generally think that the analysis and numerical experiments are useful for future study on relevant topics. Hence, I increased my score to 6.

---

> ### Author Response · Authors · 2022-11-13
> **More discussions of prior work**
>
> > **[W1]** The phenomenon of EoS was first observed by (Wu et al., 2018) at the end of training; (Jastrzebski et al., 2020) further observed the progressive sharpening phenomenon in training neural nets. (Cohen et al., 2021) followed these works by providing a systematical investigation of these phenomena and also additionally found the non-monotonic convergence at EoS. All three works are crucially important in this line of research. Only attributing the discovery of EoS to (Cohen et al., 2021) is unfair (at least in my opinion).
>
> **[A1]**
> - We thank the reviewer for the suggestions. It will be of great help to improve the manuscript. We agree with the reviewer that the suggested references are crucially important in this line of research. We modified some of the sentences with the suggested references.
>
> > **[W2-1]** First, the definition of $\rho_b$ (a concentration measure of batch gradient) is the same as the gradient diversity defined in (Yin et al., 2018) and is never mentioned.
>
> **[A2-1]**
> - We thank the reviewer for pointing that out. We would like to clarify that they are not exactly the same.
> - The gradient diversity is defined as:
> $$\text{Gradient Diversity}\equiv\frac{\sum\_i ||\nabla \ell(x\_i;w)||^2 }{||\sum\_i \nabla \ell(x\_i;w)||^2}=\frac{\sum\_i||\nabla \ell(x\_i;w)||^2 }{||n \nabla L||^2}=\frac{\sum\_i ||\nabla \ell(x\_i;w)||^2 }{n^2||\nabla L||^2}=\frac{\mathbb{E}\_x[||\nabla \ell(x;w)||^2]}{n||\nabla L||^2}$$
> $$=\frac{1}{n}\frac{\text{tr}(S\_1)}{\text{tr}(S\_n)}$$
> while $$\rho\_b\equiv\frac{||\nabla L||^2}{\mathbb{E}\_{\xi}[||g\_\xi||^2]}=\frac{\text{tr}(S\_n)}{\text{tr}(S\_b)}$$
> where
> $g_\xi=\frac{1}{b}\sum\_{x\in\mathcal{B}\_\xi}\nabla_\theta \ell(x;\theta)$.
> - Therefore, we added the following sentence: “Yin et al. (2018) call $1/(n\rho_1)$ the gradient diversity.” in the revised version.

---

> > ### Comment · Reviewer_9x87 · 2022-11-18
> > **Disscussion on prior works are not accurate**
> >
> > Thanks for the response. I read the revisioned paper but still find the discussion of prior works is not accurate.
> >  - Jastrzebski et al's work observes the progressive sharpening but do not show that sharpness can hover above/around 2/eta. The hovering over 2/eta should be attributed to [Cohen et al., 2021].
> >  - [Wu et al. 2018] empirically find that the sharpness of solutions found by GD with large LR is almost  equal to 2/eta, suggesting the EoS near the end of training (See table 2 of [Wu et al. 2018]). This fact is not mentioned in  the revisoned paper.
> >  - About [Yin et al., 2018], I would say your definition is just a mini-batch version of gradient diversity defined in [Yin et al., 2018].  But of course, this paper studies a problem, rather different from [Yin et al., 2018]. I would like to suggest that mention this point, i.e., studying different problems.  I also feel that calling this quantity "a concentration measure of the batch gradient" seems strange to me.

---

> > > ### Author Response · Authors · 2022-11-21
> > > **More discussions of prior work (2)**
> > >
> > > Thank you for the follow up comments. Some previous comments become clearer. We would like to discuss more and fix weaknesses so that we can improve our paper.
> > >
> > > > **[W1-a]** Jastrzebski et al's work observes the progressive sharpening but do not show that sharpness can hover above/around 2/eta. The hovering over 2/eta should be attributed to [Cohen et al., 2021].
> > >
> > > **[A1-a]**
> > >
> > > To avoid the confusion, we will change "For full-batch GD, it has been empirically observed that the sharpness, the top eigenvalue of the Hessian, **increases** and **then hovers above 2/(learning rate)** (Jastrz˛ebski et al., 2019; 2020; Cohen et al., 2021) as the training proceeds." to the following: "... **increases**  (Jastrz˛ebski et al., 2019; 2020) and **then hovers above 2/(learning rate)** (Cohen et al., 2021) ..."
> > >
> > > > **[W1-b]** [Wu et al. 2018] empirically find that the sharpness of solutions found by GD with large LR is almost equal to 2/eta, suggesting the EoS near the end of training (See table 2 of [Wu et al. 2018]). This fact is not mentioned in the revisoned paper.
> > >
> > > **[A1-b]**
> > > Thank you for the suggestion. We would like to discuss more about the reference.
> > > There are some differences between [Wu et al. 2018] and our paper.
> > > We are going to discuss about the following two points:
> > > **First**, [Wu et al. 2018] studied the sharpness of **solutions in the convergence limit (near the end of training)**, while we studied the sharpness (and the IAS) **at the EoS**.
> > > **Second**, [Wu et al. 2018] mainly focused on the **MSE loss**.
> > > Note that the sharpness goes to 0 in the convergence limit in the case of the cross entropy (see [this answer](https://openreview.net/forum?id=bH-kCY6LdKg&noteId=qkR3ai9rCRi)).
> > > [Wu et al. 2018] said in Sec 3.2 that
> > > > (...) the issue of asymptotic convergence to a particular critical point is only related to the local stability of that critical point, and locally, one can always make the linear approximation for the dynamical system or quadratic approximation for the objective function to be optimized, as long as the the linear or quadratic approximations are **non-degenerate**. Therefore our findings are of general relevance, even for problems for which the loss function is non-convex, as long as the non-degeneracy holds.
> > > However, (...) the quadratic approximation (...) is **not** suited for (...) the case (...) the **cross entropy** used as the loss function. In this paper, we will focus on the case when the quadratic approximation is locally valid. Therefore in the following experiments, we use the mean squared error rather than the cross entropy as the loss function.
> > >
> > > Therefore, their Table 2 is **only for the MSE loss**.
> > > As shown in their Figure 4, they said they do not consider the cross entropy because the Hessian vanishes **at the minima**.
> > > However, we consider the sharpness and the IAS at the EoS, not at the minima.
> > > At the EoS, the quadratic approximation is also suited for the cross entropy (the non-degeneracy holds).
> > >
> > > > **[W1-c]** About [Yin et al., 2018], I would say your definition is just a mini-batch version of gradient diversity defined in [Yin et al., 2018]. But of course, this paper studies a problem, rather different from [Yin et al., 2018]. I would like to suggest that mention this point, i.e., studying different problems. I also feel that calling this quantity "a concentration measure of the batch gradient" seems strange to me.
> > >
> > > **[A1-c]**
> > > Yin et al. (2018) studied the generalization for a restricted setting of **(strongly) convex loss function** with the **differential** gradient diversity from the algorithmic stability perspective, and also studied the convergence speed saturation with the gradient diversity.
> > > Here, the gradient diversity is $\Delta\_D =\frac{1}{n\rho\_1}$ and the batch-size bound is $B\_D\equiv n\Delta\_D=\frac{1}{\rho\_1}$.
> > > We used the term "concentration measure" because it measures the concentration of the batch gradient in the similar sense with the mean resultant length (MRL) $\bar R\_b^2$ and the concentration parameter of vMF (von Mises–Fisher) distribution (cf. Excercise 2.55 in Bishop's PRML).
> > > For example, MRL is long when the batch gradient (direction) is concentrated as shown in Figure 4 of [the illustration of the MRL](https://www.frontiersin.org/articles/10.3389/fpsyg.2018.02040/full).
> > > As also suggested in [another review](https://openreview.net/forum?id=bH-kCY6LdKg&noteId=zfxLPQbpTT) **[C1]**, we will change it to “a measure of concentration” of the batch gradient.

---

> ### Author Response · Authors · 2022-11-13
> **On Eq (16) ($\rho_b$ does not change much at the EoS.)**
>
>
> > **[W2-2]**  I find equation (16) misleading since both sides depend on where the iterate locates. In iterations, both sides keep changing. The statement that $||H||\_{S\_b}$ is implicitly regularized and bounded to be less than $2\rho\_b/\eta$ is technically correct. But it implies nothing since both sides keep changing. For example, we cannot claim that SGD tends to find solutions with smaller $||H||\_{S\_b}$ since $\rho\_b$ is not a constant.
>
> **[A2-2]**
> - We thank the reviewer for the valuable feedback. The original manuscript lacked explanation on the part pointed out by the reviewer. We think this is the most critical weak point among **[A1]-[A3]**.
> - First of all, **Figure 33 and Figure 34 (left)** in Appendix (Please kindly check these Figures. We will move this to the main text in the later version) showed that $||H||\_{S\_b}$ is implicitly regularized and bounded to be less than some threshold (let’s say $t(b)$ for each $b$) and this threshold $t(b)$ is lower for a smaller batch size $b$. We also note that $t(b)$ saturates as $b$ becomes larger. And together with Figure 4 (d), we can conclude that $t(b)\approx\frac{2\rho\_b}{\eta}$.
> - From Figure 4(d), Figure 33 and Figure 34 (left), we can expect that **$\rho\_b$ does not change much at the EOS** since the threshold $t(b)$ does not change much at the EoS in Figure 33 and Figure 34 (left).
> - To make this point clearer, we provided **Figure 41 (revision #1)** which shows that, at the EOS, $\rho_b$ does not change much. As shown in Figure 40 (revision #1), $\rho_1$ slightly decreases, and thus, when $b$ is small, $\rho_b\approx \rho_1b$ also slightly decreases but does not change much. Similar patters are also observed in Figure 6 in Cohen et al. (2022).
> We will move Figure 41 (revision #1) to the main text in the later version.
> - At least, it shows that the thresholds $\frac{2\rho\_b}{\eta}$ for $||H||\_{S\_b}$ are well distinguished from each other according to $b$, i.e., the smaller the batch size, the lower the threshold.
> - We can claim that SGD (with a smaller $b$) tends to find solutions with smaller $||H||\_{S\_b}$.
>
> **[References]**
> - Cohen et al. (2022), Adaptive Graident method at the Edge of Stability (arxiv:2207.14484)

---

> > ### Comment · Reviewer_9x87 · 2022-11-18
> > **Thanks for new results**
> >
> > Very interesting result.

---

> ### Author Response · Authors · 2022-11-13
> **inductive bias of SGD characterized by $(\eta,b)$ pair**
>
> > **[W3]**
> In Section 5, I do not understand why keeping $2\rho_b/\eta$ to be constant is a good choice for generalization. This is never explained.
>
> **[A3]**
> - We did not argue that keeping $\frac{2\rho\_b}{\eta}$ to be constant is good for generalization. (It may restrict the hypothesis space, but this is not our focus.)
> - We claimed that the pairs $(b,\eta)$ and $(b’,\eta’)$ with the same $\frac{2\rho\_b}{\eta}=\frac{2\rho\_{b’}}{\eta’}$ lead to **a similar inductive bias** ($\text{IAS}\approx \frac{2\rho\_b}{\eta}=\frac{2\rho\_{b'}}{\eta'}$) of SGD and to a similar generalization performance. We call this the Linear and Saturation Scaling Rule (LSSR).
> - We only argued that the similar inductive bias leads to a similar generalization performance (from the scaling rule perspective), and did not argue any of other effects such as which value of $\frac{2\rho\_b}{\eta}$ leads to a good generalization.
> - For an overparameterized model (e.g., DNN), the (training) loss minimization has multiple global minima and thus different algorithms may converge to different global minima with different generalization performance. Therefore, it is important to understand the above “inductive bias” ($||H||\_{S\_b}\approx\frac{2\rho\_b}{\eta}$) of the optimization algorithm toward different global minima from the generalization perspective.
> - Please correct us if our understanding on **[W3]** is wrong.

---

> > ### Comment · Reviewer_9x87 · 2022-11-18
> > **Still feel confused**
> >
> > Thanks for the response.  Indeed, by your explanation, keeping $2\rho_b/eta$ the same  leads to a similar IAS inductive bias. But if IAS is not relevant for generalization, why do we should keep it the same? I understood that this choice leads to the LSSR, which is consistent with practical observation. In the heatmaps of Figure 5,  LSSR is observed for the best test performances.

---

> > > ### Author Response · Authors · 2022-11-22
> > > **Optimization within a constraint set characterized by the IAS**
> > >
> > > **[W3-a]**
> > > > But if IAS is not relevant for generalization, why do we should keep it the same?
> > >
> > > **[A3-a]**
> > > The question may be "why do two algorithms with similar IAS at the EoS have a similar generalization?"
> > > (We may misunderstand the reviewer's point. Please correct us if our understanding is wrong.)
> > >
> > > For an overparameterized model (e.g., DNN), the (training) loss minimization has multiple global minima and thus different algorithms may converge to different global minima with different generalization performance. Therefore, it is important to understand the “inductive bias”.
> > >
> > > We've shown that SGD with $(b,\eta)$ has a trajectory where the IAS is around $\frac{2\rho\_b}{\eta}$, i.e., $\theta\_t \in\Theta^\ast(b,\eta)\equiv${$\theta: \text{IAS}(\theta)\approx \frac{2\rho\_b}{\eta} $} at the EoS.
> > > Therefore, the inductive bias of SGD can be expressed with the following constrained optimization problem:
> > > $$\min L(\theta)\ \text{s.t.}\ \theta\in\Theta^\ast(b,\eta)$$
> > > and thus the iterate converges to a certain set of the global minima determined by the constraint set $\Theta^\ast(b,\eta)$.
> > > It would be more accurate to say that the IAS is relevant for generalization in the sense that the model parameter is optimized within a constraint set (feasible domain) $\Theta^\ast(b,\eta)$ characterized by the IAS value at the EoS.
> > > Note that $\Theta^\ast(b,\eta)\approx \Theta^\ast(b',\eta')$ holds when $\rho\_b/\eta=\rho\_{b'}/\eta'$.
> > > We think this bias (IIR) is fundamental because it is determined only by the optimization algorithm (regardless of the network architecture).

---

> ### Author Response · Authors · 2022-11-16
> **About the comments (Part 1/3)**
>
> > **[C1]** In the second paragraph, the authors ask three questions. But it seems that this paper only answers the third one. Therefore, claiming "we provide a new characterization of the EoS for SGD, which can serve as an answer to the above questions" is not appropriate.
>
> **[A1]**
> - $||H||$: the sharpness, $||H||\_{S\_b}$: the IAS (interaction-aware sharpness).
> - We ask the three questions: (i) why and (ii) to what extent the sharpness hover above $2/\eta$ (GD) and (iii) how can this be generalized to minibatch SGD. We answer these with the new characterization of the EoS that $||H||_{S_b}\approx\frac{2\rho_b}{\eta}$ at EoS ($1\leq b\leq n$) which explains the both cases of $b=n$ (GD) and $b<n$ (SGD). To answer (i) and (ii), we consider GD ($b=n$).
> - We’ve shown that the IAS is around $2/\eta$ at the EoS and the sharpness $||H||$ is larger than the IAS (by definition), which answers how much the sharpness is larger than $2/\eta$ at the EoS. It is equivalent to answer how much the IAS is smaller than the sharpness at the EoS.
> - Explicitly, the IAS is smaller than the sharpness with the following relation:
> $$\frac{||H||\_{S\_n}}{||H||} = \frac{\text{tr}(HS\_n)}{||H||\text{tr}(S\_n)}= \frac{\nabla L ^\top H \nabla L}{||H||||\nabla L||^2} \leq 1$$
> where the equality holds iff $\nabla L$ is aligned with the top eigenvector $q_1$ of $H$.
> - Let $c_i$ denote the cosine similarity between $\nabla L$ and the i-th normalized eigenvector $q_i$ (for the i-th largest eigenvalue $\lambda\_i$) of $H$ ($|c_i|\leq 1$ and $\sum\_i c\_i^2=1$). Then, since $H=\sum_i \lambda_i q_iq_i^\top$ and $\nabla L ^\top H \nabla L = \sum_i \lambda_i c_i^2 ||\nabla L||^2$, we have $||H||$ around $\frac{2}{\eta a}$ at the EoS
>  where $$a=\sum_i \frac{\lambda\_i}{\lambda\_1}c\_i^2\leq\sum_ic\_i^2=1.$$
> Therefore, $a$ quantitatively measures how much $2/\eta$ is smaller than the sharpness, i.e., $2/\eta$ is about ${a}$ times smaller than the sharpness at the EoS.
> - Roughly speaking, the alignment between $\nabla L$ and the subspace spanned by the top eigenvectors of $H$ (together with the relative eigenvalues $\frac{\lambda_i}{\lambda\_1}$) determines how much $2/\eta$ is smaller than the sharpness at the EoS.
> - Please kindly check Figure 23-25, which show that the IAS is slightly smaller than the sharpness when $|c_1|=|\cos(q_1,\nabla L)|$ is close to 1. We also note that the sharpness sometimes oscillates around $2/\eta$, especially for MSE loss (e.g., Fig 29), but this does not violate our claim. This may happen when $|c_1|$ is close to 1.
>
> > **[C2]** In Section 2, I suggest focusing on the case of sampling with replacement for simplicity. Otherwise, the statement is much more complicated and no new insights are brought.
>
> **[A2]**
> - The **typical** SGD is more like the case of sampling **without** replacement, i.e., SGD usually does not use the same sample more than once in a single epoch.
> - More importantly, when $b$ is large, the “with” and “without” methods have very different $\gamma_{n,b}$. For example, $\gamma_{n,n}=1$ ($\gamma_{n,n}=0$) for the “with” ("without") method, respectively. However, we found that they still have similar $\rho_b$ even when $b$ is large and $\gamma_{n,b}$ is different (because $\rho_b$ is a function of $\frac{\gamma_{n,b}}{b}$ as shown in (*) in Eq (17)). It may seem there is no insight because the results (IIR, LSSR) from the two methods are similar in the end.
> - For sampling without replacement, many previous works approximate $\gamma_{n,b}\approx 1$ assuming $b\ll n$, but we consider the whole range of $1\leq b\leq n$ ($0\leq \gamma_{n,b}\leq 1$ with $\gamma_{n,1}=1$ and $\gamma_{n,n}=0$).
>
> > **[C3]** In Section 3, the authors claim that existing works using $\text{tr}(HS\_b)$ make some additional assumptions, but they do not. In my opinion, this claim is quite misleading, since this paper also needs to make some assumptions, e.g., the validity of local quadratic approximation. In my opinion, the real difference is that this paper uses this quantity to analyze EoS, which is not carried out previously.
>
> **[A3]**
> - If the loss function is smooth enough, then we can always find a neighborhood in which the loss can be approximated with a quadratic function. This is not a strong assumption. Empirically, we observed this quadratic approximation holds locally as shown in Fig 1 when the stepsize $||\delta_t||$ is small enough with respect to the loss landscape curvature. On the other hand, Zhu et al. (2019); Yaida (2019); Thomas et al. (2020) made much stronger assumptions (e.g., strong convexity).

---

> ### Author Response · Authors · 2022-11-16
> **About the comments (Part 2/3)**
>
> > **[C4]** In Figure 1, I do not understand why the authors only mark the endpoint with 'x' for cross-entropy loss. Also, it should be clarified in the caption the meaning of MSE and CE.
>
> **[A4]**
> - Thank you for the suggestion. We clarified with MSE=mean squared error, CE=cross entropy in the revised version.
> - The endpoint ('x') indicates that the iterate enters the unstable region for CE+GD because it is clear when exactly the iterate enters the unstable region.
> But for MSE+GD (Fig 1), we plot the graph for a few more steps after the iterate enters the unstable region (i.e., it enters the EoS) because it is not very important when exactly the iterate enters the unstable region and the iterate stays very close to the EoS in this case (see Fig 2).
>
> > **[C5]** In Theorem 1, the batch gradient never follows a normal distribution, thus the Gaussian assumption is too strong. Also, I do not understand why (8) and (9) are interesting.
>
> **[A5]**
> - In Theorem 1 (fisrt part), without the normal assumption, we proved **the expected loss** increases iff the iterate $\theta_t$ is in the unstable region $\mathbb{U}$.
>  So if we consider a subregion $\mathbb{V}\subset\mathbb{U}$ with a finite maximum loss ($\max_{\theta\in\mathbb{V}}L(\theta)\leq M<\infty$) near and including $\theta_t$ ($\mathbb{B}(\theta_t,r)\subset \mathbb{V}$ for some radius $r>0$) in which the loss is approximately quadratic, then the iterate **tends to** increase the loss by Theorem 1 (first part), so eventually it is repelled from $\mathbb{V}$.
> The probability of the loss to increase is as follows:
> $$\mathbb{P}(L_{t+1}>L_t|\theta_t)=\mathbb{P}(L_{t+1}-\mathbb{E}[L_{t+1}]>L_{t}-\mathbb{E}[L_{t+1}] |\theta_t).$$
> And, from the first part, we have $L_{t}-\mathbb{E}[L_{t+1}]=-a$ for some $a>0$.
> Therefore, together with (9) in the second part of Theorem 1 (with $x=\frac{a^2}{2\beta}$), we proved that the probability of the loss to increase is higher than $1-\exp(-\frac{a^2}{2\beta})$. Actually we don't require the batch gradient to be normally distributed to concentrate the $L_{t+1}$ near the mean value $\mathbb{E}\_\xi[L\_{t+1}]$.
> The exact formulations of (8) and (9) are not important, but it is important that the $L_{t+1}$ is concentrated near the expected loss $\mathbb{E}\_\xi[L\_{t+1}]$ in some sense. We just used the normal assumption to explicitly formulate this concentration. We do understand this is a false assumption, but many studies often assume the normal assumption for the SDE analysis (Mandt et al., 2016;2017; Hu et al., 2019, Li et al., 2017;2019a; Jastrzebski et al., 2017; Smith & Le, 2018; Chaudhari & Soatto, 2018) (see Section D for these references).
>
> > **[C6]** In the definition of the new edge (11), I think the author should explain the intuitive difference between $||H||\_{S\_n}$ and $||H||$. In my understanding, the former considers the curvatures in typical directions but the latter only considers the curvature in the sharpness direction:
> $$||H||\_{S\_n}=\frac{\theta^\top H^2\theta}{\theta^\top H\theta},||H||=\sup \frac{\theta^\top H^2\theta}{\theta^\top H\theta},$$
> where the supremum is taken wrt the direction theta.
>
> **[A6]**
> - There is a misconception in the reviewer's understanding. The correct (intuitive) equation is:
> $$||H||\_{S\_n}=\frac{\nabla L^\top H \nabla L}{\nabla L^\top \nabla L}$$
> $$||H||=\sup\_{\|v\|=1} v^\top H v=\sup_{v\neq 0} \frac{v^\top H v}{v^\top v}$$
> - Here, we used $v$ (not $\theta$) to avoid confusion with the model parameter $\theta$. Note that there is no $H$ in the denominators.
> - We will add the above intuitive definitions in the main text.
>
> > **[C7]** In Figure 2, what do you mean by overestimation? In addition, are the "iteratre enter the EoS" and "in the unstable region" the same thing?
>
> **[A7]**
> - In the unstable region, the normalized increase in the loss $\left|\frac{\mathbb{E}\_\xi[L_{t+1}]-L_t}{\text{tr}(S\_n)}\right|$ is often smaller than our expectation $\frac{\eta^2}{2}\left|\frac{\text{tr}(HS\_b)}{\text{tr}(S\_n)}-\frac{2}{\eta}\right|$ from the quadratic assumption. In other words, in the unstable region, the quadratic assumption **overestimates** the loss increase since the sharpness decreases along the gradient descent direction. In the unstable region, the stepsize $||\delta_t||$ became too large for the local quadratic approximation.
> - If the iterate is in the stable region, then the sharpness increases and the iterate enters the unstable region (the progressive sharpening and Theorem 2). Also if the iterate enters the unstable region, then it tends to be repelled from the unstable region (Theorem 1). Therefore, the iterate operates near at the EoS, going back and forth between the stable and unstable regions.

---

> ### Author Response · Authors · 2022-11-16
> **About the comments (Part 3/3)**
>
> > **[C8]** In Equations (13) and (14) and Theorem 2, suddenly momentum GD is studied. I do not understand why? Do the authors want to say the EoS is the same as vanilla (S)GD? Theorem 2 is also repeatedly referred to argue the convergence of directions, but it is stated for momentum GD. In addition, in Theorem 2, what is λi? What do you mean by quadratic L? Is each sample loss also quadratic?
>
> **[A8]**
> - The generalized momentum GD in (13) and (14) reduces to the vanilla GD when $(\beta_1,\beta_2)=(0,0)$. Theorem 2 is general and applicable to the vanilla GD.
> - $\lambda_i$ is the i th largest eigenvalue of $H$
> - $L(\theta)$ is quadratic with respect to $\theta$. Each sample loss $\ell(x;\theta)$ does not need to be quadratic with respect to $\theta$.
> - However, we emphasize that our theory does not consider the globally quadratic loss. Instead, we consider a subregion in which the loss is approximately quadratic. Our theory leads to the statement: if we consider a bounded subregion $\mathbb{V}\subset\mathbb{U}$ with a finite maximum loss near and including $\theta_t$ in which the loss is approximately quadratic, then the iterate tends to escape from the region $\mathbb{V}$.
> - In other words, if the loss $L(\theta)$ is globally quadratic, then the iterate diverges when the sharpness exceeds the stability threshold. But, if the loss $L(\theta)$ is locally quadratic, then the iterate escape the region in which the loss is approximately quadratic.

---

### Official Review · Reviewer_mqQV · 2022-10-21

**Confidence:** 3
**Correctness:** 2
**Technical Novelty And Significance:** 2
**Empirical Novelty And Significance:** 3
**Recommendation:** 6

**Clarity, Quality, Novelty And Reproducibility:**

Empirical results are of good quality and originality, although related work from 2021 on might have been missed. Clarity overall is okay, but I recommend the authors to itemize and specify their main findings and contributions in a concise but quantitative fashion.

**Strength And Weaknesses:**

Strengths:

(1) Problems studied in this paper are very interesting.

(2) Empirical investigations are detailed.

Weaknesses:

The demonstration is still largely empirical, and I wish the theoretical component could be stronger. Some reasons are:

(1) some of the claims appear to be rather general, but it is actually known that EoS does not occur for all loss functions. I’d appreciate it very much if the authors could be more specific about their conjectures (which I nevertheless found interesting), such as the hypothesized conditions under which they could be true.

(2) The abstract suggests the question of “why the sharpness is somewhat larger than 2/(learning rate)” will be answered. Could the answer be more explicitly and concisely provided? Is there any quantification of how much larger it is than 2/(learning rate)? Is it loss function dependent? In fact, I wonder how the authors relate this claim (and other empirical observations) to the recent line of analytical work based on matrix factorization, e.g., [Wang et al. Large Learning Rate Tames Homogeneity. ICLR 22], [Zhu et al. Understanding Edge-of-Stability Training Dynamics with a Minimalist Example. arXiv]. The latter is concurrent so it is not a missing reference, but I’m curious if there is any contradiction: for example, these results seem to give sharpness not larger than 2/(learning rate). Although the authors seem unaware, theories in these papers do seem to be aligned with many of the empirical observations made here. However, do they align with all observations? It could be that neither is wrong, and instead it is just a matter of different loss functions. I wish these aspects could be explored more.

**Summary Of The Paper:**

This paper aims at gaining more understanding of the Edge of Stability phenomenon. Some theoretical understanding was obtained via Theorem 1, which assumes a quadratic loss function. Although EoS does not manifest for quadratic loss, this theorem was then used to motivate empirical claims. Such claims are backed by experiments.

**Summary Of The Review:**

Overall, this is an interesting paper and a helpful contribution toward an understanding of EoS. I have some concerns at this moment, but my score can be changed.

---

> ### Author Response · Authors · 2022-11-13
> **Concise and quantitative analysis on how much the sharpness is larger than $2/\eta$**
>
> > **[W2]** The abstract suggests the question of “why the sharpness is somewhat larger than 2/(learning rate)” will be answered. Could the answer be more explicitly and concisely provided? Is there any quantification of how much larger it is than 2/(learning rate)? Is it loss function dependent?
>
> **[A2]**
> - $||H||$: the sharpness, $||H||\_{S\_b}$: the IAS (interaction-aware sharpness). In this answer, we only consider GD ($b=n$).
> - We’ve shown that the IAS is around $2/\eta$ at the EoS and the sharpness $||H||$ is larger than the IAS (by definition), which answers how much the sharpness is larger than $2/\eta$ at the EoS. It is equivalent to answer how much the IAS is smaller than the sharpness at the EoS.
> - Explicitly, the IAS is smaller than the sharpness with the following relation:
> $$\frac{||H||\_{S\_n}}{||H||} = \frac{\text{tr}(HS\_n)}{||H||\text{tr}(S\_n)}= \frac{\nabla L ^\top H \nabla L}{||H||||\nabla L||^2} \leq 1$$
> where the equality holds iff $\nabla L$ is aligned with the top eigenvector $q_1$ of $H$.
> - Let $c_i$ denote the cosine similarity between $\nabla L$ and the i-th normalized eigenvector $q_i$ (for the i-th largest eigenvalue $\lambda\_i$) of $H$ ($|c_i|\leq 1$ and $\sum\_i c\_i^2=1$). Then, since $H=\sum_i \lambda_i q_iq_i^\top$ and $\nabla L ^\top H \nabla L = \sum_i \lambda_i c_i^2 ||\nabla L||^2$, we have $||H||$ around $\frac{2}{\eta a}$ at the EoS
>  where $$a=\sum_i \frac{\lambda\_i}{\lambda\_1}c\_i^2\leq\sum_ic\_i^2=1.$$
> Therefore, $a$ quantitatively measures how much $2/\eta$ is smaller than the sharpness, i.e., $2/\eta$ is about ${a}$ times smaller than the sharpness at the EoS.
> - Roughly speaking, the alignment between $\nabla L$ and the subspace spanned by the top eigenvectors of $H$ (together with the relative eigenvalues $\frac{\lambda_i}{\lambda\_1}$) determines how much $2/\eta$ is smaller than the sharpness at the EoS.
> - Please kindly check Figure 23-25, which show that the IAS is slightly smaller than the sharpness when $|c_1|=|\cos(q_1,\nabla L)|$ is close to 1. We also note that the sharpness sometimes oscillates around $2/\eta$, especially for MSE loss (e.g., Fig 29), but this does not violate our claim. This may happen when $|c_1|$ is close to 1.

---

> ### Author Response · Authors · 2022-11-13
> **More specific hypothesized conditions about the conjectures**
>
> > **[W1]** some of the claims appear to be rather general, but it is actually known that EoS does not occur for all loss functions. I’d appreciate it very much if the authors could be more specific about their conjectures (which I nevertheless found interesting), such as the hypothesized conditions under which they could be true.
>
> > **[W2-2]** In fact, I wonder how the authors relate this claim (and other empirical observations) to the recent line of analytical work based on matrix factorization, e.g., [Wang et al; Zhu et al.]. (...) I’m curious if there is any contradiction: for example, these results seem to give sharpness not larger than 2/(learning rate). (...) theories in these papers do seem to be aligned with many of the empirical observations made here. However, do they align with all observations? It could be that neither is wrong, and instead it is just a matter of different loss functions. I wish these aspects could be explored more.
>
> **[A1/A2-2]**
> - We thank the reviewer for the references.
> - We would like to emphasize two critical differences between our paper and [Wang et al.; Zhu et al.].
> - **First**, we **assume** (a hypothesized condition) that the loss function $L$ exhibits **the progressive sharpening (PS)**.
> - So there is no contradiction between our theory and  [Wang et al.; Zhu et al.]. As the reviewer mentioned, neither is wrong, and instead it is just a matter of different loss functions (whether the loss function exhibits the PS or not), but we assume the PS because it is a common phenomenon in neural network training. We also note that, a concurrent work (Damian et al., 2022) also assumes the progressive sharpening and another concurrent work (Li et al., 2022) proved the progressive sharpening under a certain restricted condition (e.g., two-layer fully-connected linear neural networks + certain assumptions).
> - We first assume the PS, and thus the followings are observed in order: (i) the sharpness increases, (ii) the sharpness exceeds $2/\eta$, (iii-iv) followed by the increase in the IAS by Theorem 2, and (v) when the IAS exceeds the threshold $2/\eta$, if we consider a bounded subregion $\mathbb{V}\subset\mathbb{U}$ with a finite maximum loss near and including $\theta_t$ in which the loss is approximately quadratic, then the iterate tends to escape from the region $\mathbb{V}$ (by Theorem 1) and the IAS tends to decrease (Sec 4.2).
> - On the other hand, Zhu et al. consider some initializations at which the loss doesn’t exhibits the PS. Please kindly check Figure 2 and Figure 5 in Zhu et al.. This setting may seem more general than ours as it does not assume the PS, but the PS is common in the neural network training, and thus the scalar factorization (Sec 4.1 in Zhu et al.) or the 4 scalar minimalist example (Sec 2.1 in Zhu et al.) with these initializations may not well represent the usual neural network training.
> - In their Fig 5a, the global minima manifold is nearly orthogonal to the sharpness level set near the given initialization. In other words, the gradient descent direction (orthogonal to the global minima manifold) may not increase the sharpness even when the sharpness is lower than $2/\eta$.
> - In their Fig 2a, the global minima manifold is almost aligned with the sharpness level set near the given initialization. In other words, the gradient descent direction (orthogonal to the global minima manifold) tends to increase the sharpness when the sharpness is lower than $2/\eta$ and thus the sharpness in the convergence limit is close to $2/\eta$. But for other initialization points, this progressive sharpening may not occur. Thus, as shown in their Fig 2b, the sharpness in the convergence limit can be much lower than $2/\eta$.
> -  Also, in Wang et al., the dynamics does not exhibit the PS and $||x_k||^2+||y_k||^2$ has a decreasing trend as GD searches for flat minimum.
> - **Second**, [Wang et al.; Zhu et al.] studied the sharpness **in the convergence limit** (let’s call this the limit sharpness), while we studied the sharpness (and IAS) **at the EoS**.
> - For example, Wang et al. showed that the limit sharpness is $\leq 2/\eta$ and Zhu et al. showed some explicit examples that the limit sharpness is much lower than $2/\eta$ (e.g., Fig 1b in [Zhu et al.]).
> - If we use GD with a cross-entropy loss, then the sharpness goes below $2/\eta$ in the later phase after the EoS (see Fig 10c in Appendix C (Cross-entropy loss) of Cohen et al. (2021) / Fig 27 (bottom left) in our paper) since $H\approx J^\top \nabla^2_z \ell J$ and the spectral norm of $\nabla^2_z\ell=\text{diag}(p)-pp^\top$ goes to $0$ in the convergence limit where $J=\nabla_\theta z,p=\text{softmax}(z)$ and $z$ is the presoftmax logit output. In the end, the sharpness goes to zero (the limit sharpness $=0$), but this is not our focus.

---

> > ### Author Response · Authors · 2022-11-14
> > **References for the above comment**
> >
> > **[References]**
> > - Wang et al. (2022), Large Learning Rate Tames Homogeneity. ICLR 22
> > - Zhu et al., (2022), Understanding Edge-of-Stability Training Dynamics with a Minimalist Example (arxiv:2210.03294)
> > - Damian et al. (2022), Self-Stabilization: The Implicit Bias of Gradient Descent at the Edge of Stability (arxiv:2209.15594)
> > - Li et al. (2022), Analyzing Sharpness along GD Trajectory: Progressive Sharpening and Edge of Stability (arxiv:2207.12678)
> > - Cohen et al. (2021), Gradient descent on neural networks typically occurs at the edge of stability. ICLR 21

---

> > ### Comment · Reviewer_mqQV · 2022-12-07
> > **IAS and EoS?**
> >
> > Dear Authors,
> >
> > Thank you for the clarifications. After extensive discussion with other (meta-)reviewers, I have one more question:
> >
> > Can I say that one of the main contributions is the following: for deterministic gradient descent, EoS has been studied and it is $\\{\theta:\|H(\theta)\|=2/\eta\\}$, and this works empirically studies stochastic gradient descent instead, for which EoS is replaced by IAS with boundary characterized by equation (11)? If yes, could you please comment on what happens as the minibatch size goes to the full batch size? Will (11) converge to EoS?

---

> > > ### Author Response · Authors · 2022-12-08
> > > **Our new EoS characterized with IAS is more accurate and general than the previous EoS.**
> > >
> > > Thank you for the additional comments!
> > >
> > > > **Q1.** Can I say that one of the main contributions is the following: for deterministic gradient descent, EoS has been studied and it is $\\{\theta: ||H(\theta)||=2/\eta \\}$, and this works empirically studies stochastic gradient descent instead, for which EoS is replaced by IAS with boundary characterized by equation (11)?
> > >
> > > Almost. We would like to elaborate more by answering the next question (Q2) together in the below.
> > > Before that we clarify the following three concepts:
> > > 1. Previous EoS (Cohen et al., 2021)
> > > $$\\{\theta: ||H(\theta)||=2/\eta\\}$$
> > > 2. Our new EoS (with IAS) for GD ($b=n$)
> > > $$\\{\theta: \frac{\text{tr}(HS\_n)}{\text{tr}(S\_n)}=2/\eta\\}=\\{\theta: ||H||_{S\_n}=2/\eta\\}\quad (\ast)$$
> > > 3. Our new EoS (with IAS) for SGD ($1\leq b\leq n$)
> > > $$\\{\theta: \frac{\text{tr}(HS\_b)}{\text{tr}(S\_n)}=2/\eta\\}=\\{\theta: ||H||_{S_b}\equiv\frac{\text{tr}(HS\_b)}{\text{tr}(S\_b)}=2\rho\_b/\eta\\}\quad \text{(11)}$$
> > >
> > > > **Q2.** If yes, could you please comment on what happens as the minibatch size goes to the full batch size? Will (11) converge to EoS?
> > >
> > > As you can see in 2 and 3, equation (11) reduces to equation ($\ast$) when $b=n$ since $\rho_n=1$ by definition.
> > > Thus, we can always say that "(11) converge to our new EoS (with IAS) in ($\ast$)" as the minibatch size $b$ goes to the full batch size $n$.
> > > Please kindly check Figure 33 which empirically verify this convergence.
> > >
> > > Now, comparing 1 and 2, they are equivalent iff the following Condition 1 holds:
> > > - **Condition 1**: the total training loss gradient $\nabla L$ is aligned with the top eigenvector $q\_1(H)$ with respect to the top eigenvalue $\lambda\_1$ of the Hessian $H$.
> > >
> > > When $\nabla L$ is aligned with the top eigenvector $q\_1(H)$, we have $||H||=||H||\_{S\_n}$ (i.e., $\lambda\_1=\frac{\text{tr}(HS\_n)}{\text{tr}(S\_n)}$) from the following
> > > $$ \text{tr}(HS\_n)= \text{tr}(H\nabla L\nabla L^\top)=\nabla L^\top \underbrace{H\nabla L}\_{\lambda\_1\nabla L}=\nabla L^\top \lambda\_1\nabla L=\lambda\_1 ||\nabla L||^2=\lambda\_1\text{tr}(S\_n)$$
> > > where the second equality holds because $\text{tr}(uv^\top)=\text{tr}(v^\top u)=v^\top u$, and the third equality comes from the alignment between $\nabla L$ and $q\_1(H)$.
> > > Therefore, 1 and 2 are equivalent when Condition 1 holds; thus, we can say that "(11) converge to the previous EoS when Condition 1 holds".
> > >
> > > However, it is hard to meet Condition 1 in the exact sense, even though the gradient $\nabla L$ is mostly observed to be in the subspace spanned by a few top eigenvectors of the Hessian [1].
> > > Therefore, the third equality becomes inequality since $|\cos(\nabla L, q\_1(H))|\leq 1$ as follows:
> > > $$ \text{tr}(HS\_n)= \text{tr}(H\nabla L\nabla L^\top)=\nabla L^\top H\nabla L \leq\nabla L^\top \lambda\_1\nabla L=\lambda\_1 ||\nabla L||^2=\lambda\_1\text{tr}(S\_n)$$
> > > and this leads to $||H||\_{S\_n}\leq ||H||$ (without requiring Condition 1).
> > > And thus, for GD, our new EoS with IAS, $||H||\_{S\_n}=2/\eta$, implies that $||H||\geq 2/\eta$, i.e., the sharpness $||H||$ hovers **above** $2/\eta$.
> > >
> > > To summarize, we have the following contributions:
> > > - Our new EoS (with IAS) is **more accurate** than previous EoS, and it can explain why $||H||$ does **not** hover **around** $2/\eta$ but it hovers somewhat **above** $2/\eta$ (see Figure 4).
> > > - Our new EoS (with IAS) is **more general** than previous EoS since it can be applied to SGD for a general $b\in[1,n]$.
> > >
> > > [1] Gur-Ari et al. (2018), Gradient Descent Happens in a Tiny Subspace (https://arxiv.org/abs/1812.04754)

---

> > > > ### Comment · Reviewer_mqQV · 2022-12-08
> > > > **Thanks for the clarification**
> > > >
> > > > I have increased my rating.

---

### Author Response · Authors · 2022-11-13
**General Comments (Revision / Contributions / Weaknesses / Short Answers)**

Dear all,

We thank the reviewers for their time and valuable feedback! During the discussion period, we hope to hear more questions or comments from the reviewers for further discussion to strengthen the paper.

**Revision**
In the revised version, the red colored texts have been modified from the original version, and the yellow highlighted texts have not changed from the original version, but we highlighted them to emphasize that they are commented in the reviews and the answers. To avoid confusion with the original numbering, (i) we added some Figures (Fig 40, 41) in the end and (ii) keep the original Figures as they were. After the discussion period, we will move them to the main text if it is necessary.

**Summarized Contributions**

- A clearer EoS (characterized with the IAS (interaction-aware sharpness))
- EoS for SGD
- A new scaling rule, LSSR (Linear and Saturation Scaling Rule)

**The major weak points raised by reviewers & the short answers**

1. (Reviewer mqQV) It requires more specific conditions about the conjectures.
    - We assume the progressive sharpening.
    - [link to the answer](https://openreview.net/forum?id=bH-kCY6LdKg&noteId=qkR3ai9rCRi)
1. (Reviewer mqQV) It requires explicit, concise and quantitative answer how much the sharpness is larger than $2/\eta$.
    - We provide a quantitative analysis in [the answer](https://openreview.net/forum?id=bH-kCY6LdKg&noteId=Ukk-WXJucI).
1. (Reviewer 9x87) Eq (16) is misleading ($\rho_b$ keep changing).
    - We can claim that SGD (with a smaller $b$) tends to find solutions with smaller $||H||\_{S\_b}$ (Fig 33).
    - $\rho_b$ does not change much at the EoS (Fig 41).
    - [link to the answer](https://openreview.net/forum?id=bH-kCY6LdKg&noteId=mnLljmag-RU)
1. (Reviewer mqQV/9x87) Insufficient discussions of prior work
    - We will add more discussions in the revised version based on the replies in the below.

(cf. **[W1]**: weak points, **[C1]**: comments, **[A1]**: answers)

---

> ### Author Response · Authors · 2022-11-16
> **Revision #1 uploaded**
>
> We uploaded the revision #1.

---

> > ### Author Response · Authors · 2022-12-05
> > **Waiting on additional comments**
> >
> > We are waiting on additional comments. Please check [the general comments](https://openreview.net/forum?id=bH-kCY6LdKg&noteId=klMdD85p2C) and the corresponding answers to each question.

---

### Decision · Program_Chairs · 2023-01-20

**Decision:**

Accept: poster

**Justification For Why Not Higher Score:**

The reviewers agree that the paper is a borderline paper however all three reviewers agreed unanimously that the positives going for the paper outweigh the negatives in the paper. The paper's experimentation can be more thorough and the presentation could be a lot more cogent.

**Justification For Why Not Lower Score:**

The reviewers agreed unanimously that the paper will be useful to the community especially expanding our understanding of the emerging phenomenon of edge of stability in the minibatch regime.

**Metareview: Summary, Strengths And Weaknesses:**

The paper proposes a new edge of stability phenomenon that applies to mini-batch stochastic gradient descent. This is achieved through a sharpness measure defined as Interaction-Aware Sharpness which measures the sharpness with respect to the batch gradient covariance. The reviewers discussed the paper in detail and while there were questions regarding the sufficiency of the experiments in terms of proving the hypothesis of the paper, the reviewers eventually agreed upon that while they can be improved they seem sufficient to support the claims. The reviewers also appreciated the core hypothesis of the paper and felt that it advances the community's knowledge in the newly emerging idea of the edge of stability. Overall I recommend acceptance based on the scores of the reviewers.

**Note From Pc:**

if the above contains the word "oral" or "spotlight" please see: "oral" presentation means -> notable-top-5% and "spotlight" means -> notable-top-25%. As stated in our emails, we are disassociating presentation type from AC recommendations

**Summary Of Ac-Reviewer Meeting:**

The meeting was largely focussed on the set of reviewers discussing the core aspects of the papers and clarifying various aspects that remained unclear to different reviewers. As highlighted one of the points discussed were the novelty aspects of the paper and the significance of the proposed hypothesis and whether the paper has done enough in terms of the experimentation to justify the hypothesis.